# Solving Differential Equations with Constrained Learning

**Viggo Moro**
University of Oxford
`viggo.moro@cs.ox.ac.uk`

**Luiz F. O. Chamon**
École polytechnique
`luiz.chamon@polytechnique.edu`

## ABSTRACT

(Partial) differential equations (PDEs) are fundamental tools for describing natural phenomena, making their solution crucial in science and engineering. While traditional methods, such as the finite element method, provide reliable solutions, their accuracy is often tied to the use of computationally intensive fine meshes. Moreover, they do not naturally account for measurements or prior solutions, and any change in the problem parameters requires results to be fully recomputed. Neural network-based approaches, such as physics-informed neural networks and neural operators, offer a mesh-free alternative by directly fitting those models to the PDE solution. They can also integrate prior knowledge and tackle entire families of PDEs by simply aggregating additional training losses. Nevertheless, they are highly sensitive to hyperparameters such as collocation points and the weights associated with each loss. This paper addresses these challenges by developing a *science-constrained learning* (SCL) framework. It demonstrates that finding a (weak) solution of a PDE is equivalent to solving a constrained learning problem with worst-case losses. This explains the limitations of previous methods that minimize the expected value of aggregated losses. SCL also organically integrates structural constraints (e.g., invariances) and (partial) measurements or known solutions. The resulting constrained learning problems can be tackled using a practical algorithm that yields accurate solutions across a variety of PDEs, neural network architectures, and prior knowledge levels without extensive hyperparameter tuning and sometimes even at a lower computational cost.

## 1 INTRODUCTION

(Partial) differential equations (PDEs) are key tools in science and engineering, playing a central role in the solution of inverse problems, systems engineering, and the description of natural phenomena (Lustig et al., 2008; Potter et al., 2010; Molesky et al., 2018; Evans, 2010). As such, a variety of numerical methods have been developed to approximate their solutions, such as the well-known finite element method (FEM). Despite their celebrated precision and approximation guarantees, these methods provide solutions to a single PDE at a time. Any change to the problem, from boundary condition to mesh size, requires the solution to be recomputed. They are therefore unable to incorporate prior knowledge, such as real-world measurements or known solutions to similar equations (Brenner & Scott, 2007; LeVeque, 2007; Katsikadelis, 2016).

Methods based on neural networks (NNs), such as physics-informed NNs (PINNs) (Lagaris et al., 1998; Raissi et al., 2019; Lu et al., 2021b) and neural operators (NOs) (Li et al., 2021; Lu et al., 2021a; Rahman et al., 2023), have been developed with these challenges in mind. Rather than discretizing the PDE, they directly fit a NN to its solution. They can therefore be trained to simultaneously solve entire families of PDEs and interpolate known solutions by simply incorporating additional losses to their training objectives (Li et al., 2021; Cho et al., 2024; Li et al., 2024). Yet, these methods are highly sensitive to hyperparameters such as the weights used to combine the training losses and the collocation points used to evaluate the PDE residuals, which often leads to low quality or trivial solutions (Krishnapriyan et al., 2021; Wight & Zhao, 2021; Wang et al., 2022b;a). This has prompted a variety of heuristics to be proposed based on *ad hoc* weight updates (Wang et al., 2021a; Maddu et al., 2022) and adaptive or causal sampling Nabian et al. (2021); Krishnapriyan et al. (2021); McClenny & Braga-Neto (2023); Penwarden et al. (2023) (see Appendix G for further related works).

This paper shows that these limitations are not methodological, but epistemological. It is not an issue of *how* the problem is solved, but *which* problem is being solved. To do so, it

- proves that obtaining a (weak) solution of a PDE requires solving a constrained learning problem with worst-case losses (Prop. 3.1), i.e., it is not enough to use either constrained formulations (Lu et al., 2021b; Basir & Senocak, 2022) *or* worst-case losses (Wang et al., 2022a; Daw et al., 2023);

- incorporates prior scientific knowledge on the structure (e.g., invariance) and value (e.g., measurements, set points) of the solution without resorting to specialized models or data transforms (Sec. 3). We therefore dub this approach *science-constrained learning* (SCL);

- develops a practical algorithm that foregoes the careful selection of loss weights and collocation points. Contrary to other methods, it explicitly approximates the (weak) solution of PDEs and yields reliability metrics that capture the difficulty of fitting specific PDE parameters and/or data points (Sec. 4);

- illustrates the effectiveness (accuracy and computational cost) of SCL (Sec. 5) for a diverse set of PDEs, NN architectures (e.g., MLPs and NOs), and problem types (solving a single or a parametric family of PDEs; interpolating known solutions; identifying settings that are difficult to fit).

## 2 PROBLEM FORMULATION

### 2.1 BOUNDARY VALUE PROBLEMS

Consider a (bounded, connected, open) region $\Omega \subset \mathbb{R}^d$ with (smooth) boundary $\partial\Omega$ and a (partial) differential operator $D_\pi$ with coefficients $\pi \in \Pi \subset \mathbb{R}^q$ defined on the domain $\mathcal{D} = \Omega \times (0, T]$. Here, $\pi$ captures a (finite) set of parameters of the phenomenon, such as the diffusion rate or viscosity. Given a space $\mathcal{F}$ of functions mapping $\mathcal{D}$ to $\mathbb{R}$, we define a boundary value problem (BVP) as

$$
\begin{aligned}
\text{find} \quad & u \in \mathcal{F} \\
\text{such that} \quad & D_\pi[u](x, t) = \tau(x, t), \quad (x, t) \in \Omega \times (0, T] \quad &\text{(PDE)} \\
& u(x, 0) = h(x, 0), \qquad x \in \bar{\Omega} \quad &\text{(IC)} \\
& u(x, t) = h(x, t), \qquad (x, t) \in \partial\Omega \times (0, T] \quad &\text{(BC)}
\end{aligned}
\tag{BVP}
$$

where $\tau : \mathcal{D} \to \mathbb{R}$ is a *forcing function* and $h : \mathcal{B} \to \mathbb{R}$ describes the boundary (BC) and initial (IC) conditions over $\mathcal{B} = (\bar{\Omega} \times \{0\}) \cup (\partial\Omega \times (0, T])$ for $\bar{\Omega} = \Omega \cup \partial\Omega$. In what follows, we always consider the initial condition as part of the boundary conditions and refer to them jointly as BC. Note that the developments in this paper also apply to formulations of (BVP) involving other BCs (e.g., *Neumann*, *periodic*), parameterized BCs, and vector-valued PDEs. We showcase a variety of phenomena that can be described by (BVP) in App. A and refer to, e.g., (Evans, 2010), for a more detailed treatment.

In general, a (strong) solution of (BVP) need not exist in any function space $\mathcal{F}$. This motivates the rise of relaxations such as the *weak formulation* that replaces the pointwise equation (PDE) in (BVP) by the integral equation[1]

$$
\int_\mathcal{D} D_\pi[u](x, t)\varphi(x, t)\, dxdt = \int_\mathcal{D} \tau(x, t)\varphi(x, t)\, dxdt, \quad \text{for all } \varphi \in \mathcal{T}.
\tag{1}
$$

The $\varphi$ are known as *test functions* and $\mathcal{T}$ is typically taken to be a *Sobolev space* due to its natural compatibility with this setting (see App. B for further details). The name *weak formulation* comes from the fact that a solution of (BVP) (when it exists) is also a solution of (1), although the converse is not necessarily true. Indeed, (1) allows for a wider range of solutions with less stringent regularity requirements, particularly with respect to continuity and differentiability (Evans, 2010). The BCs of (BVP) can often be homogeneized (i.e., $h \equiv 0$) and imposed implicitly through $\mathcal{T}$, thus fully describing its weak solution by (1) (Brenner & Scott, 2007; LeVeque, 2007; Katsikadelis, 2016).

### 2.2 SOLVING BOUNDARY VALUE PROBLEMS

There exists a wide range of numerical methods for solving BVPs, most of which rely on discretizing (BVP) (e.g., finite difference method, FDM) or its weak formulation (e.g., FEM). Their

---

[1]Typically, the weak formulation only considers the spatial domain $x$, handling $t$ separately using time-stepping methods (Thomee, 2013). Still, (1) is not uncommon and can be used to obtain weak formulations for parabolic PDEs (see, e.g., (Knabner & Angerman, 2003; Evans, 2010; Steinbach & Zank, 2020)).

well-established approximation guarantees, accuracy, and stable implementations make them ubiquitous in scientific and engineering applications (Brenner & Scott, 2007; LeVeque, 2007; Katsikadelis, 2016). These classical methods, however, only tackle one BVP at a time and do not naturally incorporate prior knowledge, such as measurements or known (partial) solutions. These challenges can be addressed by directly parameterizing the solution $u$ of BVPs, most notably using NNs. While this approach may not achieve the precision of classical methods (Krishnapriyan et al., 2021; Wight & Zhao, 2021; Grossmann et al., 2024; McGreivy & Hakim, 2024), they are able to simultaneously provide solutions for entire families of BVPs and extrapolate new solutions from existing ones. Generally speaking, these methods can be separated into *unsupervised*, that seek to solve (BVP) directly, and *supervised*, that leverage previously computed or measured solutions.

**Unsupervised methods.** These approaches train a model $u_\theta : \mathcal{D} \times \Pi \to \mathbb{R}$ with parameters $\theta \in \Theta \subset \mathbb{R}^p$ (e.g., a multilayer perceptron, MLP) to fit (BVP). This is the case, for instance, of physics-informed neural networks (PINNs) that train $u_\theta$ by solving, for fixed weights $\mu_{\text{pde}}, \mu_{\text{bc}} \geq 0$,

$$\underset{\theta \in \Theta}{\text{minimize}} \ \mu_{\text{pde}}\ell_{\text{pde}}(\theta) + \mu_{\text{bc}}\ell_{\text{bc}}(\theta), \tag{PI}$$

$$\ell_{\text{pde}}(\theta) \triangleq \sum_{i=1}^{I} \left[ \frac{1}{M} \sum_{m=1}^{M} \left( D_{\pi_i}[u_\theta(\pi_i)](x_m, t_m) - \tau(x_m, t_m) \right)^2 \right], \quad (x_m, t_m) \in \mathcal{D}, \ \pi_i \in \Pi,$$

$$\ell_{\text{bc}}(\theta) \triangleq \frac{1}{N} \sum_{n=1}^{N} \left( u_\theta(x_n, t_n) - h(x_n, t_n) \right)^2, \quad\quad\quad (x_n, t_n) \in \mathcal{B}.$$

We write $u_\theta(\pi)(x, t)$ to emphasize that we evaluate $u_\theta(\pi)$, which approximates the solution of (BVP) with coefficients $\pi$, at $(x, t) \in \mathcal{D}$. In practice, the input of the model $u_\theta$ is simply $(x, t, \pi) \in \mathcal{D} \times \Pi$. The losses $\ell_{\text{pde}}$ and $\ell_{\text{bc}}$ promote the requirements in (PDE) and (IC)–(BC) from (BVP), respectively, although the former is computed using automatic differentiation rather than discretization. Though the majority of PINNs target a single BVP, i.e., $I = 1$ in (PI), their extension to parameterized families of BVPs has been explored (Cho et al., 2024).

**Supervised methods.** Rather than directly solving the BVP, these approaches fit the model $u_\theta$ to a set of (partial) solutions $u_n^\dagger$ of (BVP). In this setting, $u_\theta$ is often a neural operator (NO) capable of handling infinite-dimensional (functional) inputs and outputs, such as forcing functions $\tau$ and ICs $h(x, 0)$. Given, e.g., forcing-solution pairs $(\tau_j, u_j^\dagger)$, these NOs are trained by solving

$$\underset{\theta \in \Theta}{\text{minimize}} \ \frac{1}{J} \sum_{j=1}^{J} \left\| u_\theta(\tau_j) - u_j^\dagger \right\|_{L_2(\mathcal{D})}^2. \tag{PII}$$

In practice, the functions $\tau_n$ and $u_n^\dagger$ are discretized (in the time or spectral domain) to enable computations (Li et al., 2021; Lu et al., 2021a; Hao et al., 2023; Wei & Zhang, 2023). While it is not uncommon to combine (PI) and (PII), these semi-supervised methods typically rely on MLPs (Raissi et al., 2019; Lu et al., 2021b), since it can be challenging to evaluate $D[u_\theta]$ for NOs (Li et al., 2024).

**Limitations.** Though effective in many applications, these methods are very sensitive to their hyperparameters. Indeed, the choice of collocation points $(x, t)$, PDE coefficients $\pi_i$, and weights $\mu_{\text{pde}}, \mu_{\text{bc}}$ affect both the quality and computational complexity of (PI) (Krishnapriyan et al., 2021; Wight & Zhao, 2021; Wang et al., 2021a). The same holds for the discretization of the objective in (PII) (Li et al., 2021; Hao et al., 2023; Wei & Zhang, 2023). Supervised methods face the additional challenge that acquiring the PDE solutions $\{u_n^\dagger\}$ used in (PII) can be expensive (relying on, e.g., classical methods) and it is challenging to obtain good performance from small datasets. This issue is aggravated by the heterogeneous difficulty of fitting each solution. A variety of heuristics for collocation points (Nabian et al., 2021; Daw et al., 2023; Penwarden et al., 2023), weights (Wang et al., 2021a; Maddu et al., 2022; McClenny & Braga-Neto, 2023), and PDE solutions (Pestourie et al., 2023; Musekamp et al., 2024) have been put forward to mitigate these challenges. Yet, they generally focus on specific "failure modes" or BVPs and seldom address the non-trivial interactions of these yperparameters, limiting their effectiveness.

## 3 SCIENCE-CONSTRAINED LEARNING

In this section, we argue that the challenges faced by previous NN-based BVP solvers arise not because of how (PI) and (PII) are solved, but because they are not the appropriate problems to solve in

the first place. To do so, we show that obtaining a (weak) solution of (BVP) is equivalent to solving a *constrained learning problem with worst-case losses*. Hence, it is not enough to use (approximations of) worst-case losses as in, e.g., (Wang et al., 2022a; Daw et al., 2023), *or* adapting loss weights as in, e.g., (Wang et al., 2021a; Lu et al., 2021b; McClenny & Braga-Neto, 2023). Building on this result, we show how to incorporate other forms of scientific knowledge that are not *mechanistic* (i.e., PDEs) without resorting to specialized models, including *structural* information (e.g., invariances) and *observations* (measurements, simulations) of the solution. The resulting *science-constrained learning* (SCL) problem accommodates a variety of knowledge settings, from unsupervised to supervised, and is amenable to a practical algorithm capable of effectively tackling entire families of BVPs and extrapolating solutions from existing ones (Sec. 4 and 5).

In the remainder of this paper, we use $u_\theta$ to refer to any parameterized model (MLP, NO, etc.). For clarity, we derive our results for a single BVP instance, omitting the dependence on $\pi$ and/or $\tau$. We consider these extensions at the end of the section.

**Mechanistic (PDE) knowledge.** We begin by showing how weak solutions of (BVP) can be obtained using constrained learning. To do so, we relax the BCs of (BVP) to relate the weak formulation (1) to a distributionally robust constraint. The BCs are then reintroduced using a constrained formulation. We start with the following proposition, where $W^{k,p}$ refers to the $(k,p)$-th order Sobolev space (see App. B) and $\mathcal{P}^2(\mathcal{S})$ denotes the space of square-integrable probability distributions supported on $\mathcal{S}$.

**Proposition 3.1.** *Let $u^\dagger \in W^{k',2}(\mathcal{D})$, where $k' \geq 1$ is the degree of the differential operator $D$, be such that $\sup_{\psi \in \mathcal{P}^2(\mathcal{D})} \mathbb{E}_{(x,t) \sim \psi}\left[\left(D[u^\dagger](x,t) - \tau(x,t)\right)^2\right] = 0$. If the dimension $d$ of $\Omega$ satisfies $d \leq 4k' - 1$, then $u^\dagger$ satisfies (1) with $\mathcal{T} = W^{k',2}(\mathcal{D})$.*

A proof is provided in Appendix B. The equality constraint suggested by Prop. 3.1 enforces (1), but does not impose the BCs of (BVP). Since they must hold for all $(x,t) \in \mathcal{B}$, it is more appropriate to incorporate them using a worst-case loss, namely, $\sup_{(x,t) \in \mathcal{B}}(u_\theta(x,t) - h(x,t))^2$, rather than an average loss as in (PI). As long as $(x,t) \mapsto (u_\theta(x,t) - h(x,t))^2$ is a function in $L^2$, this is equivalent to a distributionally robust loss similar to the one from Prop. 3.1 (see Appendix B). We therefore conclude that a weak solution of (BVP) is obtained by solving

$$
\begin{aligned}
\underset{\theta \in \Theta}{\text{minimize}} \quad & \sup_{\psi \in \mathcal{P}^2(\mathcal{B})} \mathbb{E}_{(x,t) \sim \psi}\left[\left(u_\theta(x,t) - h(x,t)\right)^2\right] \\
\text{subject to} \quad & \sup_{\psi \in \mathcal{P}^2(\mathcal{D})} \mathbb{E}_{(x,t) \sim \psi}\left[\left(D[u_\theta](x,t) - \tau(x,t)\right)^2\right] \leq \epsilon,
\end{aligned}
\tag{PIII}
$$

where $\epsilon \geq 0$ controls the trade-off between fitting the PDE and the BCs when $u_\theta$ is not expressive enough to satisfy both.

Prop. 3.1 elucidates the challenges arising from the choice of collocation points in the unsupervised approach (PI), most notably PINNs. Indeed, it is not enough to use a fixed distribution (e.g., uniform): satisfying (1) requires training against all distributions $\psi \in \mathcal{P}^2$. The use of worst-case losses in a constrained formulation is what makes (PIII) considerably different from previous adaptive sampling methods and loss-weighting schemes. In fact, contrary to previous approaches, by (approximately) solving (PIII) (as detailed in Sec. 4), we indeed (approximately) solve (BVP). At the same time, Prop. 3.1 establishes a limitation of learning-based solvers by restricting the smoothness of their solutions (essentially, solutions in $W^{(d+1)/4,2}$). This can be an issue for large-scale dynamical systems, such as those found in smart grid applications, or when transforming higher-order PDEs in higher-dimensional first-order systems. While Prop. 3.1 describes a sufficient condition for $u^\dagger$ to satisfy (1), a necessary condition can be obtained by restricting $\mathcal{P}^2$ to sufficiently smooth distributions, namely, those belonging to a Sobolev space.

**Structural knowledge.** The constrained form of (PIII) suggests that other information can be incorporated as long as they can be formulated as learning objectives, i.e., statistical losses. This is the case of certain forms of structural knowledge. Indeed, it is often possible to obtain information about the structure of the solution of a BVP, such as invariances or symmetries, without explicitly solving it (Olver, 1979; Akhound-Sadegh et al., 2023). While this structural information is already

encoded in (PIII), using it explicitly can reduce training time as well as the number of both collocation points and/or observations [as in (PII)] needed. It also helps ensure the physical validity of outcomes by explicitly avoiding degenerate solutions, a common failure mode of unsupervised methods such as PINNs (Krishnapriyan et al., 2021) (we do not observe such issues with (PIII), see Sec. 5.1). Structural knowledge can also be used to remove solution ambiguities, e.g., for the eikonal equation that is invariant to the sign of the solution (see App. A).

While structural knowledge can sometimes be incorporated into the model $u_\theta$, e.g., using equivariant architectures (Cohen & Welling, 2016; Batzner et al., 2022), it can also be imposed as a worst-case constraint. This is convenient for when such models are intricate to design. Consider, for example, a (finite) invariance group $\mathcal{G}$ whose elements $\gamma_i$ act on the domain $(x,t) \in \mathcal{D}$ such that $u^\dagger(x,t) = u^\dagger[\gamma_i(x,t)]$, where $u^\dagger$ is a solution of (BVP). This invariance can be enforced by

$$\sup_{\psi \in \mathcal{P}^2(\mathcal{D})} \mathbb{E}_{(x,t) \sim \psi}\left[\left(u_\theta(x,t) - u_\theta[\gamma_i(x,t)]\right)^2\right] \leq \epsilon, \quad \gamma_i \in \mathcal{G}. \tag{2}$$

Notice that we use the same distributionally robust formulation as for the BCs in (PIII). Similar constraints can be constructed for other structures, such as equivariance. In contrast to (PIII), where we want $\epsilon \approx 0$, it can be beneficial to use a larger values in (2) to accommodate models $u_\theta$ that cannot fully capture the solution invariances.

**Observational knowledge.** In addition to mechanistic (i.e., PDEs) and structural knowledge, we also consider (partial, noisy) observations of the BVP solution, obtained either via classical methods (e.g., FEM) or real-world measurements. Although this type of information is commonly associated with NOs, seen as they are typically trained in a supervised manner as in (PII), it is not limited to that architecture. Given observations $u_j^\dagger$, $j = 1, \dots, J$, of solutions of (BVP), we may formulate constraints of the kind

$$\mathbb{E}_{(x,t) \sim \mathfrak{m}}\left[\left(u_\theta(x,t) - u_j^\dagger(x,t)\right)^2\right] \leq \epsilon, \quad j = 1, \dots, J, \tag{3}$$

where $\mathfrak{m}$ is some distribution (typically uniform) of points on $\mathcal{D}$. Note that (3) is simply a different way of writing the $L^2$-norm from the objective of (PII). However, rather than averaging $L^2$ losses, (3) constraints the maximum error across data points. By considering each sample individually, it accounts for the heterogeneous difficulty of fitting them and enables the tolerance $\epsilon$ to be adjusted individually for each observation, e.g., using larger values for noisier samples.

**Science-constrained learning.** Combining (PIII) with (2) and (3), we are able to formulate a general SCL problem accommodating all knowledge sources considered so far. Explicitly,

$$\begin{aligned}
&\underset{\theta \in \Theta}{\text{minimize}} && \sup_{\psi \in \mathcal{P}^2(\mathcal{B})} \mathbb{E}_{(x,t) \sim \psi}\left[\left(u_\theta(x,t) - h(x,t)\right)^2\right] && \text{(M)} \\
&\text{subject to} && \sup_{\psi \in \mathcal{P}^2(\mathcal{D})} \mathbb{E}_{(x,t) \sim \psi}\left[\left(D[u_\theta](x,t) - \tau(x,t)\right)^2\right] \leq \epsilon_{\text{pde}} && \text{(M)} \\
& && \sup_{\psi \in \mathcal{P}^2(\mathcal{D})} \mathbb{E}_{(x,t) \sim \psi}\left[\left(u_\theta(x,t) - u_\theta[\gamma_i(x,t)]\right)^2\right] \leq \epsilon_{\text{s}}, \quad \gamma_i \in \mathcal{G} && \text{(S)} \\
& && \mathbb{E}_{(x,t) \sim \mathfrak{m}}\left[\left(u_\theta(x,t) - u_j^\dagger(x,t)\right)^2\right] \leq \epsilon_{\text{o}}, \quad j = 1, \dots, J. && \text{(O)}
\end{aligned}$$

(SCL)

Note that any subset of the constraints in (SCL) can be used depending on the available information. The objective of (SCL) is also not restricted to the BCs and can be replaced by any of the other terms. In fact, (SCL) can be formulated without an objective, i.e., as a feasibility problem.

It is straightforward to extend (SCL) to simultaneously solve a parameterized family of BVPs. However, rather than discretizing the parameter space as in (PI), we rely on a worst-case formulation

that considers all of its possible values rather than only a finite subset. Explicitly, we rewrite (SCL) as

$$\underset{\theta \in \Theta}{\text{minimize}} \quad \underset{\psi \in \mathcal{P}^2}{\sup} \; \mathbb{E}_{(x,t,\pi) \sim \psi, \, \tau \sim \mathfrak{p}} \left[ \left( u_\theta(\pi, \tau)(x, t) - h(x, t) \right)^2 \right] \tag{M}$$

$$\text{subject to} \quad \underset{\psi \in \mathcal{P}^2}{\sup} \; \mathbb{E}_{(x,t,\pi) \sim \psi, \, \tau \sim \mathfrak{p}} \left[ \left( D_\pi[u_\theta(\pi, \tau)](x, t) - \tau(x, t) \right)^2 \right] \le \epsilon_{\text{pde}} \tag{M}$$

$$\underset{\psi \in \mathcal{P}^2}{\sup} \; \mathbb{E}_{(x,t,\pi) \sim \psi, \, \tau \sim \mathfrak{p}} \left[ \left( u_\theta(\pi, \tau)(x, t) - u_\theta(\pi, \tau)\left[ \gamma_i(\pi)(x, t) \right] \right)^2 \right] \le \epsilon_{\text{s}}, \gamma_i \in \mathcal{G} \tag{S}$$

$$\mathbb{E}_{(x,t) \sim \mathfrak{m}} \left[ \left( u_\theta(\pi_j, \tau_j)(x, t) - u_j^\dagger(x, t) \right)^2 \right] \le \epsilon_{\text{o}}, \quad j = 1, \dots, J, \tag{O}$$

$$\tag{SCL$'$}$$

where $\mathfrak{m}, \mathfrak{p}$ are fixed distributions (e.g., uniform), $u_j^\dagger$ is a solution of (BVP) with coefficients $\pi_j$ and forcing function $\tau_j$, and the invariance $\gamma_i$ is now parametrized to account for the fact that its action may depend on $\pi$ (e.g., translation invariance with different strides). Note that the $\psi$ are now supported on $\mathcal{B} \times \Pi$ or $\mathcal{D} \times \Pi$, which we omit in (SCL$'$) for clarity. Hence, they target not only BVPs (parameters $\pi$) that are hard to fit, but also the regions of the domain responsible for this difficulty, enabling performances that would require fine discretizations (see Sec. 5). Yet, this approach is not directly applicable to infinite-dimensional parameters (e.g., $\tau$) as it requires sampling from a function space. We leave this extension for future work, considering here a fixed distribution $\mathfrak{p}$.

In the next section, we develop a practical algorithm to tackle (SCL) and (SCL$'$) by (i) leveraging non-convex duality results from constrained learning (Chamon & Ribeiro, 2020; Chamon et al., 2023) and (ii) deriving explicit approximations of the suprema over $\psi$ from which we can sample efficiently.

## 4 ALGORITHM

To develop a practical algorithm for (SCL) [and (SCL$'$)], we need to overcome the fact that it is (i) a non-convex constrained optimization problem involving (ii) worst-case losses. A typical approach to (i) is to combine the all losses as penalties into a single training objective as in (PI). Though penalties and constraints are essentially equivalent in convex optimization (strong duality, (Bertsekas, 2009)), this is not the case in the non-convex setting of (SCL). Hence, regardless of how the weights $\mu$ in (PI) are adapted (e.g., Wang et al. (2021a); Wight & Zhao (2021); Lu et al. (2021b); Basir & Senocak (2022)), it need not provide a solution of (SCL).

We overcome this issue by first tackling (ii) using the following proposition:

**Proposition 4.1.** *Let $z \mapsto \ell(z) \in L^2$. Then, for all $\delta > 0$ there exists $\alpha < \sup_z \ell(z)$ such that $\sup_{\psi \in \mathcal{P}^2} \mathbb{E}_{z \sim \psi} \left[ \ell(z) \right] \le \mathbb{E}_{z \sim \psi_\alpha} \left[ \ell(z) \right] + \delta$, where $\mathcal{P}^2 \ni \psi_\alpha(z) \propto \left[ \ell(z) - \alpha \right]_+$ for $[a]_+ = \max(0, a)$.*

A proof based on Robey* et al. (2021) can be found in App. B. Prop. (4.1) shows that the worst-case losses in (SCL)/(SCL$'$) can be approximated arbitrarily well by an expectation with respect to $\psi_\alpha$, a distribution proportional to a truncation of the underlying loss. For clarity, we consider (SCL) with only constraint (M), but similar manipulations hold for the constraints (S) and (O) as well as (SCL$'$). Explicitly, (SCL)(M) can be written as

$$\begin{aligned} \underset{\theta \in \Theta}{\text{minimize}} \quad & \mathbb{E}_{(x,t) \sim \psi_\alpha^{\text{bc}}} \left[ \left( u_\theta(x, t) - h(x, t) \right)^2 \right] \\ \text{subject to} \quad & \mathbb{E}_{(x,t) \sim \psi_\alpha^{\text{pde}}} \left[ \left( D[u_\theta](x, t) - \tau(x, t) \right)^2 \right] \le \epsilon, \end{aligned} \tag{PIV}$$

for $\psi_\alpha^{\text{bc}}(x, t) \propto \left[ \left( u_\theta(x, t) - h(x, t) \right)^2 - \alpha \right]_+$ and $\psi_\alpha^{\text{pde}}(x, t) \propto \left[ \left( D[u_\theta](x, t) - \tau(x, t) \right)^2 - \alpha \right]_+$ supported on $\mathcal{B}$ and $\mathcal{D}$ respectively.

Observe that (PIV) now has the form of a constrained learning problem, i.e., a constrained optimization problem with statistical losses. We can therefore use non-convex duality results from (Chamon & Ribeiro, 2020; Chamon et al., 2023; Elenter et al., 2024) to show that, under typical conditions from (unconstrained) learning theory and for rich enough parametrization, its solution can be approximated by solving the *empirical dual problem*

$$\underset{\lambda \ge 0}{\text{maximize}} \; \underset{\theta \in \Theta}{\min} \; \hat{L}(\theta, \lambda), \tag{DIV}$$

where $\hat{L}$ is the *empirical Lagrangian* of (PIV) based on samples $(x_n^{\mathrm{bc}}, t_n^{\mathrm{bc}}) \sim \psi_\alpha^{\mathrm{bc}}$ and $(x_n^{\mathrm{pde}}, t_n^{\mathrm{pde}}) \sim \psi_\alpha^{\mathrm{pde}}$, namely,

$$
\begin{aligned}
\hat{L}(\theta, \lambda) &\triangleq \frac{1}{N_{\mathrm{bc}}} \sum_{n=1}^{N_{\mathrm{bc}}} \left( u_\theta(x_n^{\mathrm{bc}}, t_n^{\mathrm{bc}}) - h(x_n^{\mathrm{bc}}, t_n^{\mathrm{bc}}) \right)^2 \\
&+ \lambda \left[ \frac{1}{N_{\mathrm{pde}}} \sum_{n=1}^{N_{\mathrm{pde}}} \left( D[u_\theta](x_n^{\mathrm{pde}}, t_n^{\mathrm{pde}}) - \tau(x_n^{\mathrm{pde}}, t_n^{\mathrm{pde}}) \right)^2 - \epsilon \right].
\end{aligned}
\tag{4}
$$

Contrary to previous approaches based on (PI), such as (Wang et al., 2021a; Wight & Zhao, 2021; Daw et al., 2023), (DIV) truly approximates the solution of (BVP). Indeed, (Chamon et al., 2023, Thm. 1) and (Elenter et al., 2024, Thm. 3.1) guarantee that solutions of (DIV) are near-optimal and near-feasible for (PIV) and, in view of Prop. 3.1 and (4.1), (BVP) (see App. D for details). It is worth noting that this is only possible because (PIV) is a *statistical* problem. Although similar Lagrangian formulations have been used in Lu et al. (2021b); Basir & Senocak (2022), they deal with deterministic constrained problem (fixed collocation points) for which this duality does not hold.

From a practical perspective, (DIV) does not require extensive hyperparameter tuning [such as $\mu$ in (PI)], seen as $\lambda$ is an optimization variable. What is more, despite non-convexity, the duality between (PIV) and (DIV) allows the solution $\lambda^\star$ of (DIV) to be interpreted as a sensitivity of the objective (BC residuals) to small relaxations of $\epsilon$ (Chamon & Ribeiro, 2020; Chamon et al., 2023; Hounie et al., 2023b). This information can be used to evaluate the fit of noisy measurements in (SCL)(O) or the reliability of solutions for different parameters $\pi$ in (SCL$'$) (see Sec. 5). Finally, (DIV) is amenable to practical algorithms such as dual ascent, which updates $\lambda_0 = 0$ as

$$
\lambda_{k+1} = \lambda_k + \eta \left[ \frac{1}{N_{\mathrm{pde}}} \sum_{n=1}^{N_{\mathrm{pde}}} \left( D[u_{\theta_k^\dagger}](x_n^{\mathrm{pde}}, t_n^{\mathrm{pde}}) - \tau(x_n^{\mathrm{pde}}, t_n^{\mathrm{pde}}) \right)^2 - \epsilon \right]_+, \text{ for } \theta_k^\dagger \in \underset{\theta \in \Theta}{\operatorname{argmin}} \ \hat{L}(\theta, \lambda_k).
\tag{5}
$$

Even if the empirical Lagrangian minimizer $\theta_k^\dagger$ is only computed approximately, (5) can be shown to converge to a neighborhood of a solution of (DIV) (Chamon et al., 2023; Elenter et al., 2024).

Still, to turn (5) into a practical algorithm, we need to obtain samples from $\psi_\alpha$, which is only known implicitly [see (PIV)]. Nevertheless, since the losses in (PIV) are non-negative, the $\psi_\alpha$ are smooth, fully-supported, square-integrable distributions for $\alpha = 0$. They are therefore amenable to be sampled using Markov Chain Monte Carlo (MCMC) techniques (Robert & Casella, 2004). We use the Metropolis-Hastings (MH) algorithm in our experiments since it avoids additional backward passes and higher-order derivatives resulting from differentiating $D_\pi$ while still providing strong empirical results (see App. C and Sec. 5). Exploring first-order methods (e.g., Langevin Monte Carlo) and algorithms adapted to discontinuous distributions (e.g., (Nishimura et al., 2020)) to enable faster convergence (reduce $N$) and better approximations (increase $\alpha$) is left for future work.

It is possible to replace $\psi_\alpha$ by a fixed distribution (e.g., uniform) or even fixed points for some of the constraints in (SCL)/(SCL$'$) at negligible accuracy costs. We find this is consistently the case for the BC objective, although we emphasize that this is application-dependent. Additionally, certain architectures, such as FNOs (Li et al., 2021), cannot make predictions at arbitrary points of the domain. It is then more appropriate to take $\psi_\alpha$ to be a uniform distribution over equispaced points.

The resulting method for solving (SCL$'$) is summarized in Alg. 1. Note that rather than obtaining Lagrangian minimizers as in (5), Alg. 1 alternates between optimizing for $\theta$ (step 8) and $\lambda$ (step 9). Such gradient descent-ascent schemes are commonly used in convex optimization (Arrow et al., 1958; Bertsekas, 2015), although their convergence is less well understood in non-convex settings Lin et al. (2020); Fiez et al. (2021); Yang et al. (2022). The convergence guarantees for (5) can be recovered by repeating step 8 ($\theta$-update) multiple times before updating $\lambda$. Note that Alg. 1 does not rely on training heuristics, such as adaptive or causal sampling Krishnapriyan et al. (2021); McClenny & Braga-Neto (2023); Daw et al. (2023); Penwarden et al. (2023); Wang et al. (2024), *ad hoc* weight updates (Wang et al., 2021a; Maddu et al., 2022), and conditional updates (Lu et al., 2021b; Basir & Senocak, 2022). Indeed, steps 4-7 are empirical estimates of expectations with respect to $\psi_0$ that themselves approximate the worst-case losses in SCL$'$ (Prop. 4.1). Steps 8-9 describe a traditional primal-dual (gradient descent-ascent) algorithm for solving problems such as (DIV), which itself yields approximate solutions of (SCL$'$) (Chamon & Ribeiro, 2020; Chamon et al., 2023; Elenter et al., 2024) and, consequently, (BVP) (Prop. 3.1).

---

**Algorithm 1** Primal-dual method for (SCL)

---

1: **Inputs**: Differential operator $D_\pi$, invariant transformations $\gamma_i \in \mathcal{G}$, observations set $(\pi_j, \tau_j, u_j^\dagger)$, parameterized model $u_{\theta_0}$, and $\lambda_0^{\mathrm{pde}} = \lambda_0^{s_i} = \lambda_0^{o_j} = 0$

2: **for** $k = 1, \ldots, K$

3: $\quad \ell_k^{\mathrm{bc}} = \dfrac{1}{N_{\mathrm{BC}}} \sum_{n=1}^{N_{\mathrm{BC}}} \left( u_{\theta_k}(\pi_n^{\mathrm{bc}})(x_n^{\mathrm{bc}}, t_n^{\mathrm{bc}}) - h(x_n^{\mathrm{bc}}, t_n^{\mathrm{bc}}) \right)^2, \qquad\qquad (x_n^{\mathrm{bc}}, t_n^{\mathrm{bc}}, \pi_n^{\mathrm{bc}}) \sim \psi_0^{\mathrm{BC}}$

4: $\quad \ell_k^{\mathrm{pde}} = \dfrac{1}{N_{\mathrm{pde}}} \sum_{n=1}^{N_{\mathrm{pde}}} \left( D_{\pi_n^{\mathrm{pde}}} \left[ u_{\theta_k}(\pi_n^{\mathrm{pde}}) \right](x_n^{\mathrm{pde}}, t_n^{\mathrm{pde}}) - \tau(x_n^{\mathrm{pde}}, t_n^{\mathrm{pde}}) \right)^2, \quad (x_n^{\mathrm{pde}}, t_n^{\mathrm{pde}}, \pi_n^{\mathrm{pde}}) \sim \psi_0^{\mathrm{PDE}}$

5: $\quad \ell_k^{s_i} = \dfrac{1}{N_s} \sum_{n=1}^{N_s} \left( u_{\theta_k}(\pi_n^{s_i})(x_n^{s_i}, t_n^{s_i}) - u_{\theta_k}(\pi_n^{s_i}) \left[ \gamma_i(\pi_n^{s_i})(x_n^{s_i}, t_n^{s_i}) \right] \right)^2, \qquad (x_n^{s_i}, t_n^{s_i}, \pi_n^{s_i}) \sim \psi_0^{\mathrm{ST}_i}$

6: $\quad \ell_k^{o_j} = \dfrac{1}{N_o} \sum_{n=1}^{N_o} \left( u_{\theta_k}(\pi_j, \tau_j)(x_n^{o_j}, t_n^{o_j}) - u_j^\dagger(x_n^{o_j}, t_n^{o_j}) \right)^2, \qquad\qquad (x_n^{o_j}, t_n^{o_j}) \sim \mathfrak{m}$

7: $\quad \theta_{k+1} = \theta_k - \eta_p \left[ \nabla_\theta \ell_k^{\mathrm{bc}} + \lambda^{\mathrm{pde}} \nabla_\theta \ell_k^{\mathrm{pde}} + \sum_{i=1}^{I} \lambda_k^{s_i} \nabla_\theta \ell_k^{s_i} + \sum_{j=1}^{J} \lambda_k^{o_j} \nabla_\theta \ell_k^{o_j} \right]$

8: $\quad \lambda_{k+1}^{\mathrm{pde}} = \left[ \lambda_k^{\mathrm{pde}} + \eta_d (\ell_k^{\mathrm{pde}} - \epsilon_{\mathrm{pde}}) \right]_+ ; \; \lambda_{k+1}^{s} = \left[ \lambda_k^{s} + \eta_d (\ell_k^{s} - \epsilon_s) \right]_+ ; \; \lambda_{k+1}^{o_j} = \left[ \lambda_k^{o_j} + \eta_d (\ell_k^{o_j} - \epsilon_o) \right]_+$

9: **end**

---

## 5 EXPERIMENTS

In this section, we showcase the use of SCL by training MLPs and FNOs (Li et al., 2021) to solve six PDEs (convection, reaction-diffusion, eikonal, Burgers', diffusion-sorption, and Navier-Stokes). We consider different subsets of constraints from (SCL)/(SCL′) to illustrate a variety of knowledge settings, but train only the most suitable model in each case since our goal is to illustrate the natural uses of SCL rather than exhaust its potential. Detailed descriptions are provided in the appendices, including BVPs (App. A), training procedures (App. E), and further results (App. F). Code to reproduce these experiments is available at `https://github.com/vmoro1/scl`. In the sequel, we use fixed points $(x, t)$ for the BC objective rather than $\psi_0^{\mathrm{BC}}$ to illustrate how computational complexity can be reduced without significantly affecting the results. We still use $\psi_0^{\mathrm{BC}}$ for $\pi$.

### 5.1 SOLVING A SPECIFIC BVP

We begin by solving (BVP) for fixed parameters $\pi$, forcing function $\tau$, and BCs. Though this may not be the best application for NN-based solvers [see, e.g., McGreivy & Hakim (2024)], it remains a valid demonstration. We use (SCL)(M) to train MLPs to solve convection, reaction-diffusion, and eikonal problems, comparing the results to PINNs [i.e., (PI) with weights chosen as in (Daw et al., 2023)]. All experiments use 1000 collocation points per epoch obtained by (PINN) sampling uniformly (we find this performs better than using fixed points as in, e.g., (Raissi et al., 2019; Lu et al., 2021b)), (R3) using the adaptive heuristic from (Daw et al., 2023), and (SCL) using MH after 4000 burn-in steps. Table 1 shows that SCL matches and often outperforms other methods, particularly

Table 1: Relative $L_2$ error for solving specific BVPs (average $\pm$ standard deviation across 10 seeds).

| | | PINN (PI) | R3 (Daw et al., 2023) | (SCL)(M) |
|---|---|---|---|---|
| **Convection** | $\beta = 30$ | $1.17 \pm 0.65\,\%$ | $0.999 \pm 0.53\,\%$ | $0.971 \pm 0.30\,\%$ |
| | $\beta = 50$ | $56.5 \pm 20\,\%$ | $29.0 \pm 33\,\%$ | $3.74 \pm 0.87\,\%$ |
| **React.-Diff.** | $(\nu, \rho) = (3, 3)$ | $0.745 \pm 0.014\,\%$ | $0.736 \pm 0.068\,\%$ | $1.82 \pm 0.74\,\%$ |
| | $(\nu, \rho) = (3, 5)$ | $79.6 \pm 0.27\,\%$ | $0.665 \pm 0.046\,\%$ | $0.762 \pm 0.11\,\%$ |
| **Eikonal** | | $87.1 \pm 39\,\%$ | $85.5 \pm 27\,\%$ | $9.95 \pm 1.9\,\%$ |

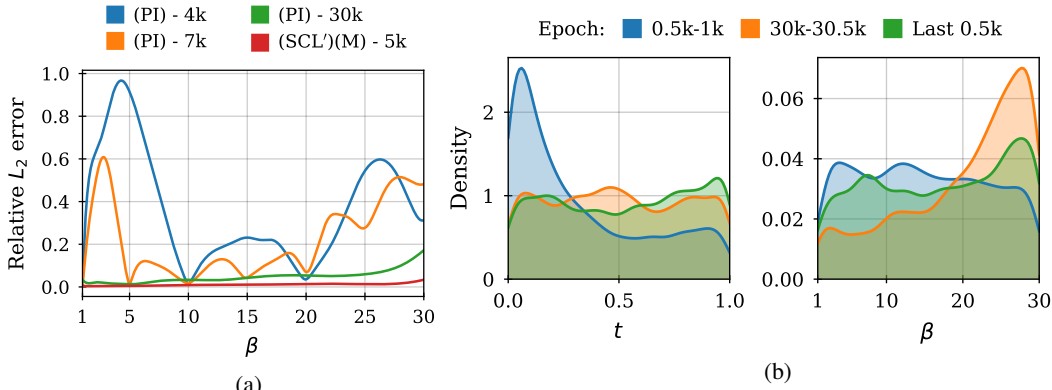

Figure 1: Solving parametric convection BVPs: (a) relative $L_2$ error vs. $\beta$ (legend reports number of differential operator evaluations per epoch). (b) Samples from $\psi_0^{\mathrm{PDE}}$ at different training stages.

in challenging scenarios (e.g., convection with $\beta = 50$ or non-linear eikonal). This is because it jointly adapts the weight of each loss and the points used to evaluate them during training (see Alg. 1). Although R3 also targets points with high PDE residuals, it does not approximate the worst-case loss needed to guarantee a (weak) solution (Prop. 3.1). The improved performance of SCL comes at a higher computational cost ($\approx 5\mathrm{x}$), although this is largely offset when solving parametric problems.

## 5.2    Solving parametric families of BVPs

A key advantages of NN-based approaches is their ability to solve entire families of BVPs at once. Consider fitting an MLP $u_\theta(x, t, \beta)$ to solve convection problems for $\beta \in [1, 30]$ using (SCL')(M) and (PI) as in Cho et al. (2024). We do not use their tailored "P$^2$INN" architecture, although it would be compatible with SCL. The $\pi_j = \beta_j$ are taken to be 4, 7, and 30 equispaced values in $[1, 30]$. Fig. 1a shows that (PI) can only achieve the error of (SCL')(M) for the finest discretization, at which point it evaluates the PDE loss 6 times more per epoch. The same pattern holds for reaction-diffusion (4–7 times) and 2D Helmholtz (4–5 times) problems (App. F). This effectiveness is due to the worst-case loss of (SCL')(M) jointly selecting $(x, t, \beta)$ to target challenging coefficients as well as the domain regions responsible for that difficulty. In fact, inspecting $\psi_0^{\mathrm{PDE}}$ at different stages of training (Fig. 1b) shows that SCL first fits the solution "causally," focusing on smaller values of $t$. While this has been proposed in (Krishnapriyan et al., 2021; Penwarden et al., 2023; Wang et al., 2024), it arises naturally by solving (SCL')(M). As training advances, however, Alg. 1 shifts focus to fitting higher convection speeds $\beta$. This occurs *without* any prior knowledge of the problems or manual tuning.

As we have argued, this is a use case for which SCL is particularly well-suited. Indeed, consider solving a 2D Helmholtz equation for parameters $(a_1, a_2) \in [1, 2]^2$ using an FEM solver with mesh size chosen to obtain a similar accuracy as SCL. Across 100 experiments, the average relative $L_2$ error for the FEM solver was 0.036 for an average runtime of 4.3 minutes per solution (see App. F). Hence, in the time it took to train the SCL model (31.1 hours), we could evaluate less than 440 parameters combinations using the FEM solver. This is 20 times less than the $10^4$ combinations used to evaluate the error of (SCL')(M) (average error 0.013 for a runtime of 4 minutes). It is worth noting that these numbers are for a highly-optimized FEM implementation (Baratta et al., 2023).

## 5.3    Leveraging invariance when solving BVPs

Next, we showcase how SCL can be used to overcome computational limitations or scarce mechanistic knowledge. Consider training an MLP to solve a convection BVP with $\beta = 30$ and periodic initial condition $h(0, x) = \sin(x)$. This problem is commonly used to showcase a "failure mode" of (PI) since it yields degenerate solutions when using fixed collocation points (Fig. 2a) (Krishnapriyan et al., 2021). Note that the constrained (SCL)(M) also fails when using fixed collocations points (Fig. 2b), even though its stochastic version finds accurate solutions (Table 1). One way to overcome this limitation is by leveraging additional knowledge. In this case, we know from the BVP structure that its solution must be periodic with period $\pi/15$. By incorporating this invariance using (SCL)(S), we

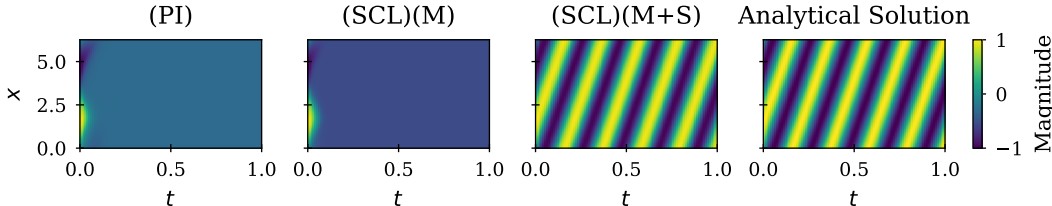

Figure 2: Using invariance in convection BVPs (BC and PDE losses in (PI) and (SCL) are evaluated using a *fixed set of collocation points*).

Table 2: Relative $L_2$ error on test set (average across 10 seeds, see App. F for standard deviation).

|  | $\nu$ | (PII) | (SCL)(O) |
|---|---|---|---|
| **Burgers'** | $10^{-3}$ | $0.0540\,\%$ | $0.0444\,\%$ |
| **Navier-Stokes** | $10^{-3}$ | $4.29\,\%$ | $3.31\,\%$ |
|  | $10^{-4}$ | $32.2\,\%$ | $29.9\,\%$ |
|  | $10^{-5}$ | $27.6\,\%$ | $26.0\,\%$ |
| **Diffusion-Sorption** |  | $0.274\,\%$ | $0.218\,\%$ |

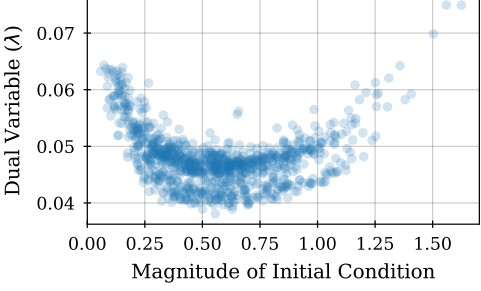

Figure 3: Final value of dual variables ($\lambda^{OB_j}$) vs. magnitude of IC for Burgers' equation.

can obtain accurate solutions despite our use of fixed collocation points for (SCL)(M) (Fig. 2c). Note that the worst-case loss in (SCL)(S) is fundamental to avoid degenerate solutions.

## 5.4 SUPERVISED SOLUTION OF BVPS

We conclude by exploring supervised methods that rely only on pairs of initial conditions $h_j(x, 0)$ and corresponding (weak) solutions $u_j^\dagger$. To showcase the versatility of SCL, we train FNOs rather than MLPs, fixing $\mathfrak{m}$ to a uniform distribution over a regular grid to accommodate their predictions. We also cast the SCL problem using only the observational constraints (SCL)(O), i.e., without any objective. In contrast to (PII), which minimizes the average error, this formulation enforces a maximum error of $\epsilon_o$ across samples. Though apparently minor, this leads to better prediction quality, as shown in Table 2. This occurs because the difficulty of fitting PDE solutions varies across ICs. While the average tends to emphasize the majority of "easy-to-fit samples," enforcing a maximum error gives more weight to challenging data points. This becomes clear in Fig. 3, which compares the magnitude of each IC in the training set with its final dual variables $\lambda^{OB_j}$ for the Burgers' equation. Immediately, we notice a trend where ICs with either small or large magnitudes appear harder to fit. This information can be leveraged to guide the collection of additional data points or improve the NO architecture. Indeed, any model improvement can immediately take advantage of SCL since it is (virtually) independent of the choice of $u_\theta$. The benefits of having $\lambda^{OB}$ are clear when predicting which IC is hard to fit is intricate, as is the case for the Navier-Stokes equation (see App. F).

## 6 CONCLUSION

This paper developed SCL, a technique for solving BVPs based on constrained learning. It demonstrated that finding (weak) solutions of PDEs is equivalent to solving constrained learning problems with worst-case losses, which also allows prior knowledge to be naturally incorporated to the solution of the BVP, e.g., structural constraints (e.g., invariances), real-world measurements, and previously known solutions. It developed a practical algorithm to tackle SCL problems and showcased its performance across a variety of PDEs and NN architectures. SCL not only yields accurate solutions, but also tackles many challenges faced by previous methods, such as extensive hyperparameter tuning, degenerate solutions, and fine discretizations of domain and/or coefficients. Future work includes exploring worst-case losses for functional parameters such as $\tau$ and the use of SCL for active learning.

ACKNOWLEDGMENTS

This work was partly funded by the Deutsche Forschungsgemeinschaft (DFG, German Research Foundation) under Germany's Excellence Strategy (EXC 2075-390740016). It was performed in part on the HoreKa supercomputer funded by the Ministry of Science, Research and the Arts Baden-Württemberg and by the Federal Ministry of Education and Research. Viggo Moro thanks the International Max Planck Research School for Intelligent Systems (IMPRS-IS) for their support and G-Research for supporting the travel costs.

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

# A   APPLICATIONS OF (BVP)

We showcase here a variety of phenomena that can be described using (BVP). The PDEs detailed in this section are used throughout our experiments to illustrate the performance of SCL.

## A.1   CONVECTION EQUATION

The one-dimensional convection equation models the transport of a scalar quantity $u(x, t)$, such as temperature or concentration, along the spatial dimension $x$. We consider the convection problem

$$\frac{\partial u(x,t)}{\partial t} + \beta\frac{\partial u(x,t)}{\partial x} = 0, \qquad\qquad (x,t) \in (0, 2\pi) \times (0, 1] \qquad (6a)$$

$$u(x,0) = \sin(x), \qquad\qquad x \in [0, 2\pi] \qquad (6b)$$

$$u(0,t) = u(2\pi, t), \qquad\qquad t \in (0, 1] \qquad (6c)$$

where the coefficient $\beta$ denotes the *convection speed*. Despite its use of periodic BCs, namely, (6c), this problem (and its variations) can be cast as a straightforward extension of (BVP).

## A.2   REACTION-DIFFUSION EQUATION

The one-dimensional reaction-diffusion equation describes a variety of phenomena, including chemical reactions, population dynamics, and heat transfer, depending on the form of its reaction term. In this paper, we consider the reaction-diffusion problem with periodic BC and Gaussian IC. Explicitly,

$$\frac{\partial u(x,t)}{\partial t} - \nu\frac{\partial^2 u(x,t)}{\partial x^2} = \rho u(x,t)\big(1 - u(x,t)\big), \qquad\qquad (x,t) \in (0, 2\pi) \times (0, 1] \qquad (7a)$$

$$u(x,0) = \exp\left(-\frac{1}{2}\Big(\frac{x-\pi}{\pi/4}\Big)^2\right), \qquad\qquad x \in [0, 2\pi] \qquad (7b)$$

$$u(0,t) = u(2\pi, t), \qquad\qquad t \in (0, 1] \qquad (7c)$$

where $\nu > 0$ and $\rho$ are the diffusion and reaction coefficients, respectively.

## A.3   EIKONAL EQUATION

The eikonal equation is encountered in many applications involving wave propagation, e.g., electromagnetism. It also describes the (signed) distance between any point $x \in \Omega$ and some fixed boundary $\partial S$, hence its usage in vision applications. In this case, we consider the BVP

$$\|\nabla u(x,y)\| = 1, \quad (x,y) \in \Omega \qquad (8a)$$

$$u(x,y) = 0, \quad (x,y) \in \partial S \qquad (8b)$$

where $\Omega = (-1, 1)^2$ and $\partial S$ is a complex shape, in our case, the gears figure from (Daw et al., 2023). In order to ensure that negative (positive) distances are assigned to the interior (exterior) of the shape, we add a structural constraint to the solution which enforces that $u$ is non-negative on the boundary of $\Omega$. Explicitly, we enforce that

$$u(x,y) \geq 0, \quad (x,y) \in \partial\Omega. \qquad (9)$$

This can be done by adding a structural constraint to (SCL), i.e., by replacing (SCL)(S) with a loss that induces (9), explicitly

$$\sup_{\psi \in \mathcal{P}^2(\partial\Omega)} \mathbb{E}_{(x,y) \sim \psi}\left[\big[-u_\theta(x,y)\big]_+\right] \leq \epsilon_s. \qquad (10)$$

In our experiments, we find that this constraint is not particularly difficult to enforce and that we can obtain good results by replacing the worst-case $\psi_\alpha$ by fixed, equispaced points on $\partial\Omega$.

## A.4   HELMHOLTZ EQUATION

The two-dimensional Helmholtz equation models wave propagation and vibration phenomena in various physical contexts. Here, we consider the problem

$$\nabla^2 u(x,y) + k^2 u(x,y) = \tau(x, y, \pi), \qquad\qquad (x,y) \in \text{interior}(S) \qquad (11a)$$

$$u(x,y) = \sin(\pi a_1 x)\sin(\pi a_2 y), \qquad\qquad (x,y) \in \partial S \qquad (11b)$$

where $\mathcal{S} = [0,1]^2$; $k > 0$ is the wave number; coefficients $\pi = (a_1, a_2)$ represent the spatial frequencies in the $x$ and $y$ directions, respectively; and the forcing function is $\tau(x, y, \pi) = \left(k^2 - \pi^2 a_1^2 - \pi^2 a_2^2\right)\left(\sin(\pi a_1 x)\sin(\pi a_2 y)\right)$.

## A.5 BURGERS' EQUATION

The one-dimensional Burgers' equation models the behavior of a scalar field $u(x, t)$ under the combined effects of nonlinear convection and diffusion. It is commonly used in fluid dynamics and traffic flow to describe shock waves and turbulence. In particular, we consider the following BVP with periodic BCs

$$\frac{\partial u(x,t)}{\partial t} + \frac{1}{2}\frac{\partial u^2(x,t)}{\partial x} = \nu\frac{\partial^2 u(x,t)}{\partial x^2}, \qquad (x,t) \in (0,1) \times (0,1] \tag{12a}$$

$$u(x,0) = h_0(x), \qquad x \in [0,1] \tag{12b}$$

$$u(0,t) = u(1,t), \qquad t \in (0,1] \tag{12c}$$

where $\nu > 0$ is the viscosity coefficient, which governs the strength of the diffusion term. We consider ICs $h_0$ from the same distribution as Li et al. (2021).

## A.6 DIFFUSION-SORPTION EQUATION

The diffusion-sorption equation models the transport of a scalar field $u(x, t)$ (e.g., a contaminant concentration) in a porous medium. It is commonly used in environmental science and chemical engineering. In terms of this PDE, we consider the following BVP

$$\frac{\partial u(x,t)}{\partial t} = \frac{\nu}{R\big(u(x,t)\big)}\frac{\partial^2 u(x,t)}{\partial x^2}, \qquad (x,t) \in (0,1) \times (0,500] \tag{13a}$$

$$u(x,0) = h_0(x), \qquad x \in [0,1] \tag{13b}$$

$$u(0,t) = 1, \quad u(1,t) = \nu\frac{\partial u(1,t)}{\partial x}, \qquad t \in (0,500] \tag{13c}$$

where $\nu = 5 \times 10^{-4}$ is the diffusion coefficient and $R$ is the retardation factor defined as

$$R(u) = 1 + \frac{(1-\phi)}{\phi}\rho_s k n_f u^{n_f - 1},$$

where $\rho_s = 2880$ is the bulk density, $\phi = 0.29$ is the medium porosity, $k = 3.5 \times 10^{-4}$ is Freundlich's sorption parameter, and $n_f = 0.874$ is Freundlich's exponent. We consider the distribution of ICs $h_0$ from Takamoto et al. (2022).

A notable aspect of (13) is its nonlinearity due to the dependence on $u(x, t)$ of the effective diffusion coefficient $\frac{\nu}{R(u)}$. What is more, it can become singular when $u = 0$, making this equation particularly challenging to solve.

## A.7 NAVIER-STOKES EQUATION

The two-dimensional, incompressible Navier-Stokes equation in vorticity form removes the pressure term and focuses on the dynamics of rotational flow. It is used to describe the local rotation of a fluid, i.e., the vorticity $\omega(x, t) = \nabla \times u(x, t)$, where $u$ is the two-dimensional velocity field and $\nabla \times f$ denotes the *curl* of $f$. Although vorticity is more challenging to model than velocity, it offers a deeper understanding of the flow dynamics. The velocity field can be recovered from the vorticity using Poisson's equation.

Explicitly, for $x = [x_1, x_2]^\top$, we consider the BVP

$$\frac{\partial \omega(x,t)}{\partial t} + u(x,t)^\top \nabla\omega(x,t) = \nu\nabla^2\omega + \tau(x), \qquad (x,t) \in (0,1)^2 \times (0,T] \tag{14a}$$

$$\omega(x,0) = \omega_0, \qquad x \in [0,1]^2, \tag{14b}$$

$$\nabla \cdot u(x,t) = 0, \qquad (x,t) \in (0,1)^2 \times (0,T] \tag{14c}$$

where $\nu > 0$ is the viscosity coefficient and $\nabla \cdot f$ denotes the divergence of $f$. The forcing function is taken to be

$$\tau(x) = 0.1\big(\sin\big(2\pi(x_1 + x_2)\big) + \cos\big(2\pi(x_1 + x_2)\big)\big)$$

and the ICs $\omega_0$ are taken from the same distribution as in Li et al. (2021). We consider three settings, namely, $\nu = 10^{-3}$ with $T = 50$, $\nu = 10^{-4}$ with $T = 30$, and $\nu = 10^{-5}$ and $T = 20$.

# B  WEAK SOLUTIONS AND ROBUST LEARNING

The space $\mathcal{T}$ of test functions plays a fundamental role when defining the weak formulations (1). Typically, it is chosen to be a Sobolev space due to its natural compatibility with BVPs and the fact that it leads to less stringent differentiability requirements on $u$. It therefore overcomes the main issues with the strong formulation (BVP). A Sobolev space consists of functions in some Lebesgue space $L^p$ whose *weak derivatives* are also in $L^p$. Recall that $L^p$ is the space of $p$-integrable functions. To define a Sobolev space, we therefore need to start by defining a weak derivative.

A locally integrable function $f$ defined on an open set $\mathcal{Z} \subset \mathbb{R}^d$ is weakly differentiable with respect to $z_i$ if there exists $Df$ also locally integrable (i.e., $f, Df \in L^1_{\text{loc}}(\mathcal{Z})$) such that

$$\int_{\mathcal{Z}} Df(z)\xi(z)dz = -\int_{\mathcal{D}} \varphi(z)\frac{\partial \xi(z)}{\partial z_i}dz, \quad \text{for all } \xi \in C^\infty_c(\mathcal{Z}), \tag{15}$$

where $C^\infty_c(\mathcal{Z})$ is the space of infinitely differentiable, compactly supported functions. We say $Df$ is the weak derivative of $f$. To generalize (15) to higher-order derivatives, consider the multi-index $\alpha = (\alpha_1, \ldots, \alpha_d)$ to be a $d$-tuple of non-negative integers and let $|\alpha| = \sum_{i=1}^d \alpha_i$. We then define the $\alpha$-weak derivatives of $f$, denoted $D^\alpha f$, as

$$\int_{\mathcal{Z}} D^\alpha f(z)\xi(z)dz = -\int_{\mathcal{D}} \varphi(z)\frac{\partial^{|\alpha|}\xi(z)}{\partial z_1^{\alpha_1} \cdots \partial z_d^{\alpha_d}}dz, \quad \text{for all } \xi \in C^\infty_c(\mathcal{Z}). \tag{16}$$

We can now define what we mean by Sobolev space (Evans, 2010).

**Definition B.1** (Sobolev space). For an integer $k \geq 0$ and $1 \leq p \leq \infty$, we define the Sobolev space $W^{k,p}(\mathcal{Z}) = \{f \in L^p(\mathcal{Z}) \mid D^\alpha f \in L^p(\mathcal{Z}) \text{ for all multi-indices } \alpha \text{ with } |\alpha| \leq k\}$.

We write $W^{k,p}$ whenever the set $\mathcal{Z}$ is clear from the context. Note that since $L^p \subset L^1_{\text{loc}}$ for $p \geq 1$, Sobolev spaces impose more restrictions than weak differentiability. Also, while $W^{k,p}$ is in general a Banach space, $W^{k,2}$ is a Hilbert space (Evans, 2010).

Having set the groundwork, we can now proceed with the proof of Prop. 3.1.

## B.1  PROOF OF PROP. 3.1

The proof follows by constructing a measure of the deviation from the weak formulation (1) and showing that it is dominated by the proposed worst-case statistical loss. This immediately implies Prop. (3.1). Explicitly, note that the weak formulation (1) can equivalently be expressed as

$$\left[\int_{\mathcal{D}} \Big(D[u](x,t) - \tau(x,t)\Big)\varphi(x,t)dxdt\right]^2 = 0, \quad \text{for all } \varphi \in W^{k',2} \tag{17}$$

where we omitted the dependence on the coefficients $\pi$ for conciseness. Using Jensen's inequality, we can upper bound (17) for any $\varphi$ as in

$$\left[\int_{\mathcal{D}} \Big(D[u](x,t) - \tau(x,t)\Big)\varphi(x,t)dxdt\right]^2 \leq \int_{\mathcal{D}} \Big(D[u](x,t) - \tau(x,t)\Big)^2\varphi^2(x,t)dxdt. \tag{18}$$

Since $\varphi \in W^{k',2} \subset L^2$, the partition function $Z_\psi = \int_{\mathcal{D}} \varphi(x,t)^2 dxdt = \|\varphi\|^2_{L^2} < \infty$ is well-defined. We can therefore consider the normalized $\psi = \varphi^2/Z_\psi$ in (18) to get

$$\left[\int_{\mathcal{D}} \Big(D[u](x,t) - \tau(x,t)\Big)\varphi(x,t)dxdt\right]^2 \leq Z_\psi \int_{\mathcal{D}} \Big(D[u](x,t) - \tau(x,t)\Big)^2\psi(x,t)dxdt. \tag{19}$$

Since $\psi$ is a non-negative, normalized function, it is the density of a probability measure, i.e., $\psi \in \mathcal{P}$. The following technical lemma shows that it is in fact in $\mathcal{P}^2$, i.e., it is a square-integrable probability density.

**Lemma B.2.** *Let $\varphi \in W^{k',2}$ and $\psi \propto \varphi^2$ be a probability distribution. Then, $\psi \in \mathcal{P}^2$.*

Before proving Lemma B.2, let us conclude the proof. The hypothesis on $u^\dagger$ implies that

$$\sup_{\psi \in \mathcal{P}^2} \int_{\mathcal{D}} \left( D[u^\dagger](x,t) - \tau(x,t) \right)^2 \psi(x,t) dx dt = 0$$

and since $Z_\psi$ is bounded, we obtain from (19) that

$$\left[ \int_{\mathcal{D}} \left( D[u^\dagger](x,t) - \tau(x,t) \right) \varphi(x,t) dx dt \right]^2 \le 0.$$

Noticing that this holds for all $\psi \in W^{k',2}$, we recover (17), which concludes the proof. ∎

*Proof of Lemma B.2.* To show $\psi \in \mathcal{P}^2$, we must show that $\phi \in L^4$. Indeed,

$$\int_{\mathcal{D}} \psi(x,t)^2 dx dt = \frac{1}{Z^2} \int_{\mathcal{D}} \varphi(x,t)^4 dx dt = \left( \frac{\|\varphi\|_{L^4}}{\|\varphi\|_{L^2}} \right)^4. \tag{20}$$

Since $\varphi \in W^{k',2} \subset L^2$, suffices it to show that the numerator is finite. To do so, we can use the Gagliardo-Nirenberg interpolation inequality (Nirenberg, 1959; Fiorenza et al., 2021) to write

$$\|\varphi\|_{L^4} \le C \left( \|D^m \varphi\|_{L^2}^\alpha \|\varphi\|_{L^2}^{1-\alpha} + \|\varphi\|_{L^2} \right), \quad \text{for } \alpha \in [0,1], \tag{21}$$

as long as (i) $k' \ge m \in \mathbb{N}$ and (ii) $4\alpha m = d + 1$. An additional condition must hold in particular cases:

$$\text{(iii) if } m - \frac{d+1}{2} \text{ is a non-negative integer, then } \alpha < 1.$$

Note that $d + 1$ is the dimension of the space-time domain $\mathcal{D}$. Since $\varphi \in W^{k',2}$, the right-hand of (21) is finite. We therefore only need to show that (i)–(iii) are satisfied under the hypothesis of the theorem. To do so, we consider three cases:

(a) $\boldsymbol{d = 0}$: in this case, $d + 1$ is odd so that condition (iii) does not apply. Immediately, (21) holds for $m = 1$ and $\alpha = 0.25$.

(b) $\boldsymbol{d = 1}$: since (ii) requires $m \ge 1$, we now have that $m - \frac{d+1}{2}$ is a non-negative integer. Hence, condition (iii) applies. Nevertheless, we can still take $m = 1$ and $\alpha = 0.5$ in (21).

(c) $\boldsymbol{d \ge 2}$: we can now consider all other cases by taking

$$m = \left\lceil \frac{d+1}{4} \right\rceil \quad \text{and} \quad \alpha = \frac{d+1}{4m}.$$

Indeed, (ii) holds by construction. Additionally, from the hypothesis of the theorem, we have $d + 1 \le 4k'$, which by the monotonicity of the ceiling operation implies that (i) holds. It suffices to show that (iii) never applies. For $2 \le d \le 3$, we have $m = 1$ and $(d+1)/2 > 1$. For $d > 3$, we have

$$\left\lceil \frac{d+1}{4} \right\rceil - \frac{d+1}{2} \le \left( \frac{d+1}{4} + 1 \right) - \frac{d+1}{2} < 0. \tag{22}$$

Thus, under the hypothesis of the theorem, (21) implies that $\|\varphi\|_{L^4} < \infty$. From (20), this in turn implies that $\psi \in \mathcal{P}^2$. ∎

## B.2 PROP. 4.1

For the sake of completeness, we provide a short discussion of the preliminary material needed to prove this proposition.

### B.2.1 PRELIMINARIES

We begin by showing that the supremum can be written as a distributionally robust optimization problem. Indeed, consider the worst-case loss $\bar{\ell}(\theta) = \sup_{(x,t) \in \mathcal{D}} \ell(u_\theta(x,t))$ and assume that $(x,t) \mapsto \ell(u_\theta(x,t))$ is a function in $L^2$. This loss can be written in epigraph form as

$$\bar{\ell}(\theta) = \inf_{t \in \mathbb{R}} \ t \text{ subject to } \ell(u_\theta(x,t)) \leq t, \quad \text{for all } (x,t) \in \mathcal{D}. \tag{PV}$$

Writing (PV) in Lagragian form (Bertsekas, 2009, Ch. 4), we obtain

$$\bar{\ell}(\theta) = \inf_{t \in \mathbb{R}} \sup_{\psi \in L_+^2} \ L_{\text{PV}}(\theta, t, \psi), \tag{PVI}$$

where $L_+^2$ denotes the subspace of almost everywhere non-negative functions of $L^2$. Here, the Lagrangian $L_{\text{PV}}(\theta, t, \psi)$ is defined as

$$\begin{aligned}
L_{\text{PV}}(\theta, t, \psi) &= t + \int_{\mathcal{D}} \psi(x,t)\big[\ell(u_\theta(x,t)) - t\big]dxdt \\
&= t\bigg[1 - \int_{\mathcal{D}} \psi(x,t)dxdt\bigg] + \int_{\mathcal{D}} \psi(x,t)\ell(u_\theta(x,t))dxdt,
\end{aligned} \tag{23}$$

Since (23) is a linear function of $t$, $\bar{\ell}(\theta)$ is the optimal value of a linear program parametrized by $\theta$. Hence, strong duality holds (Bertsekas, 2009, Ch. 4) and we obtain that

$$\bar{\ell}(\theta) = \sup_{\psi \in L_+^2} \ d_{\text{PV}}(\psi) \quad \text{where} \quad d_{\text{PV}}(\psi) \triangleq \min_{t \in \mathbb{R}} \ L_{\text{PV}}(\theta, t, \psi). \tag{24}$$

Since $t$ is unconstrained and $L_{\text{PV}}$ is linear in $t$, the dual function diverges to $-\infty$ unless $\int_{\mathcal{D}} \psi(x,t)dxdt = 1$. From (23) and (24), we thus obtain that

$$\bar{\ell}(\theta) = \sup_{\psi \in \mathcal{P}^2} \int_{\mathcal{D}} \psi(x,t)\ell(u_\theta(x,t))dxdt \tag{25}$$

We also quickly review the necessary variational results for normal integrands. The majority of this exposition is adapted from Rockafellar & Wets (2004). Throughout, we let the tuple $(T, \mathcal{A})$ denote a measurable space, where $T$ is a nonempty set and $\mathcal{A}$ is a $\sigma$-algebra of measurable sets belonging to $T$.

**Definition B.3** (Carathéodory integrand). A function $f : T \times \mathbb{R}^n \to \mathbb{R}$ is called a **Carathéodory integrand** if it is measurable in $t$ for each $x$ and continuous in $x$ for each $t$.

**Definition B.4** (Decomposable space). A space $\mathcal{F}$ of measurable functions $g : T \to \mathbb{R}^n$ is **decomposable** in association with a measure $\mu$ on $\mathcal{A}$ if for every function $g_0 \in \mathcal{F}$, for every set $A \in \mathcal{A}$ with $\mu(A) < \infty$, and for every bounded, measurable function $g_1 : A \to \mathbb{R}^n$, the space $\mathcal{F}$ contains the function $g : T \to \mathbb{R}^n$ defined by

$$g(t) = \begin{cases} g_0(t) & \text{for } t \in T \backslash A, \\ g_1(t) & \text{for } t \in A. \end{cases} \tag{26}$$

Note that Lebesgue spaces $L^p$ are decomposable for all $p \in [1, \infty]$ (see, e.g., (Rockafellar & Wets, 2004, Ch. 14)). We can now state a crucial result concerning the interchangability of maximization and integration.

**Theorem B.5** (Thm. 14.60 in Rockafellar & Wets (2004)). *Let $\mathcal{F}$ be a decomposable space of measurable functions and $F : T \times \mathbb{R}^n$ be a Carathéodory integrand. Then, as long as $\int_T f(\tau, \phi(\tau))d\tau \neq 0$ for all $\phi \in \mathcal{F}$,*

$$\inf_{\phi \in \mathcal{F}} \int_T f(\tau, \phi(\tau))d\tau = \int_T \left[\inf_{x \in \mathbb{R}^n} f(\tau, x)\right]d\tau. \tag{27}$$

*Moreover, as long as this common value is not $-\infty$, one has that*

$$\bar{\phi} \in \operatorname*{argmin}_{\phi \in \mathcal{F}} \int_T f(\tau, \phi(\tau))d\tau \iff \bar{\phi}(\tau) \in \operatorname*{argmin}_{x \in \mathbb{R}^n} f(\tau, x) \quad \text{for almost every } \tau \in T. \tag{28}$$

## B.3 PROOF OF PROPOSITION 3.2

*Proof.* Consider a sequence $\psi_n^\star \in \mathcal{P}^2$ converging to $\bar{\ell}(\theta)$ in $L^2$, i.e., a solution of (25). In other words, for every $\delta > 0$, there exists $N_\delta < \infty$ such that

$$\bar{\ell}_\delta(\theta) \triangleq \int_{\mathcal{D}} \psi_n^\star(x,t)\ell(u_\theta(x,t))dxdt \geq \bar{\ell}(\theta) - \delta, \quad \text{for all } n \geq N_\delta. \tag{29}$$

Consider now $c_\delta = \min_{n \leq N_\delta} \|\psi_n^\star\|_{L^2}^2$. Since $\psi_n^\star \in \mathcal{P}^2 \subset L^2$, $\|\psi_n^\star\|_{L^2}^2$ is finite for all $n$. And since $N_\delta$ is finite, so is $c_\delta$. We can therefore rewrite (29) as

$$\bar{\ell}_\delta(\theta) = \sup_{\psi \in L_+^2} \quad \int_{\mathcal{D}} \psi(x,t)\ell(u_\theta(x,t))dxdt$$

$$\text{subject to} \quad \int_{\mathcal{D}} \psi(x,t)dxdt = 1, \quad \|\psi\|_{L^2}^2 \leq c_\delta, \tag{PVII}$$

where we rewrote $\mathcal{P}^2$ as $\{\psi \in L_+^2 \mid \int_{\mathcal{D}} \psi(x,t)dxdt = 1\}$. Notice that (PVII) is a convex quadratic program in $\psi$. Furthermore, note that a zero-mean normal distribution with variance $c^2/2$ is strictly feasible for (PVII) (it belongs to $\mathcal{P}^2$ and strictly satisfies the $L^2$-norm constraint). Hence, Slater's condition holds and we find that (PVII) is strongly dual (Bertsekas, 2009, Ch. 4). We therefore conclude that for every $c_\delta > 0$, there exists $\mu_\delta \in \mathbb{R}$ and $0 \leq \gamma_\delta < \infty$ such that

$$\bar{\ell}_\delta(\theta) = \sup_{\psi \in L_+^2} \int_{\mathcal{D}} \psi(x,t)\ell(u_\theta(x,t))dxdt + \alpha_\delta\left[\int_{\mathcal{D}} \psi(x,t)dxdt - 1\right] + \gamma_\delta\left[\|\psi\|_{L^2}^2 - c_\delta\right]$$

$$= \sup_{\psi \in L_+^2} \int_{\mathcal{D}} \left[\psi(x,t)\ell(u_\theta(x,t)) + \gamma_\delta\psi(x,t)^2 + \alpha_\delta\psi(x,t)\right]dxdt - \gamma_\delta c_\delta - \alpha_\delta.$$

To conclude, we use the fact that $L_+^2$ is decomposable and since that the integrand is Carathéodory to exchange the supremum and the integral to obtain that

$$\bar{\ell}_\delta(\theta) = \int_{\mathcal{D}} \left[\sup_{\psi \in \mathbb{R}} \psi\ell(u_\theta(x,t)) + \gamma_\delta\psi^2 + \alpha_\delta\psi\right]dxdt - \gamma_\delta c_\delta - \alpha_\delta$$

A straightforward calculation of the inner maximization problem shown above yields that the solution to (PVII) is given by

$$\psi_\alpha(x,t) = \frac{\left[\ell(u_\theta(x,t)) - \alpha\right]_+}{2\gamma}, \tag{30}$$

where $[z]_+ = \max(0,z)$ denotes the projection onto the non-negative orthant and $\alpha, \gamma$ are chosen so that

$$\int_{\mathcal{D}} \psi_\alpha^\star(x,t)dxdt = 1 \quad \text{and} \quad \|\psi_\alpha\|_{L^2}^2 \leq c_\delta.$$

Hence, for any $\delta > 0$ in (29), we can find $\alpha$ such that (30) approximates $\bar{\ell}$. ∎

## C    Sampling with the Metropolis-Hastings algorithm

Algorithm 1 uses samples from one or more $\psi_0$ in order to compute the losses in steps 4–7. This is, however, not straightforward unless those distributions are fixed to, e.g., a uniform (as we typically do for the BCs). That is because we only know $\psi_0$ up to a normalization factor. We overcome this issue using MCMC techniques, more specifically, the Metropolis-Hastings algorithm (Robert & Casella, 2004). In Alg. 2, we consider the general case of sampling from $\psi_0 \propto \ell$, where $\ell$ is a non-negative, scalar-valued loss. We denote by $z_n$ the desired samples and $\mathcal{R}$ their support. In Alg. 1, for instance, we would take $z_n = (x_n^{\text{pde}}, t_n^{\text{pde}}, \pi_n^{\text{pde}})$, $\mathcal{R} = \mathcal{D} \times \Pi$, and

$$\ell(z_n) = \left( D_{\pi_n^{\text{pde}}} \left[ u_{\theta_k}(\pi_n^{\text{pde}}) \right] (x_n^{\text{pde}}, t_n^{\text{pde}}) - \tau(x_n^{\text{pde}}, t_n^{\text{pde}}) \right)^2 \tag{31}$$

in step 5 and $z_n = (x_n^{\text{o}_j}, t_n^{\text{o}_j})$, $\mathcal{R} = \mathcal{D}$, and

$$\ell(z_n) = \left( u_{\theta_k}(\pi_j, \tau_j)(x_n^{\text{o}_j}, t_n^{\text{o}_j}) - u_j^{\dagger}(x_n^{\text{o}_j}, t_n^{\text{o}_j}) \right)^2$$

in step 7.

Typically, the covariance $\Sigma$ is taken to be diagonal (independent proposals), e.g., $\sigma^2 I$. The choice of parameter $\sigma^2$ of the proposal (step 3) affects the *mixing rate*, i.e., how fast the samples converge to the desired distribution. Smaller values of $\sigma^2$ will lead to slower mixing chains since the algorithm will not explore the space efficiently. On the other hand, large values will cause the acceptance probability in step 4 to be too small, so that the algorithm will remain stuck (step 5). Oftentimes, the parameter $\sigma^2$ is adapted during a burn-in phase to hit a specific acceptance rate, around $30\%$ as a rule-of-thumb (Robert & Casella, 2004). In our experiments, we find a reasonable value for $\sigma^2$ and keep it fixed throughout training. We also consider a burn-in period by using only the last $N_0$ samples generated by Alg. 2. For the single BVP experiments in Sec. 5.1 we use $N_0 = 1000$ and for the parameterized BVPs in Sec. 5.2 we use $N_0 = 2500$.

Given that we sample only from bounded domains (i.e., some subset of $\mathcal{D} \times \Pi$), the target distribution $\psi_\alpha$ has finite tails for any $\alpha$, satisfying sufficient conditions for uniform ergodicity [see, e.g., (Jarner & Hansen, 2000)]. The law of the samples obtained from Alg. 2 therefore converge (in Kullback-Leibler divergence) to $\psi_\alpha$. Additionally, since we prove in Prop. 3.1 and 4.1 that $\psi_0$ (and more generally, $\psi_\alpha$) are square-integrable, alternative sampling technique with faster mixing rates can be used. That is the case, for instance, of Langevin Monte Carlo (LMC) (Robert & Casella, 2004). Yet, the LMC algorithm uses first-order information of $\ell$. For the PDE loss in (31), this means higher-order space-time derivatives of $u_\theta$ and thus, additional backward passes. It is not clear that the benefits of faster mixing outweigh the increase in computational complexity, especially given that good results can be obtained using Alg. 2.

---

**Algorithm 2** Metropolis-Hastings algorithm with Gaussian proposal

---

1: $z_0 \sim \text{Uniform}(\mathcal{R})$          ▷ Sample initial state
2: **for** $n = 0, \ldots, N-1$
3:     $\hat{z} \sim \text{Gaussian}(z_n, \Sigma)$          ▷ Draw proposal
4:     $p_n = \min\left(1, \dfrac{\ell(\hat{z})}{\ell(z_n)}\right) \mathbb{I}(\hat{z} \in \mathcal{R})$          ▷ Evaluate acceptance probability
5:     $\begin{cases} z_{n+1} = \hat{z}, & \text{with probability } p_n \\ z_{n+1} = z_n, & \text{with probability } 1 - p_n \end{cases}$          ▷ Update state
6: **end**
7: **return** $\{z_1, \ldots, z_N\}$

---

# D    GENERALIZATION RESULTS

Here, we formalize the generalization guarantees for when solutions of the empirical dual problem are (probably approximately) near-optimal and near-feasible for the statistical primal problem. These were first detailed in Chamon & Ribeiro (2020); Chamon et al. (2023). As done in Sec. 4, for simplicity and clarity, we consider (SCL)(M). Specifically, we are interested when solutions of (DIV) are (probably approximately) near-optimal and near-feasible for (PIV). Generalization guarantees for the for the complete (SCL) as well as its parametric extension (SCL$'$) follow in the same way.

We begin with the essential (non-convex) duality assumptions. In particular, we assume that the hypothesis space $\mathcal{F}_\theta = \{u_\theta : \theta \in \Theta\}$ is sufficiently expressive (Assumption D.1) and that there exists a function $u_\theta \in \mathcal{F}_\theta$ that is strictly feasible (Assumption D.2). For NNs in particular, universal approximation theorems indicate that these assumptions are satisfied for large enough models (see, e.g., (Hornik, 1991)). Finally, we impose a learning theoretic limit on the complexity of the hypothesis space $\mathcal{F}_\theta$ in order to ensure that our empirical approximations are well-posed (Assumption D.3). The main theorem our results are based on, namely (Chamon et al., 2023, Thm. 1), also require the losses to be convex, $M$-Lipschitz continuous, and $[0, B]$ bounded. Since we only consider quadratic losses on the bounded domain $\mathcal{D}$, these assumptions hold immediately.

**Assumption D.1.** The parametrization $u_\theta$ is rich enough that for each $\theta_1, \theta_2 \in \Theta$ and $\beta \in [0, 1]$, there exists $\theta \in \Theta$ such that $\sup_{(x,t)\in\mathcal{D}} |\beta u_{\theta_1}(x, t) + (1 - \beta)u_{\theta_2}(x, t) - u_\theta(x, t)| \le \nu$.

**Assumption D.2.** There exist $\theta'$ such that $u_{\theta'}$ is strictly feasible for PIV, i.e., such that

$$\mathbb{E}_{(x,t)\sim\psi_\alpha^{\text{pde}}}\left[\left(D[u_\theta](x, t) - \tau(x, t)\right)^2\right] \le \epsilon - M\nu.$$

**Assumption D.3.** There exist $\zeta(N, \delta)$ monotonically decreasing with $N$ such that with probability $1 - \delta$ over samples $(x_n^{\text{bc}}, t_n^{\text{bc}}) \sim \psi_\alpha^{\text{bc}}$ and $(x_n^{\text{pde}}, t_n^{\text{pde}}) \sim \psi_\alpha^{\text{pde}}$, it holds for all $\theta \in \Theta$ that

$$\left| \mathbb{E}_{(x,t)\sim\psi_\alpha^{\text{bc}}}\left[\left(u_\theta(x, t) - h(x, t)\right)^2\right] - \frac{1}{N}\sum_{n=1}^N \left(u_\theta(x_n^{\text{bc}}, t_n^{\text{bc}}) - h(x_n^{\text{bc}}, t_n^{\text{bc}})\right)^2 \right| \le \zeta(N, \delta)$$

$$\left| \mathbb{E}_{(x,t)\sim\psi_\alpha^{\text{pde}}}\left[\left(D[u_\theta](x, t) - \tau(x, t)\right)^2\right] - \frac{1}{N}\sum_{n=1}^N \left(D[u_\theta](x_n^{\text{pde}}, t_n^{\text{pde}}) - \tau(x_n^{\text{pde}}, t_n^{\text{pde}})\right)^2 \right| \le \zeta(N, \delta).$$

Under these assumptions, we can bound the empirical duality gap between (PIV) and (DIV), i.e., $\Delta = |P^\star - D^\star|$, where

$$P^\star = \underset{\theta\in\Theta}{\text{minimize}} \quad \mathbb{E}_{(x,t)\sim\psi_\alpha^{\text{bc}}}\left[\left(u_\theta(x, t) - h(x, t)\right)^2\right]$$
$$\text{subject to} \quad \mathbb{E}_{(x,t)\sim\psi_\alpha^{\text{pde}}}\left[\left(D[u_\theta](x, t) - \tau(x, t)\right)^2\right] \le \epsilon$$

and

$$D^\star = \max_{\lambda\ge 0} \min_{\theta\in\Theta} \hat{L}(\theta, \lambda).$$

**Proposition D.4.** *Under Assumptions D.1–D.3, it holds with probability $1 - (3m + 2)\delta$ that*

$$\Delta \le O\left(\lambda^\star(M\nu + \zeta)\right),$$

*where $\lambda^\star$ is a solution of* (DIV).

Prop. D.4 is obtained directly from (Chamon et al., 2023, Thm. 1). This duality gap bound is enough to guarantee that the dual ascent algorithm in (5) provides a near-optimal and near-feasible randomized solution of (PIV). Since all our losses are strongly convex (quadratic), we can further show that randomization is not necessary using the last iterate guarantees from (Elenter et al., 2024, Prop. 4.1). This is in spite of the fact that (PIV) is a non-convex optimization problem.

On the other hand, the convergence of primal-dual methods such as Alg. 1 in non-convex settings is the subject of active research, see, e.g., (Yang et al., 2020; Lin et al., 2020; Fiez et al., 2021; Boroun et al., 2023). Transferring the guarantees from (5) to Alg. 1 requires additional conditions, e.g., step size separation as in (Yang et al., 2020). Such convergence guarantees are, however, beyond the scope of this paper and left for future work.

# E  EXPERIMENTAL DETAILS

## E.1  HYPERPARAMETERS AND IMPLEMENTATION DETAILS

Throughout our experiments, we use the relative $L_2$ error as a performance metric, which we define as

$$e_{\text{rel}}(\pi, h) = \sqrt{\frac{\sum_{n=1}^{N} \left[ u_\theta(\pi, h)(x_n, t_n) - u^\dagger(\pi, h)(x_n, t_n) \right]^2}{\sum_{n=1}^{N} \left[ u^\dagger(\pi, h)(x_n, t_n) \right]^2}}, \tag{32}$$

where $u^\dagger$ is the solution of (BVP) obtained either analytically or by using classical numerical methods. For MLPs, the collocation points $\{(x_n, t_n)\}$ are taken from a dense regular grid of points (see exact numbers below), and for FNOs, they are determined by the test sets from (Li et al., 2021; Takamoto et al., 2022). For parametrized problems, we report the average error

$$\bar{e}_{\text{rel}} = \frac{1}{J} \sum_{j=1}^{J} e_{\text{rel}}(\pi_j, h_j),$$

evaluated either on a dense regular grid of points (for coefficients $\pi$, see exact numbers below) or based on the test sets from (Li et al., 2021; Takamoto et al., 2022).

To provide sensitivity measures, we run all experiments for 10 different seeds and report average and standard deviations of the results. We find that for certain difficult problem (e.g., diffusion with $\beta = 50$ or reaction-diffusion with $(\nu, \rho) = (3, 5)$) the hyperparameters of SCL and R3 sometimes need to be adjusted for certain seeds. This occurs rarely, but shows that there may not be one-size-fits-all hyperparameter settings. For PINNs in (PI), we were unable to find any hyperparameters that solved those problems.

### E.1.1  SOLVING A SPECIFIC BVP (SEC. 5.1)

In this section, we formulated SCL problems of the form (SCL)(M) in order to use the same information that PINNs traditionally rely on. Recall that we do use the worst-case distribution $\psi_0$ (or even random points) for the BCs, but instead consider fixed, regularly distributed points. This reduces the overall computational complexity of the problem at essentially no performance cost. Explicitly, we consider the following problem

$$\underset{\theta \in \Theta}{\text{minimize}} \quad \frac{1}{N} \sum_{n=1}^{N} \left( u_\theta(x_n^{\text{bc}}, t_n^{\text{bc}}) - h(x_n^{\text{bc}}, t_n^{\text{bc}}) \right)^2$$

$$\text{subject to} \quad \mathbb{E}_{(x,t) \sim \psi_0^{\text{PDE}}} \left[ \left( D[u_\theta](x, t) - \tau(x, t) \right)^2 \right] \leq \epsilon_{\text{pde}}.$$

To compute the objective for the convection and reaction-diffusion PDEs, we use 256 points $(x^{\text{bc}}, 0)$, $x^{\text{bc}} \in [0, 2\pi]$, for the IC and 100 points equally spaced in $t \in (0, 1]$ to evaluate the period BC. For the eikonal PDE, recall from (10) we use an additional structural constraint. In this case, we therefore formulate the SCL problem

$$\underset{\theta \in \Theta}{\text{minimize}} \quad \frac{1}{M} \sum_{m=1}^{M} \left[ u_\theta(x_m, y_m) \right]^2$$

$$\text{subject to} \quad \mathbb{E}_{(x,t) \sim \psi_0^{\text{PDE}}} \left[ \left( D[u_\theta](x, t) - \tau(x, t) \right)^2 \right] \leq \epsilon_{\text{pde}}$$

$$\frac{1}{N} \sum_{n=1}^{N} \left[ -u_\theta(x_n, y_n) \right]_+ \leq \epsilon_{\text{s}},$$

where we use fixed collocation points for the BCs and structural constraint, namely, $M = 2234$ points on $\partial S$ (the gears figure from (Daw et al., 2023)) and $N = 40$ points on $\partial \Omega$. We use the exact same points for (PI).

**Problem hyperparameters.** For SCL, the tolerance $\epsilon_{\text{pde}}$ was selected by starting with a small value (e.g., $10^{-4}$) and increasing it when the dual variables became too large during training to accommodate difficult problems. After a coarse hyperparameter search, we kept the weights $\mu$ in (PI) used in (Daw et al., 2023). Note that we used different weights for the BC and IC to solve the eikonal PDE, since in this case the BCs play a less critical role. All values are displayed in Table 3. When solving the Eikonal equation with SCL, we used $\epsilon_{\text{s}} = 10^{-3}$ as the tolerance for the structural constraint.

**Model.** We used MLPs with 4 hidden layers for $u_\theta$ each with 50 neurons for the convection and reaction-diffusion equations or 128 neurons for the eikonal equation and hyperbolic tangent activation function.

**Training.** To evaluate the PDE loss, all methods used 1000 collocation points sampled uniformly at random at the beginning of each epoch (PINN), obtained using the R3 from (Daw et al., 2023) (R3), or using Alg. 2 (SCL). For R3, we use the hyperparameters from (Daw et al., 2023). For Alg. 2, we use $\Sigma = \text{diag}(0.25, 0.01)$ for drawing proposals for $x$ and $t$ respectively for both convection and reaction-diffusion. For the eikonal PDE, we use $\Sigma = 0.04 \times I$. In both cases, we run the algorithm for $N = 5000$ and use only the last 1000 samples. All methods were trained using Adam with the default parameters from (Kingma & Ba, 2017) and learning rates described in Table 4. Note that the baselines only use learning rate $\eta_p$, since they do not use dual methods.

**Testing.** The solution of the convection and reaction-diffusion PDEs were tested on a dense regular grid of $256 \times 100$ points $(x, t) \in \mathcal{D}$ against their analytical solutions. The solution of the eikonal PDE was tested on a dense regular grid of $384 \times 384$ points $(x, y) \in \Omega$ against the ground truth predictions from (Daw et al., 2023).

### E.1.2 SOLVING PARAMETRIC FAMILIES OF BVPS (SEC. 5.2)

The SCL problem we formulate here is similar to the previous section, although we used the parameterized version (SCL′)(M). Once again, we replace the $(x, t)$ marginals of the worst-case distribution $\psi_0^{\text{BC}}$ by a fixed, uniform distribution. Note, however, that we keep the worst-case formulation for the coefficients $\pi$. Explicitly, we consider the SCL problem

$$\underset{\theta \in \Theta}{\text{minimize}} \quad \mathbb{E}_{\pi \sim \psi_0^{\text{BC}}} \left[ \frac{1}{N_{\text{bc}}} \sum_{n=1}^{N_{\text{bc}}} \left( u_\theta(\pi)(x_n^{\text{bc}}, t_n^{\text{bc}}) - h(\pi)(x_n^{\text{bc}}, t_n^{\text{bc}}) \right)^2 \right]$$

$$\text{subject to} \quad \mathbb{E}_{(x,t,\pi) \sim \psi_0^{\text{PDE}}} \left[ \left( D_\pi[u_\theta(\pi)](x, t) - \tau(\pi)(x, t) \right)^2 \right] \leq \epsilon_{\text{pde}}$$

Once again, we compute the objective for the convection and reaction-diffusion PDEs using 256 points $(x, 0)$, $x \in [0, 2\pi]$, for the IC and 100 points equally spaced in $t \in (0, 1]$ to evaluate the period BC. For the Helmholtz PDE, we use $4 \times 256$ points equally space around $\partial\Omega$ to evaluate the BC. We use the exact same points for (PI). Note that we include the coefficients $\pi$ in the forcing function to account for the Helmholtz BVP (see Sec. A).

**Problem hyperparameters.** Once again, the tolerance $\epsilon_{\text{pde}}$ were selected by starting with a small value (e.g., $10^{-4}$) and increasing when the dual variables achieved too large a value during training to accommodate difficult problems. The weights $\mu$ in (PI) for the baselines were taken from (Daw et al., 2023). Exact values are displayed in Table 5.

Table 3: Problem hyperparameters for solving a specific BVP

|  | $\mu_{\text{D}}$ | $\mu_{\text{BC}}$ | $\mu_{\text{IC}}$ | $\epsilon_{\text{pde}}$ |
|---|---|---|---|---|
| Convection: $\beta = 30$ | 1 | 100 | 100 | $10^{-3}$ |
| Convection: $\beta = 50$ | 1 | 100 | 100 | $5 \times 10^{-3}$ |
| Reaction-diffusion: $(\nu, \rho) = (3, 3)$ | 1 | 100 | 100 | $10^{-2}$ |
| Reaction-diffusion: $(\nu, \rho) = (3, 5)$ | 1 | 100 | 100 | $5 \times 10^{-3}$ |
| Eikonal | 1 | 10 | 500 | $5 \times 10^{-1}$ |

Table 4: Training hyperparameters for solving a specific BVP

|  | $\eta_p$ | $\eta_d$ (only SCL) | Learning rate decay | Iterations |
|---|---|---|---|---|
| Convection: $\beta = 30$ | $10^{-3}$ | $10^{-4}$ | $0.9\eta$ every $5\,000$ iter. | $175\,000$ |
| Convection: $\beta = 50$ | $10^{-3}$ | $10^{-4}$ | $0.9\eta$ every $5\,000$ iter. | $200\,000$ |
| Reaction-diffusion: $(\nu, \rho) = (3, 3)$ | $10^{-3}$ | $10^{-4}$ | — | $200\,000$ |
| Reaction-diffusion: $(\nu, \rho) = (3, 5)$ | $10^{-3}$ | $10^{-4}$ | — | $200\,000$ |
| Eikonal | $10^{-3}$ | $10^{-4}$ | $0.9\eta$ every $5\,000$ iter. | $60\,000$ |

**Model.** We used MLPs with 4 hidden layers of 50 neurons.

**Training.** When training using (PI), we used 1000 collocation points per coefficient value, sampled uniformly at random at the beginning of each epoch. For (SCL$'$)(M), we used Alg. 2. For the PDE loss, we used $\Sigma = \mathrm{diag}(0.25, 0.01, \sigma_\pi^2)$ to sample from $(x, t, \pi)$ for both the convection ($\sigma_p i^2 = 9$ for coefficient $\beta$) and reaction-diffusion [$\sigma_p i^2 = (1, 1)$ for coefficients $(\nu, \rho)$] equations. For the Helmholtz PDE, we used $\Sigma = 0.04 \times I$ to sample from $(x, t, \pi)$ for coefficients $\pi = (a_1, a_2)$. The same variances $\sigma_\pi^2$ were used to sample worst-case coefficients for the BC (recall that the distribution over collocation points is fixed). In all cases, SCL uses the last 2500 out of 5000 samples generated by the MH algorithm, except for the Helmholtz PDE with $(a_1, a_2) \in [1, 3]^2$, where we use all $5,000$ samples to account for the additional difficulty of the problem.

All models were trained for $200,000$ using Adam with the default parameters from (Kingma & Ba, 2017) and learning rate of $10^{-3}$ for (PI). For SCL, the dual learning rate was $\eta_d = 10^{-4}$. In all cases, we decayed the learning rates by a factor of 0.9 every 5000 epochs, with the exception of the reaction-diffusion PDE where we found it better to keep the learning rate constant.

**Testing.** The solution of the convection and reaction-diffusion PDEs were tested on a dense regular grid of $256 \times 100 \times 1000$ points $(x, t, \pi) \in \mathcal{D} \times \Pi$ and $256 \times 100 \times 100 \times 100$ points $(x, t, \nu, \rho) \in \mathcal{D} \times \Pi$, respectively, against their analytical solutions. The solution of the Helmholtz PDE was tested on a dense regular grid of $256 \times 256 \times 100 \times 100$ points $(x, y, a_1, a_2) \in \Omega \times \Pi$ against its analytical solution.

### E.1.3 LEVERAGING INVARIANCE WHEN SOLVING BVPS (SEC. 5.3)

The SCL problems formulated in this section are of the form (SCL)(M+I). To showcase the advantages of integrating additional knowledge, such as the structure of the BVP solution, we consider fixed collocation points for the constraints (SCL)(M). This is in fact not uncommon for PINNs, see, e.g., (Raissi et al., 2019; Lu et al., 2021b). These points are sampled uniformly at random once and then kept constant throughout training. Recall that for our convection BVP (Sec. A), the solution is

Table 5: Problem hyperparameters for solving a parametric family of BVPs

|  | $\mu_D$ | $\mu_{BC}$ | $\epsilon_{pde}$ |
|---|---|---|---|
| Convection | 1 | 100 | $10^{-3}$ |
| Reaction-diffusion: $(\nu, \rho) \in [0, 5]^2$ | 1 | 100 | $5 \times 10^{-3}$ |
| Reaction-diffusion: $(\nu, \rho) \in [0, 10]^2$ | 1 | 100 | $10^{-2}$ |
| Reaction-diffusion: $(\nu, \rho) \in [0, 20]^2$ | 1 | 100 | $10^{-1}$ |
| Helmholtz: $(a_1, a_2) \in [1, 2]^2$ | 1 | 100 | $5 \times 10^{-1}$ |
| Helmholtz: $(a_1, a_2) \in [1, 3]^2$ | 1 | 100 | 5 |

Table 6: Problem hyperparameters for supervised solutions

| | $\epsilon_\text{o}$ | # training samples | # validation samples | # test samples | FNO architecture |
|---|---|---|---|---|---|
| Burgers' | $10^{-3}$ | 800 | 200 | 200 | 16 modes, 4 layers |
| Diffusion-sorption | $10^{-3}$ | 1000 | 500 | 500 | 8 modes, 5 layers |
| Navier-Stokes: $\nu = 10^{-3}$ | $10^{-2}$ | 1000 | 500 | 500 | 8 modes, 8 layers |
| Navier-Stokes: $\nu = 10^{-4}$ | $5 \times 10^{-2}$ | 1000 | 500 | 500 | 8 modes, 8 layers |
| Navier-Stokes: $\nu = 10^{-5}$ | $10^{-2}$ | 800 | 200 | 200 | 8 modes, 8 layers |

periodic with period $2\pi/\beta$. We therefore use the problem

$$\underset{\theta \in \Theta}{\text{minimize}} \quad \frac{1}{N} \sum_{n=1}^{N} \Big( u_\theta(x_n, t_n) - h(x_n, t_n) \Big)^2$$

$$\text{subject to} \quad \frac{1}{M} \sum_{m=1}^{M} \Big( D[u_\theta](x_m, t_m) - \tau(x_m, t_m) \Big)^2 \leq \epsilon_\text{pde}$$

$$\mathbb{E}_{(x,t) \sim \psi_0^\text{ST}} \left[ \Big( u_\theta(x,t) - u_\theta\Big[x, t + \frac{2\pi}{\beta}\Big] \Big)^2 \right] \leq \epsilon_\text{s}$$

For both (PI) and (SCL′)(M) we use a total of $N = 456$ collocation points, namely 256 points $(x, 0)$, $x \in [0, 2\pi]$, for the IC and 100 points equally spaced in $t \in (0, 1]$ to evaluate the period BC. We use $M = 100$ collocation points sampled uniformly at random in the beginning of training and kept fixed throughout for the PDE loss.

**Problem hyperparameters.** For SCL, we take $\epsilon_\text{pde} = 10^{-3}$ and $\epsilon_s = 10^{-3}$. For (PI), we use the weights $\mu$ from (Daw et al., 2023), namely, $\mu_\text{D} = 1$, $\mu_\text{BC} = 100$, and $\mu_\text{IC} = 100$.

**Model.** We used MLPs with 4 hidden layers of 50 neurons.

**Training.** For (SCL′)(I), we used Alg. 2. For the invariance loss, we used $\Sigma = \text{diag}(0.5, 0.1)$ to sample from $(x, t)$. All models were trained for 200 000 epochs using Adam with the default parameters from (Kingma & Ba, 2017) and learning rate of $10^{-3}$ for (PI). For SCL, the dual learning rate was $\eta_d = 10^{-4}$. We decayed the learning rates by a factor of 0.9 every 5000 epochs.

**Testing.** The solution was tested on a dense regular grid of $256 \times 100$ points $(x, t) \in \mathcal{D}$ against its analytical solution.

### E.1.4 SUPERVISED SOLUTION OF BVPs (SEC. 5.4)

For supervised experiments, we formulate an SCL without objective using only data constraints (observational knowledge). Since we use FNOs, that can only make predictions on uniform grids, we replace $\psi_0^{OB}$ in (SCL) with a uniform distribution over a fixed regular grid. The problem the FNOs tackle is that of predicting the solution $u^\dagger$ of a BVP given its IC $h(x, 0)$. Hence, the training data is composed of pairs $(u_j^\dagger, h_j)$ describing ICs and their corresponding solution. We therefore pose the SCL problem

$$\underset{\theta \in \Theta}{\text{minimize}} \quad 0$$

$$\text{subject to} \quad \frac{1}{N} \sum_{n=1}^{N} \Big( u_\theta(h_j)(x_n, t_n) - u_j^\dagger(x_n, t_n) \Big)^2 \leq \epsilon_\text{o}, \quad j = 1, \ldots, J.$$

**Problem hyperparameters.** For SCL, the tolerance was chosen as before, using a coarse hyperparameter search. The final values are reported in Table 6.

**Model.** We used the FNO architecture from (Li et al., 2021) with 64 hidden channels, 128 projection channels, and no lifting channels. The number of modes and layers are reported in Table 6.

**Training and Testing.**    The datasets from (Li et al., 2021) were used for Burgers' and Navier-Stokes equation, whereas the diffusion-sorption dataset was taken from (Takamoto et al., 2022). All models were trained for $500$ epochs using Adam with the default settings from (Kingma & Ba, 2017) with learning rate $10^{-3}$ and batch size of $20$. For SCL, the dual learning rate was $\eta_d = 10^{-4}$. All learning rates were decreased by a factor of $0.5$ every $100$ epochs. All test errors are reported for the model that achieved the lowest validation error during training. The sizes of the training, validation and test sets are reported in Table 6.

## F    ADDITIONAL EXPERIMENTS

### F.1    SOLVING PARAMETRIC FAMILIES OF BVPS

We begin by presenting additional experiments focused on solving parametric families of BVPs (Sec. 5.2) and show how the samples from MH can be used to gain insights into the PDE and the training process.

In what follows, we report the "relative (computational) complexity" of (SCL′) in terms of differential operator evaluations per epoch. Explicitly,

$$\text{Relative complexity} = \frac{\text{\# differential operator evaluations per epoch for (SCL}')}{\text{\# differential operator evaluations per epoch for (PI)}} \times 100\%$$

Recall that in order to evaluate the PDE loss, (PI) uses 1000 collocation points per discretized coefficient $\pi_j$ whereas (SCL′) takes 5000 steps of Alg. 2.

**Convection equation.**    Table 7 considers simultaneously solving all BVPs corresponding to the convection equation with $\beta \in [1, 30]$ and compares (SCL′)(M) with (PI). We see that (SCL′)(M) outperforms or matches (PI) while being more efficient. In particular, (SCL′)(M) significantly outperforms (PI) in terms of relative $L_2$ error for all but the finest discretization where they perform similarly. However, for that discretization, (SCL′)(M) is much more efficient that (PI). In that sense, it strikes a better compromise between error and computational cost. This is even clearer from Fig. 4, particularly when we normalize the $x$-axis in terms of differential operator evaluations.

Table 7: Relative $L_2$ error and computational efficiency for the parametric convection problem.

| Discretization for (PI) | Average Relative $L_2$ Error | | Relative complexity (SCL′) ÷ (PI) |
|---|---|---|---|
| | (PI) | (SCL′)(M) | |
| $\{1.0, 10.0, 20.0, 30.0\}$ | 0.365 | | 125% |
| $\{1.0, 5.0, 10.0, 15.0, 20.0, 25.0, 30.0\}$ | 0.220 | 0.0110 | 71% |
| $\{1.0, 2.0, 3.0, 4.0, 5.0, \ldots, 30\}$ | 0.0476 | | 16% |

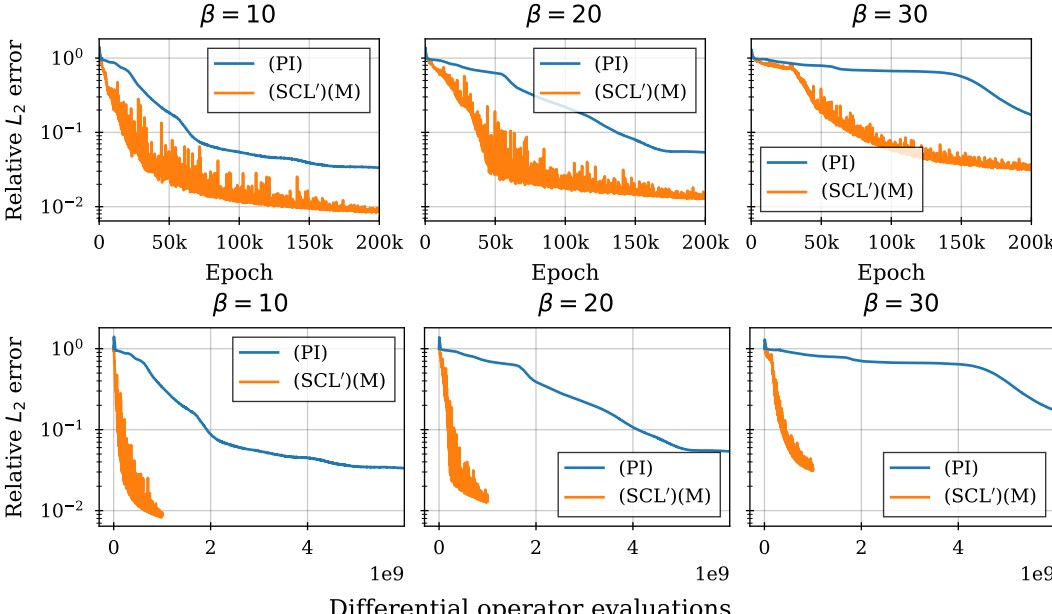

Figure 4: Relative $L_2$ error as a function of training epoch and differential operator evaluations for the parametric convection problem.

**Reaction-diffusion equation.** Additional results for the parametric reaction-diffusion BVP are shown in Table 8. Same as for the convection PDE, $(SCL')(M)$ is more efficient than $(PI)$, striking a better compromise between computational complexity and performance. Indeed, in order to achieve the same error as $(SCL')(M)$, $(PI)$ requires between 5 and 7 times more evaluations of the PDE loss, i.e., of the differential operator $D$, per epoch. This is once again clear when looking at the evolution of the error during training (Fig. 5), especially when the $x$-axis is displayed is terms of PDE evaluations. The distribution of errors across parameters is also more homogeneous for the SCL solution (Fig. 7).

Finally, we can once again inspect the samples from $\psi_0$ throughout training to understand where the advantage of SCL comes from (Fig. 6). First, we do not note any interesting behavior over the $x$-marginal (the samples are mostly uniform and the histogram is therefore omitted). Once again, we see that $(SCL')(M)$ starts by focusing more on earlier times $t$, fitting the solution of the PDE "causally." Additionally, since the diffusion term tends to make the solution more homogeneous for larger times, it is clear that these are regions that are easier to fit and therefore require less attention. Once again, this behavior is not manually encouraged, but arises naturally from Alg. 1. As for the $(\nu, \rho)$, we see that the distributions shift during training, indicating the change in difficulty of fitting the solution of the reaction-diffusion PDE. In the end, the samples for $\rho$ are quite uniform, while we notice that there remains a strong focus on smaller values of $\nu$. Note that these distributions reflect the error patterns of the final solution (Fig. 7).

Table 8: Relative $L_2$ error and computational efficiency for the parametric reaction-diffusion problem.

| Coefficients range | Discretization for (PI) | Average relative $L_2$ error | | Relative complexity $(SCL') \div (PI)$ |
|---|---|---|---|---|
| | | (PI) | (SCL')(M) | |
| $\nu \in [0, 5]$ $\rho \in [0, 5]$ | $\{0.0, 2.5, 5.0\}^2$ | 0.0793 | 0.0126 | 55.6% |
| | $\{0.0, 1.67, 3.33, 5.0\}^2$ | 0.0190 | | 31.3% |
| | $\{0.0, 1.25, 2.5, 3.75, 5.0\}^2$ | 0.0119 | | 20% |
| | $\{0.0, 1.0, 2.0, 3.0, 4.0, 5.0\}^2$ | 0.0105 | | 13.9% |
| $\nu \in [0, 10]$ $\rho \in [0, 10]$ | $\{0.0, 5.0, 10.0\}^2$ | 0.636 | 0.0133 | 55.6% |
| | $\{0.0, 2.5, 5.0, 7.5, 10.0\}^2$ | 0.0228 | | 20% |
| | $\{0.0, 2.0, 4.0, 6.0, 8.0, 10.0\}^2$ | 0.0131 | | 13.9% |
| $\nu \in [1, 20]$ $\rho \in [1, 20]$ | $\{1.0, 10.0, 20.0\}^2$ | 0.841 | 0.0204 | 55.6% |
| | $\{1.0, 5.0, 10.0, 15.0, 20.0\}^2$ | 0.0128 | | 20% |

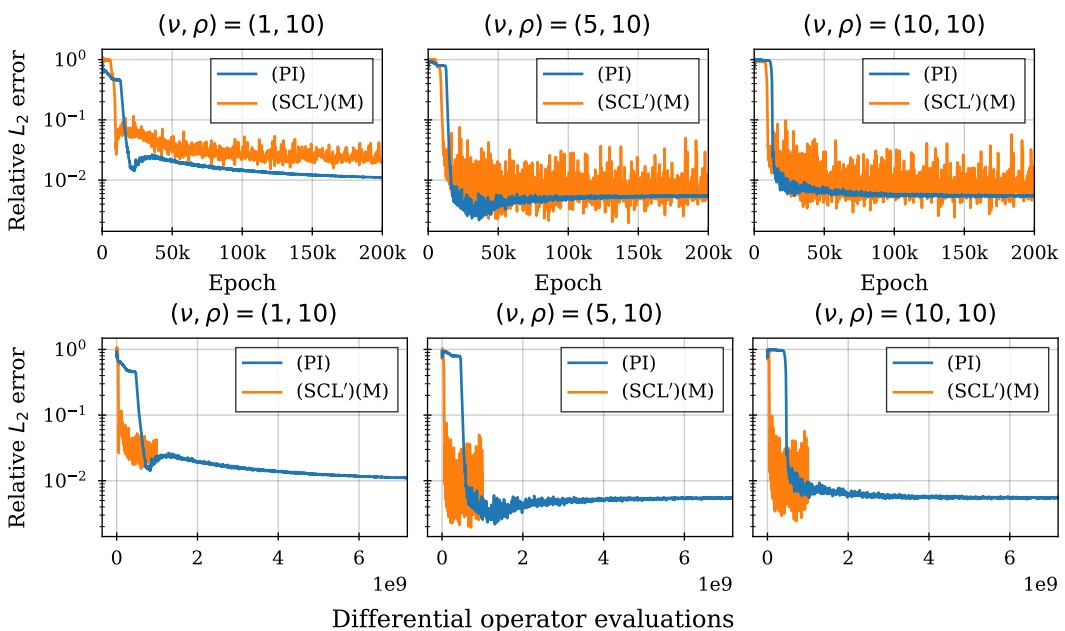

Figure 5: Relative $L_2$ error as a function of training epoch and differential operator evaluations for the parametric reaction-diffusion problem. (PI) uses the discretization $(\nu, \rho) \in \{0.0, 2.0, 4.0, 6.0, 8.0, 10.0\}^2$

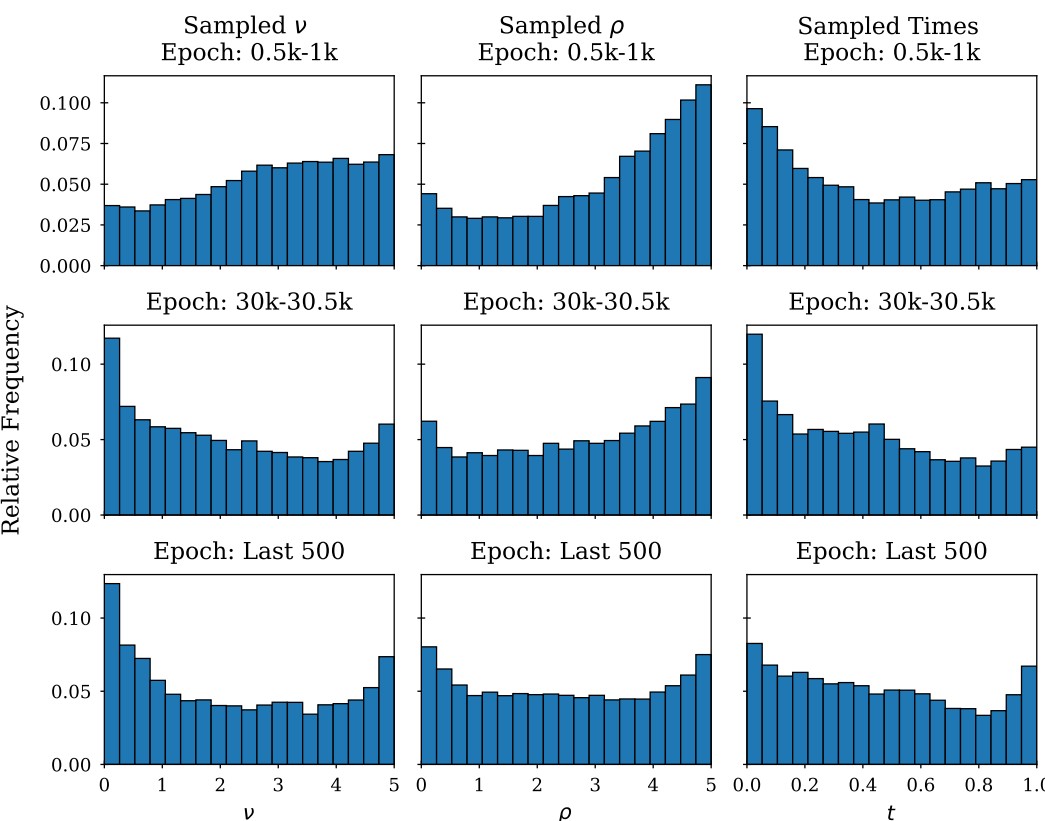

Figure 6: Histogram of (marginalized) MH samples of $\psi_0$ for the parametric reaction-diffusion equation.

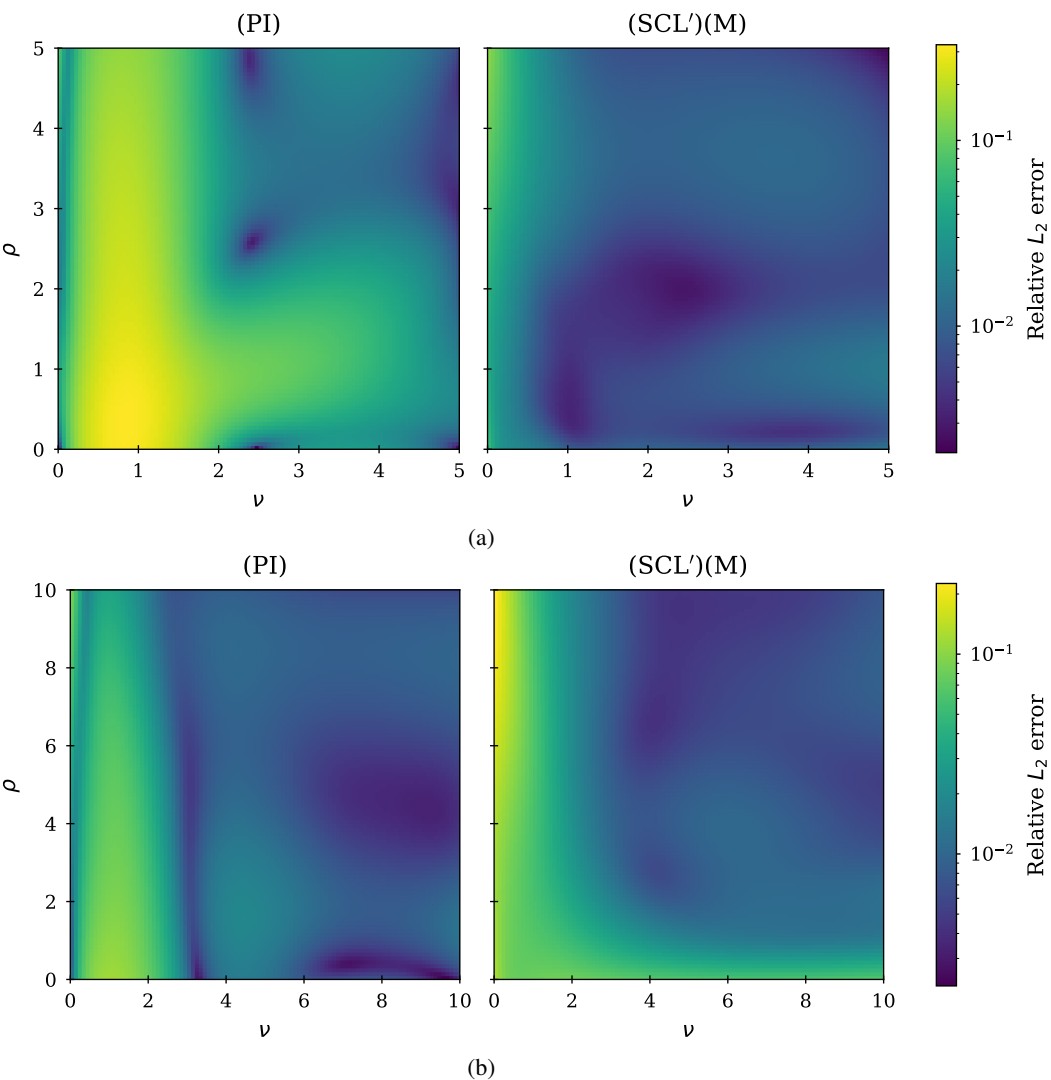

Figure 7: Relative $L_2$ error for reaction-diffusion solutions trained using (SCL$'$) and (PI) with discretization (a) $(\nu, \rho) \in \{0.0, 2.5, 5.0\}^2$ and (b) $(\nu, \rho) \in \{0.0, 2.0, 4.0, 6.0, 8.0, 10.0\}^2$.

**Helmholtz equation.** Once again, we see from Table 8 that (SCL')(M) makes more efficient use of computations than (PI). Indeed, in order to achieve the same error as (SCL')(M), (PI) requires between 3 and 4 times more evaluations of the PDE loss (i.e., of the differential operator $D$) per epoch. This is clear by looking at the evolution of the error during training after normalizing the $x$-axis in terms of computational complexity (Fig. 5). Naturally, taking finer discretizations eventually leads to lower errors (Fig. 9), but the computational cost associated also rises considerably. On the other hand, we keep the computational cost of (SCL') fixed throughout all experiments, showcasing its good performance across scenarios with little to no manipulation.

We can also inspect the samples from $\psi_0$ throughout training to gain a better understanding of the difficulties perceived by the MLP to fit solutions of this problem (Fig. 10). We display only the $x$ and $a_1$ marginals, seen as they display the same behaviors as $y$ and $a_2$ respectively due to the symmetry of the Helmholtz equation. Here, we notice that the distribution of $x$ has an alternating pattern initially. This makes sense seen as the solution of the Helmholtz equation is periodic. SCL clearly picks up on this pattern, focusing on the modes of the solution. As training continues, the sampling becomes more uniform, although with a focus on the boundaries of the domain where the MLP clearly has difficulties fitting the solution. With respect to the problem coefficients, we notice that $\psi_0$ concentrates on larger values of $a_1$, especially in the beginning of training. These are indeed coefficients for which the solution of the problem is harder to fit (as evidenced by Fig. 9).

Table 9: Relative $L_2$ error and computational efficiency for the parametric reaction-diffusion problem.

| Coefficients range | Discretization for (PI) | Average relative $L_2$ error | | Relative complexity (SCL') ÷ (PI) |
|---|---|---|---|---|
| | | (PI) | (SCL')(M) | |
| $a_1 \in [1, 2]$ $a_2 \in [1, 2]$ | $\{1.0, 1.5, 2.0\}^2$ | 0.0307 | | 55.6% |
| | $\{1.0, 1.25, 1.5, 1.75, 2.0\}^2$ | 0.00593 | 0.0125 | 20% |
| | $\{1.0, 1.2, 1.4, 1.6, 1.8, 2.0\}^2$ | 0.00463 | | 13.9% |
| $a_1 \in [1, 3]$ $a_2 \in [1, 3]$ | $\{1.0, 2.0, 3.0\}^2$ | 1.34 | | 55.6% |
| | $\{1.0, 1.5, 2.0, 2.5, 3.0\}^2$ | 0.00943 | 0.0549 | 20% |
| | $\{1.0, 1.4, 1.8, 2.2, 2.6, 3.0\}^2$ | 0.00953 | | 13.9% |

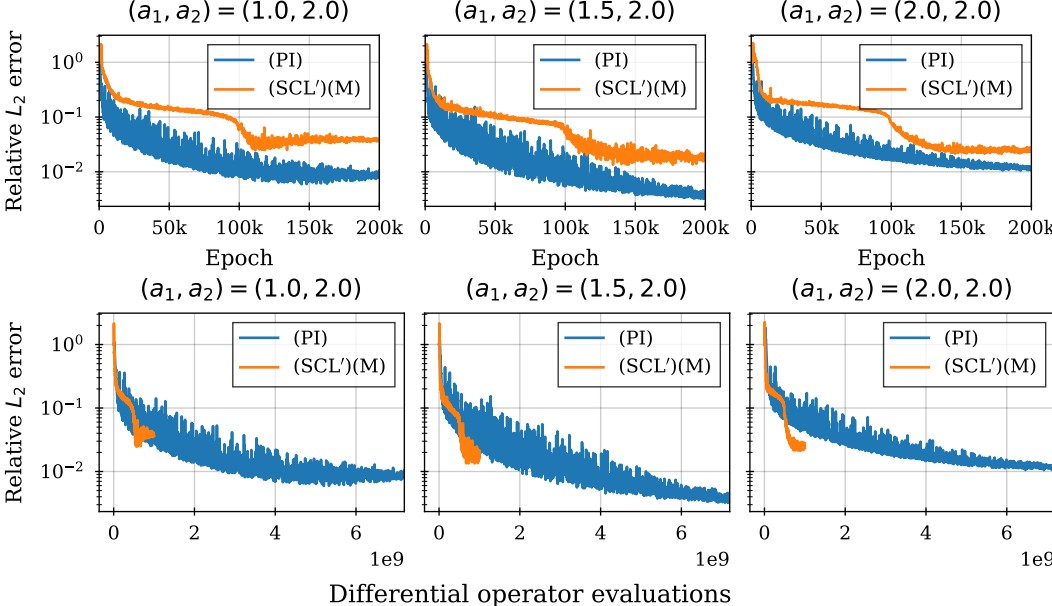

Figure 8: Relative $L_2$ error as a function of training epoch and differential operator evaluations for the parametric Helmholtz problem. (PI) uses the discretization $(a_1, a_2) \in \{1.0, 1.2, 1.4, 1.6, 1.8, 2.0\}^2$

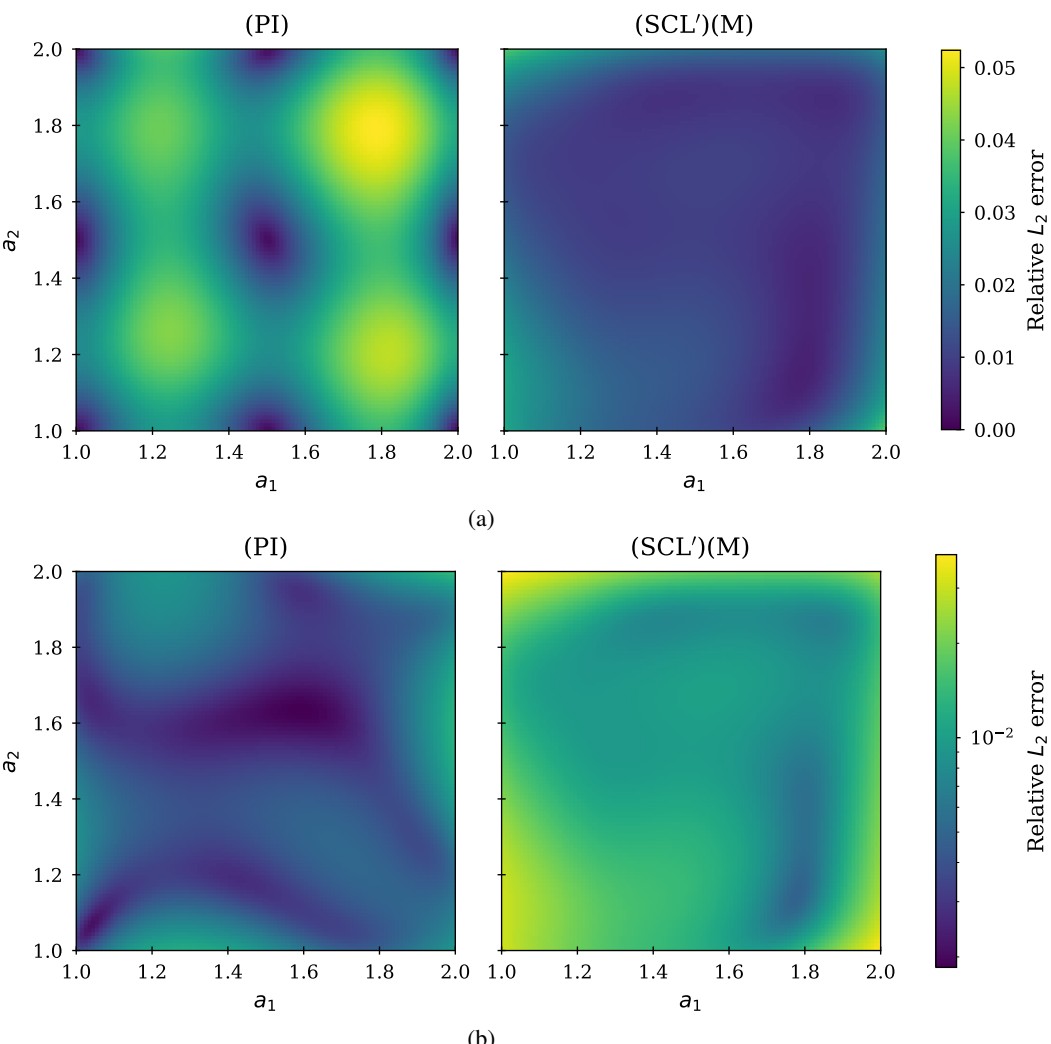

Figure 9: Relative $L_2$ error for Helmholtz solutions trained using (SCL$'$) and (PI) with discretization (a) $(a_1, a_2) \in \{1.0, 1.5, 2.0\}^2$ and (b) $(a_1, a_2) \in \{1.0, 1.2, 1.4, 1.6, 1.8, 2.0\}^2$.

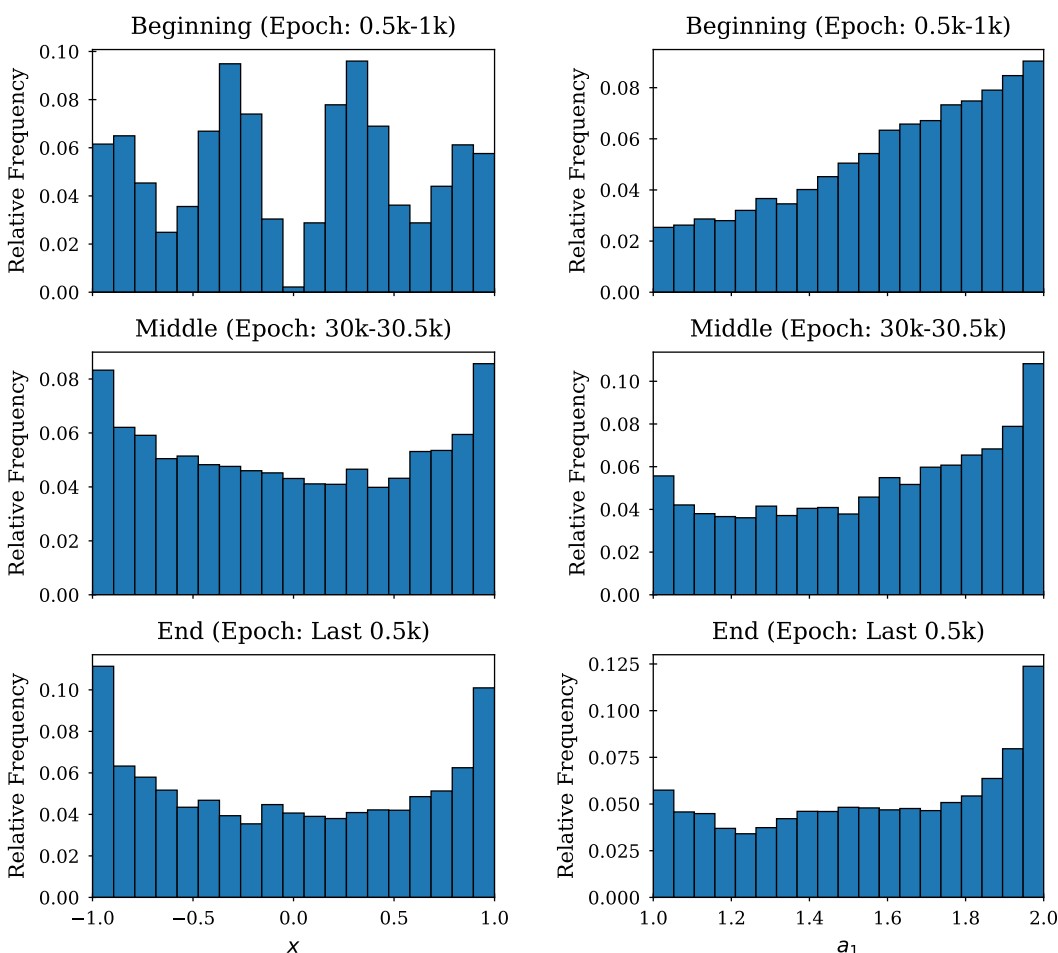

Figure 10: Histogram of (marginalized) MH samples of $\psi_0$ for the parametric Helmholtz equation.

### F.2 SUPERVISED SOLUTION OF BVPS

**Burgers' equation.** Fig 11 shows the box plots for the relative $L_2$ error across samples. They show that not only is the average error across samples smaller when using (SCL)(O), but in fact the whole error distribution is shifted down. This is due to the variety of weights given to different samples, weights that in fact vary during the training process (Fig 12). The few large dual variables are related to ICs that are harder to fit and can provide important information for data collection or architecture improvements. We have already shown which ICs are harder for FNOs to fit in Fig. 3.

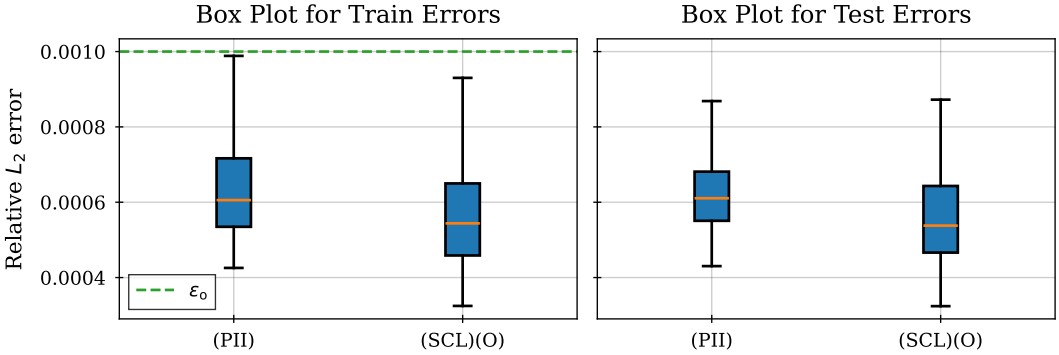

Figure 11: Distribution of train and test errors (across data points) for the Burgers' equation (orange line indicates the median).

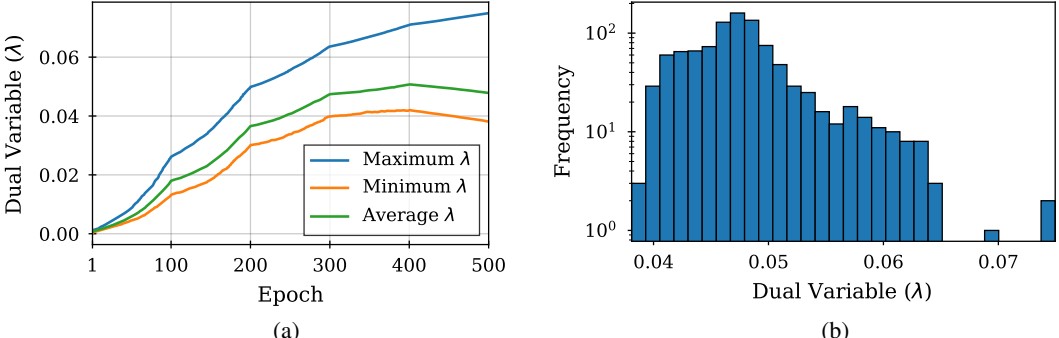

Figure 12: Dual variables obtained by Alg. 1 for the Burgers' equation: (a) evolution during training and (b) distribution of the dual variables after training.

**Diffusion-sorption equation.** We once again display the distribution of the relative $L^2$ errors across training and test data points for models trained using (PII) and (SCL)(O) (Fig. 13). Here, we clearly see that by bounding the maximum error rather than minimizing its average leads to a more homogeneous fit across samples. This is once again due to the different weight assigned to each data sample, weights that also evolve throughout training (Fig 14a). By inspecting the ICs with large and small values of $\lambda$, we notice the pattern showcased in Fig 14b, where IC with either large or small magnitude are more challenging to fit than those with moderate ones.

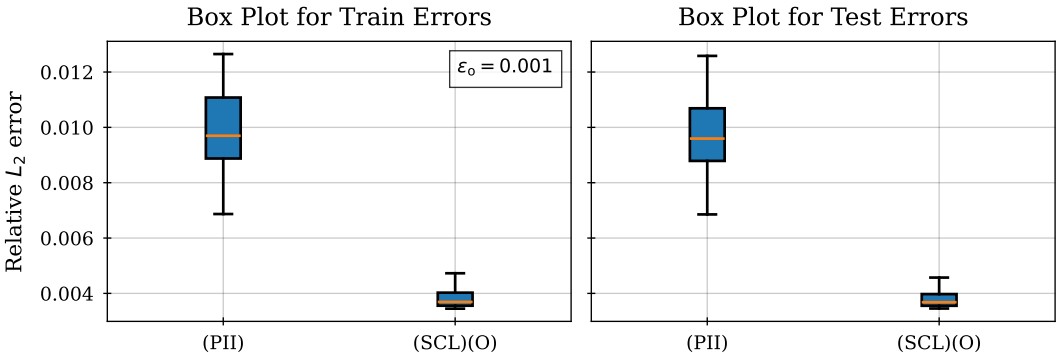

Figure 13: Distribution of train and test errors (across data points) for the diffusion-sorption equation (orange line indicates the median).

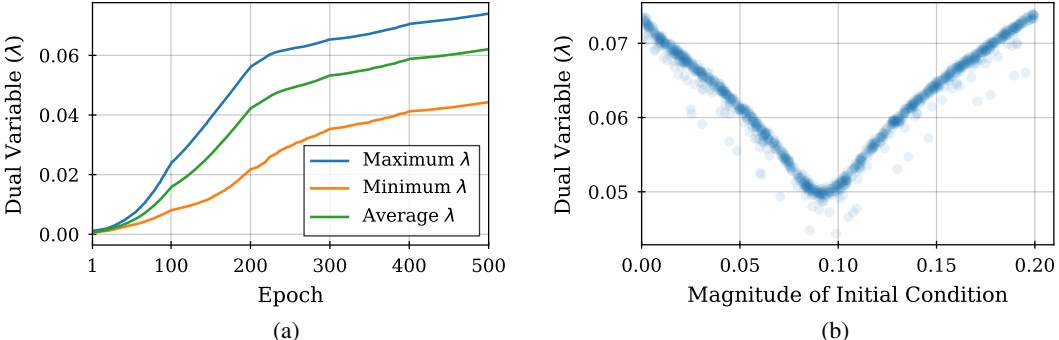

Figure 14: Dual variables obtained by Alg. 1 for the diffusion-sorption equation: (a) evolution during training and (b) value as a function of the IC magnitude.

Table 10: Relative $L_2$ error on test set (mean $\pm$ standard deviation).

| | $\nu$ | (PII) | (SCL)(**O**) |
|---|---|---|---|
| **Burgers'** | $10^{-3}$ | $0.0540 \pm 0.0027\,\%$ | $0.0444 \pm 0.0020\,\%$ |
| | $10^{-3}$ | $4.29 \pm 0.40\,\%$ | $3.31 \pm 0.16\,\%$ |
| **Navier-Stokes** | $10^{-4}$ | $32.2 \pm 0.87\,\%$ | $29.9 \pm 0.54\,\%$ |
| | $10^{-5}$ | $27.6 \pm 0.63\,\%$ | $26.0 \pm 0.33\,\%$ |
| **Diffusion-Sorption** | | $0.274 \pm 0.049\,\%$ | $0.218 \pm 0.036\,\%$ |

**Navier-Stokes equation.** We start by displaying an extended version of Table 2 including standard deviations of results over 10 runs to show the consistency of our results across random seeds (Table 10). We then turn to Fig 15 which shows that (SCL)(O) not only improves the average relative $L_2$ error, but its entire distribution across train and test data points. For the Navier-Stokes equations, however, it is harder to find a relation between IC properties and the difficulty of fitting the solution. That is because, as we show in Fig 16, the dual variables do not have such extremely different values. This is certainly due to the fact that the tolerance $\epsilon_o$ is set very loose (0.01) and that in these situations, all ICs are similarly difficult to fit. Still, some ICs have outlier values of $\lambda$ (Fig 16b), which does point to the fact that the FNO does struggle more to fit certain conditions. That being said, it not easy to identify what in those conditions make them hard (Fig 17). Nevertheless, this is not an issue as we need not know beforehand which ICs are challenging: suffices it to run Alg. 1 to solve (SCL)(O).

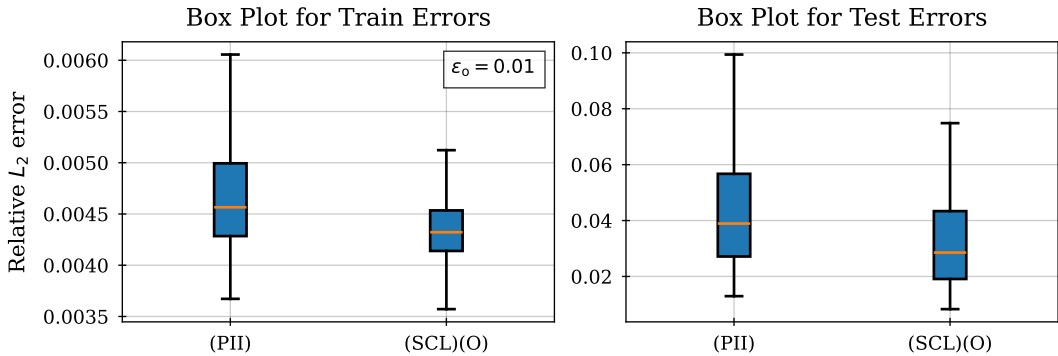

Figure 15: Distribution of train and test errors (across data points) for the Navier-Stockes equation with $\nu = 10^{-3}$ (orange line indicates the median).

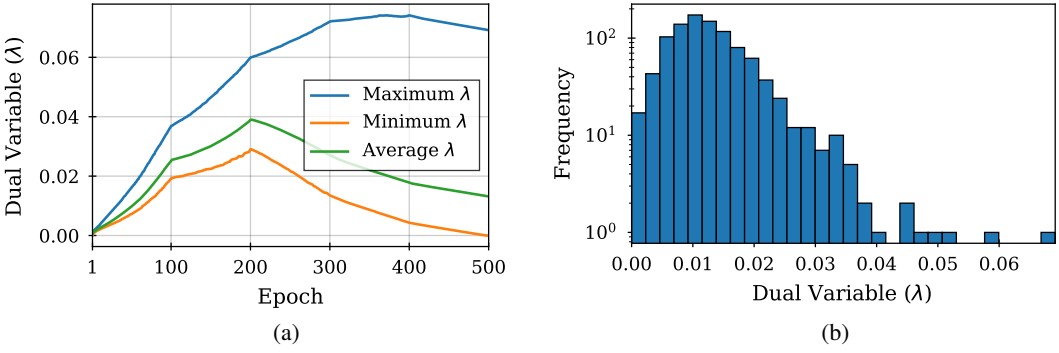

Figure 16: Dual variables obtained by Alg. 1 for the Navier-Stokes equation with $\nu = 10^{-3}$: (a) evolution during training and (b) distribution of the dual variables after training.

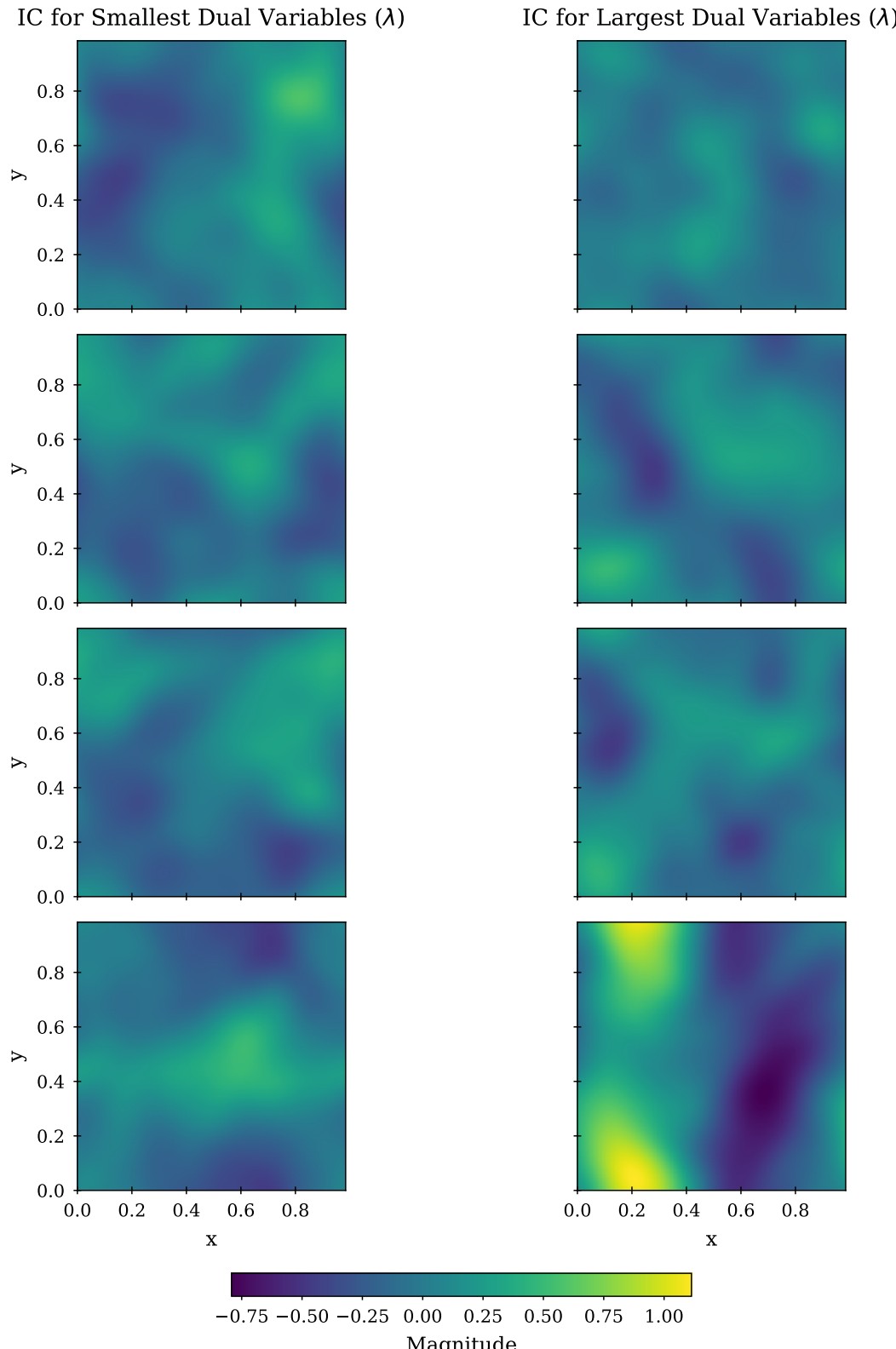

Figure 17: Initial conditions corresponding to the smallest and largest final dual variables for Navier-Stokes equation with $\nu = 10^{-3}$.

# G ADDITIONAL RELATED WORK

## G.1 PHYSICS-INFORMED NEURAL NETWORKS

Fitting MLPs to the solution of BVPs goes back to (Psichogios & Ungar, 1992; Dissanayake & Phan-Thien, 1994; Lagaris et al., 1998). The advent of differentiable programming and automatic differentiation, however, lead to an increased interest in this approach, which has since been used to tackle both forward and inverse problems involving a variety of PDEs (see, e.g., (Raissi et al., 2019; Wight & Zhao, 2021; Chen et al., 2020; Lu et al., 2021b; Basir & Senocak, 2022; Yu et al., 2022; Xu et al., 2023)). This led to new architectures tailored for PDEs (Raissi et al., 2019; Fathony et al., 2021; Gao et al., 2021; Wang et al., 2021a; Kang et al., 2023; Moseley et al., 2023; Cho et al., 2024; Chalapathi et al., 2024), leveraging positional embedding (Wang et al., 2021b) and adaptive activation functions (Jagtap et al., 2020).

**Training PINNs.** PINNs tend to be very sensitive to training hyperparameters, particularly the choice of collocation points and loss weights. Many works have theoretically and empirically investigated the origins of these issues (Krishnapriyan et al., 2021; Markidis, 2021; Wight & Zhao, 2021; Wang et al., 2021a; 2022b;a; Grossmann et al., 2024). Based on these observations, adaptive heuristics have been proposed to select the collocation points based on importance sampling (Nabian et al., 2021; Wu et al., 2023), adversarial training Wang et al. (2022a), rejection sampling (Daw et al., 2023), and causality-inspired rules (Penwarden et al., 2023; Wang et al., 2024). Similarly, empirical rules for determining the loss weights [$\mu$ in (PI)] have been developed using the magnitude of the gradients (Wang et al., 2021a), eigenvalues of the neural tangent kernel (Wang et al., 2022b), inverse-Dirichlet weighting (Maddu et al., 2022), soft attention mechanisms (McClenny & Braga-Neto, 2023), and (augmented) Lagrangian formulations (Lu et al., 2021b; Basir & Senocak, 2022). Other works have addressed these challenges by changing the problem formulation inspired by traditional numerical methods (Kharazmi et al., 2021; Chiu et al., 2022; Patel et al., 2022), proposing different objective functions (Yu et al., 2022; Son et al., 2021), and using sequential (Wight & Zhao, 2021; Krishnapriyan et al., 2021) and transfer learning (Goswami et al., 2020; Chakraborty, 2021; Desai et al., 2022) techniques.

In contrast to these approaches, we address these issues jointly by using worst-case losses and constrained learning to obviate these hyperparameters. In fact, we prove that the constrained learning problems we pose yield (weak) solutions of BVPs. Hence, it is not enough to use either adversarial training to estimate the worst-case loss as in (Wang et al., 2022a) or constrained learning to manipulate the loss weights as in (Lu et al., 2021b; Basir & Senocak, 2022). Both are required simultaneously. Leveraging findings from adversarially robust learning, we also replace the gradient methods used in (Wang et al., 2022a), i.e., the technique from (Mądry et al., 2018), by the sampling-based approach in (Robey* et al., 2021).

## G.2 NEURAL OPERATORS

In contrast to the MLPs and convolutional NNs (CNNs) typically used in PINNs, NOs are NNs capable of handling infinite-dimensional inputs and outputs. They can therefore be trained to find BVP solutions for different IC or forcing functions. Many different architectures have been proposed, such as DeepONets (Lu et al., 2021a), FNOs (Li et al., 2021), and NO based on U-Nets (Gupta & Brandstetter, 2023). FNOs in particular have become quite popular and garnered many efforts towards addressing their limitations, such as improving memory efficiency (Rahman et al., 2023), designing equivariant FNOs (Helwig et al., 2023), extending FNOs to general geometries (Li et al., 2023), factorizing the Fourier transform (Tran et al., 2023), and leveraging multiwavelets (Gupta et al., 2021).

**Training NOs.** Regardless of these improvements, the vast majority of NOs are trained in a supervised manner by minimizing their average error across samples as in (PII) (see, e.g., (Lu et al., 2021a; Li et al., 2020; Kovachki et al., 2023)). Unless substantial domain knowledge has been used during data collection, challenging cases may be underrepresented in the dataset, which could hinder the accuracy of the NO. Although semi-supervised techniques involving PDE losses have also been used (Li et al., 2024), computing the space-time derivatives needed to evaluate $D_\pi[u]$ is challenging for NOs.

In this paper, we do not develop new NO architectures, but focus on the problem of training them. Explicitly, rather than targeting the average error, we target the maximum error across samples. This is much better suited to handle the heterogeneous difficulty in fitting the data. We also incorporate structure in the solution during training, without the need to design new architectures.

### G.3 CONSTRAINED AND ADVERSARIALLY ROBUST LEARNING

The main tool used in the development of SCL is constrained learning, or more specifically, robustness-constrained learning. Constrained learning is a technique to train ML systems under requirements, such as fairness (Kearns et al., 2018; Cotter et al., 2019; Chamon & Ribeiro, 2020; Chamon et al., 2023) and robustness (Chamon & Ribeiro, 2020; Robey* et al., 2021; Hounie et al., 2023a; Chamon et al., 2023), or to handle applications in which we want to attain good performance with respect to more than one metric. As in unconstrained learning, it is formulated as statistical risk minimization problem, albeit with constraints. Despite its non-convexity in virtually every modern ML task, certain duality properties hold when using sufficiently expressive parametrizations, leading to a practical learning rule with generalization guarantees (Chamon & Ribeiro, 2020; Chamon et al., 2023). These duality results have also been exploited to automatically adapt each constraint specification to their underlying difficulty, striking better compromises between objective and requirements Hounie et al. (2023b).

Aside from dealing with the nominal accuracy vs. robustness trade-off typical in ML systems, constrained learning has itself been used to optimize robust losses. Typically, this is done using some combination of gradient ascent and random initialization, restart, and pruning heuristics (Goodfellow et al., 2015; Mądry et al., 2018; Dhillon et al., 2018; Wu et al., 2020; Cheng et al., 2022). Using semi-infinite optimization techniques, however, these deterministic methods can be replaced by a sampling approach that has been successful in a variety of domains (Robey* et al., 2021). Different MCMC methods (Robert & Casella, 2004) have been used in this context, including Langevin Monte Carlo (LMC) (Robey* et al., 2021) and Metropolis-Hastings (MH) (Hounie et al., 2023a).

