# OpenReview forum: "Solving Differential Equations with Constrained Learning"
_ICLR.cc/2025/Conference — ICLR 2025 Poster_

### Official Review · Reviewer_RnPx · 2024-10-20

**Soundness:** 2
**Presentation:** 2
**Contribution:** 2
**Rating:** 5
**Confidence:** 3

**Summary:**

This paper developed SCL, a technique for solving BVPs based on constrained learning.
It then developed a practical algorithm to tackle these problems and showcased its performance across a variety of PDEs.
SCL not only yields accurate BVP solutions, but tackles many of the challenges faced by previous methods, such as extensive hyperparameter tuning and computational costs.

**Strengths:**

- The authors tackle important problems in physics-informed learning: reducing computational costs and avoiding hyperparameter tuning.

- The experimental results show that the proposed formulation outperforms baselines in some cases.

**Weaknesses:**

- The key contribution, Algorithm 1, can be seen as a variant of DNN training with learnable loss weights, which can often be found in multi-task learning, lacking novelty.

- The dependence on random seeds is unknown, potentially causing reproducibility issues. Error bars should be added.

- The paper is difficult to follow. The paper benefits from paragraph writing, and please present information in relevant paragraphs and sections. It was challenging to discern whether certain sentences conveyed key ideas or were merely supplementary notes.

- I have several critical questions. Please see Questions below. I am open to discussion.

- (Minor) (Lines 1730, 1733, etc.) Fig -> Fig.

**Questions:**

- The proposed formulation seems to be a rephrasing of PDE boundary value problems with invariance, observation data into a single constraint problem. The resulting Lagrangian problem is then solved straightforwardly in Algorithm 1, reminiscent of standard multi-task learning algorithms. Is there more to it than this?

- (Lines 214-) "Compared to penalty methods, (Dˆ -CSL) does not require extensive tuning of coefficients [such as μ in (PI)] and guarantees generalization individually for each constraint (near-optimality and near-feasibility) rather than for their aggregated value.": Could you elaborate on this? It seems to be a key statement regarding the paper's contribution. Could you provide some references to support this?

- (Lines 58-) "Instead, we use semi-infinite constrained learning techniques to develop a hybrid sampling-optimization algorithm that tackles both problems jointly without extensive hyperparameter tuning and training heuristics (Sec. 4).":  I am not fully convinced by this claim. Algorithm 1 still includes hyperparameters, such as $\eta_p$ and $\eta_d$, and training heuristics are still necessary.

- (Lines 472-) "it (causality) arises naturally by solving (SCL′)(M). As training advances, however, Alg. 1 shifts focus to fitting higher convection speeds β. Note that this occurs without any prior knowledge of the problems or manual tuning.": Could you clarify why causality naturally arises in this context?

- Under what conditions, does Algorithm 1 converge?  (Chamon et al., 2023; Elenter et al., 2024) could be valuable references, to my understanding.

---

> ### Author Response · Authors · 2024-11-21
>
> > **W1:** The key contribution, Algorithm 1, can be seen as a variant of DNN training with learnable loss weights, which can often be found in multi-task learning, lacking novelty.
>
> **Response:**
> While Alg. 1 is certainly important, we do not believe it to be the key contribution of this paper. In fact, as we note in our response to Reviewer 4CoR, Alg. 1 is based on constrained optimization duality, so that the updates in step 8-9 are reminiscent of classical primal-dual or dual ascent methods going back to [Arrow et al., "Studies in linear and non-linear programming," 1958]. What is crucial of Alg. 1 is the combination of such primal-dual methods with the sampling-based approximation of worst-case losses (steps 4-7).
>
> We prove that this combination is needed to solve BVPs in Prop. 3.2. It is therefore not enough to use either constrained optimization and Lagrangian formulations as in (Lu et al., 2021b; Basir & Senocak, 2022) or worst-case losses as in (Wang et al., 2022a; Daw et al., 2023). Indeed, note that using a constrained formulation with fixed collocation points does not recover a solution of the convection PDE (Fig. 2) and that R3 from (Daw et al., 2023) fails to recover a solution for the Eikonal equation (Table 1). In contrast, Alg. 1 succeeds in both cases (see Table 1) because it **solves (1) constrained learning problems with (2) worst-case losses**.
>
> 1. **Solving constrained learning problems**: As Reviewer x438 notes, Alg. 1 is based on constrained optimization duality and tackles max-min problems of the form ($\hat{\text{D}}$-CSL) from Sec. 3.1. This is also the case for other approaches involving (augmented) Lagrangian formulations of PINNs, such as (Lu et al., 2021b; Basir & Senocak, 2022). Yet, this is **not** the same as solving (SCL), the constrained optimization problem arising from Prop. 3.2 because they are not convex.
>
>     We overcome this issue by posing a constrained *learning* problem (i.e., with statistical losses), namely (SCL), which allows us to use non-convex duality results from (Chamon & Ribeiro, 2020; Chamon et al., 2023) to provide approximation guarantees between (P-CSL) and ($\hat{\text{D}}$-CSL). Combined with results from (Cotter et al., 2019; Elenter et al., 2024), we can obtain convergence guarantees for the variant of Alg. 1 presented in (3) (as we explain in the paper, line 377) towards a (probably approximately) near-feasible and near-optimal solution of (SCL) (see response to Q5 for more details on convergence).
>
> 2. **Worst-case losses**: Using steps 8-9 with fixed collocation points (regardless of their distribution) is still not enough to obtain a solution of a BVP: Prop. 3.2 also requires the use of worst-case losses. Alg. 1 leverages Prop. 3.1 to replace these worst-case losses by statistical losses against specially designed distributions, namely the $\psi_0$ in steps 4-7. It is the combination of these two properties (duality of constrained learning problems and worst-case formulation) that ensures that Alg. 1 seeks a weak solution of the BVP.
>
> More explicitly, our contributions, as summarized along lines 54-65, are therefore:
>
> - prove that obtaining a (weak) solution of (BVP) is equivalent to solving a constrained learning problem with worst-case losses, namely (PIII) (Prop. 3.2);
> - incorporate other scientific prior knowledge to (PIII), such as structural (e.g., invariance) and observational (e.g., measurements, known solutions) information,  without resorting to specialized models or data transforms (SCL, Sec. 3.2);
> - develop a practical hybrid sampling-optimization algorithm (Alg. 1) that requires neither tuning weights to balance different objectives nor the careful selection of collocation points. Alg. 1 also yield trustworthiness measures by capturing the difficulty of fitting PDE instances or data points (Sec. 4);
> - illustrate the effectiveness (accuracy and computational cost) of this method in a diverse set of PDEs, NN architectures (e.g., MLPs and NOs), and problem types (solving a single or a parametric family of PDEs; interpolating known solutions; identifying PDE instances or data points that are difficult to fit, Sec. 5).
>
> As Reviewer x438 and 4CoR suggest, we will include this explicit list in the camera-ready. It is worth noting that Prop. 3.2 is not trivial in that it establishes limits under which learning approaches work. Indeed, they can only find very smooth solutions for high-dimensional state spaces (essentially, solutions in the Sobolev space $W^{(d+1)/4,2}$). This can be an issue for large-scale dynamical systems, such as those found in smart grid applications, or when transforming higher-order PDEs in higher-dimensional first-order systems.

---

> ### Author Response · Authors · 2024-11-21
>
> > **W2:** The dependence on random seeds is unknown, potentially causing reproducibility issues. Error bars should be added.
>
> **Response:**
> The reviewer has a point. We are running additional experiments with different seeds for the camera-ready. In the meantime, we report here preliminary results to reassure them that our results are not cherry-picked. We ran ten experiments with different seeds for the convection equation both in the case of solving a specific PDE (Sec. 5.1, $\beta=30$ as in Table 1) and solving a parametric family of BVPs (Sec. 5.2, $\beta \in [1, 30]$ compared to the finest discretization in Table 7 and Fig. 1a).
>
> - Solving a specific PINN: mean (standard deviation)
>     * **PINN:** 1.19 (0.56) %
>     * **R3:** 1.00 (0.09) %
>     * **(SCL)(M):** 1.02 (0.39) %
> - Solving parametric families of BVPs: mean (standard deviation)
>     * **(PI):** 4.03 (3.39) %
>     * **(SCL')(M):** 1.11 (0.22) %

---

> ### Author Response · Authors · 2024-11-21
>
> > **W3:** The paper is difficult to follow. The paper benefits from paragraph writing, and please present information in relevant paragraphs and sections. It was challenging to discern whether certain sentences conveyed key ideas or were merely supplementary notes.
>
> **Response:**
> We will copy-edit the paper for clarity (particularly as we shorten the introductory sections to make space for additional material suggested by Reviewer x438). However, we do not see the need to make major changes to the presentation (aside from the list of contributions above) since other reviewers do not appear to have issues with it (our presentation scores range from 3 to 4). We understand, however, that clarity is in the eye of the beholder and would be happy to modify and add clarifications to specific parts that the reviewer found particularly difficult to follow.

---

> ### Author Response · Authors · 2024-11-21
>
> > **W4:** (Minor) (Lines 1730, 1733, etc.) Fig -> Fig.
>
> **Response:**
> We have fixed it and proofread the manuscript for other typos.

---

> ### Author Response · Authors · 2024-11-21
>
> > **Q1:** The proposed formulation seems to be a rephrasing of PDE boundary value problems with invariance, observation data into a single constraint problem. The resulting Lagrangian problem is then solved straightforwardly in Algorithm 1, reminiscent of standard multi-task learning algorithms. Is there more to it than this?
>
> **Response:**
> While the transformations mentioned by the reviewer may appear straightforward, they are not. In fact, none of the derivations needed to obtain Alg. 1 from (BVP) hold in general.
>
> - **From (BVP) to (SCL)**: In Prop. 3.2, we prove that a weak solution of (BVP) can be obtained by **solving (1) a constrained learning problems with (2) worst-case losses**. Hence, as we detail in W1, solving a constrained problem is not enough to solve (BVP). In fact, using a constrained formulation with fixed collocation points fails to find a solution to the convection PDE (Fig. 2), while Alg. 1 succeeds (Table 1). This relation is not trivial and establishes limitations of learning approaches (see response to W1). By then applying Prop. 3.1, we are able to transform the worst-case losses into statistical losses (against the specially crafted distributions $\psi_\alpha$), leading to (SCL).
>
> - **From (SCL) to the empirical dual problem (3)**: As we explain in Sec. 3.1 and W1, the relation between (SCL) and its Lagrangian dual of the form ($\hat{\text{D}}$-CSL) is not straightforward due to non-convexity. This is an issue because primal-dual and dual ascent methods such as Alg. 1 and other Lagrangian-based approaches (Lu et al., 2021b; Basir & Senocak, 2022) solve dual problems and not (SCL). We overcome this issue by using statistical losses and non-convex duality results from (Cotter et al., 2019; Chamon & Ribeiro, 2020; Chamon et al., 2023; Elenter et al., 2024), obtaining convergence guarantees for (3), a variant of Alg. 1 (see Q5 for more details on convergence).
>
> It is important to contrast Alg. 1 with other primal-dual methods from multi-task learning and previous works (Lu et al., 2021b; Wang et al., 2021a; Basir & Senocak, 2022; Wang et al., 2022b; Maddu et al., 2022; McClenny & Braga-Neto, 2023). Indeed, Prop. 3.2 shows that the limitations of previous approaches are not methodological, but epistemological. It is not an issue of *how* they solve the problem, but *which* problem they solve. Alg. 1, however, (approximately) solves (SCL), a constrained problem with worst-case losses, and therefore (approximately) solves (BVP). As we illustrated in W1, this issue is not merely theoretical and can cause failures in practice (see Table 1 and Fig. 2).

---

> ### Author Response · Authors · 2024-11-21
>
> > **Q2:** (Lines 214-) "Compared to penalty methods, (\^{D} -CSL) does not require extensive tuning of coefficients [such as $\mu$ in (PI)] and guarantees generalization individually for each constraint (near-optimality and near-feasibility) rather than for their aggregated value.": Could you elaborate on this? It seems to be a key statement regarding the paper's contribution. Could you provide some references to support this?
>
> **Response:**
> This statement is based on the references cited immediately after on line 217, namely (Chamon & Ribeiro, 2020; Chamon et al., 2023). When solving problems such as (PI), i.e., where losses are aggregated into a single objective with fixed hypermarapeters (weights) $\mu$, classical learning theory only provides generalization guarantees about the value of that aggregate loss. In other words, we can show that
> $$
> \mu_D \ell_D(\theta^\star) + \mu_{BC} \ell_{BC}(\theta^\star)
>     \approx \mathbb{E} \big[ \mu_D \ell_D(\theta^\star) + \mu_{BC} \ell_{BC}(\theta^\star) \big],
> $$
> where the empirical losses $\ell$ are defined in (PI). This is different from saying that $\ell_D(\theta^\star) \approx \mathbb{E} \big[ \ell_D(\theta^\star) \big]$ and $\ell_{BC}(\theta^\star) \approx \mathbb{E} \big[ \ell_{BC}(\theta^\star) \big]$, which is required to obtain approximate solutions of (PIII) [and (SCL)]. This is what using the max-min problem ($\hat{\text{D}}$-CSL) enables. As we argued in Q1, this is not straightforward for non-convex optimization problems, such as (PI) and (SCL). We overcome this issue by using the non-convex duality results from (Chamon et al., 2023, Thm. 1). Combined with results from (Cotter et al., 2019; Elenter et al., 2024), we can obtain convergence guarantees for the variant of Alg. 1 presented in (3) (as we explain in the paper, line 377) towards a (probably approximately) near-optimal and near-feasible solution of (SCL) (see Q5 below for more details on convergence). Using Prop 3.2, we can finally show that such a solution is in fact also a (probably approximately) weak solution of (BVP).
>
> Since these learning theoretic details are not the focus of this work, we did not emphasize them in the manuscript. But we agree with the reviewer that these are needed to better understand the distinction between (SCL) and (PI). We will expand on these points in the text and include a more thorough discussion on the subject in the appendix.

---

> ### Author Response · Authors · 2024-11-21
>
> > **Q3:** (Lines 58-) "Instead, we use semi-infinite constrained learning techniques to develop a hybrid sampling-optimization algorithm that tackles both problems jointly without extensive hyperparameter tuning and training heuristics (Sec. 4).": I am not fully convinced by this claim. Algorithm 1 still includes hyperparameters, such as $\eta_p$ and $\eta_d$, and training heuristics are still necessary.
>
> **Response:**
> - **Hyperparameters**: Alg. 1 has hyperparameters, namely step sizes ($\eta_p$, $\eta_d$) and the number of samples in steps 4-7 ($N$). This is already less than for (PI) or other Lagrangian-based methods, such as (Wang et al., 2021a; Wang et al., 2022b; Maddu et al., 2022; McClenny & Braga-Neto, 2023), that must choose a weight $\mu$ for each loss. But most importantly, the nature of these hyperparameters is fundamentally different. The weights $\mu$ from (PI), in particular, need to be retuned whenever the PDE (or parameters) changes. Even the addition of collocation points to (PI) may change the balance of the losses and require adjusts to $\mu$. That is without considering the fact that using fixed $\mu$ do not provide a solution of (BVP) (see response to W1 and W3). As for the collocation points, they are sampled from $\psi_0$ to estimate worst-case losses. Hence, Alg. 1 can often operate with smaller $N$ than when sampling uniformly at random (particularly when solving parametrized families of PDEs as in Fig. 1a).
>
> - **Training heuristics**: Alg. 1 does *not* rely on training heuristics (such as adaptive or causal sampling, *ad hoc* weight updates, or conditional updates). All steps are justified as approximations in the manuscript. More specifically,
>
>     * steps 4-7 are empirical approximations of expections with respect to $\psi_0$, which Prop. 3.1 show approximate the worst-case losses in (SCL);
>     * this sampling could be performed by any MCMC method. We use the Metropolis-Hastings algorithm described in Alg. 2 (Appendix C), which provides approximate samples from $\psi_0$ [see, e.g., (Robert & Casella, 2004) and response to Reviewer JwZx for a more detailed discussion of its convergence properties];
>     * steps 8-9 describe a traditional primal-dual (gradient descent-ascent) algorithm for solving max-min problems such ($\hat{\text{D}}$-CSL). Guarantees on generalization of this solution are given in (Chamon et al., 2023), which in combination with results from (Cotter et al., 2019; Elenter et al., 2024) provides guarantees that (3), a variant of Alg. 1, yields (probably approximately) near-optimal and near-feasible solution of (SCL) (see Q5 for more details);
>     * Prop 3.2 provides the last result needed to show that solutions of (SCL) approximate weak solutions of (BVP).
>
> We understand, however, that we may have missed something in the reviewer's comment. Please do not hesitate to let us know if this is the case, we would be happy to address their concerns.

---

> ### Author Response · Authors · 2024-11-21
>
> > **Q4:** (Lines 472-) "it (causality) arises naturally by solving (SCL')(M). As training advances, however, Alg. 1 shifts focus to fitting higher convection speeds $\beta$. Note that this occurs without any prior knowledge of the problems or manual tuning.": Could you clarify why causality naturally arises in this context?
>
> **Response:**
> We repeat here the full quote from the manuscript for context:
>
> >> by inspecting $\psi_0^\text{PDE}$ at different stages of training (Fig. 1b) it becomes clear that SCL begins by fitting the solution 'causally,' focusing first on smaller values of $t$. While doing so has been proposed to improve training (Krishnapriyan et al., 2021; Wang et al., 2024), it arises naturally by solving (SCL')(M). As training advances, however, Alg. 1 shifts focus to fitting higher convection speeds $\beta$. Note that this occurs without any prior knowledge of the problems or manual tuning.
>
> In the first part, we refer to "causality" in terms of "focusing first on smaller values of $t$." This is an approach that has been put forward in prior works to address PINN failures, such as (Krishnapriyan et al., 2021; Wang et al., 2024). We do not encourage this "causal" fitting of the solution in any way. However, we observe that, for certain BVPs (e.g., the convection PDE in Fig. 1), Alg. 1 tends to focus on earlier time instants in the beginning of training (i.e., $\psi_0$ puts more mass on smaller values of $t$, Fig. 1b). This is not enforced by the algorithm, but happens "naturally." This is an important fact given that there are cases where this notion of causality is not clear (e.g., the Helmholtz equation in Fig. 10).
>
> In fact, Alg. 1 adapts to the underlying needs of the problem without any modification. Indeed, note that $\psi_0$ is actually a joint distribution of collocation points and parameters $(x,t,\beta)$. Hence, while in later stages of training Alg. 1 does not distinguish between values of $t$ (it has a uniform marginal), it does tend to focus more on higher convection speeds (larger $\beta$). Once again, this is not a behavior we encourage based on prior knowledge, but something that arises "naturally" from the dynamic of Alg. 1.
>
> We hope that this makes our assertion clearer. We see now that these statements could be confusing as they are written and will reformulate them to better separate the "causality" arising from sampling $t$ from the changes in $\beta$.

---

> ### Author Response · Authors · 2024-11-21
>
> > **Q5:** Under what conditions, does Algorithm 1 converge? (Chamon et al., 2023; Elenter et al., 2024) could be valuable references, to my understanding.
>
> **Response:**
> Convergence of primal-dual methods such as Alg. 1 in non-convex settings is the subject of active research, see, e.g.,
>
> - Yang et al., "Global convergence and variance reduction for a class of nonconvex-nonconcave minimax problems," 2020
> - Lin et al., "Near-optimal algorithms for minimax optimization," 2020
> - Fiez et al., "Global convergence to local min-max equilibrium in classes of nonconvex zero-sum games," 2021
> - Boroun et al., "Accelerated primal-dual scheme for a class of stochastic nonconvex-concave saddle point problems," 2023
>
> As we explain in the paper (line 377), Alg. 1 is actually a variant of the dual ascent method shown in (3). For the latter method, (Elenter et al., 2024, Prop. 4.1) provides a convergence guarantee as long as the losses are strongly convex and smooth (satisfied by our quadratic losses) and the model capacity is large enough (i.e., we use large neural networks). Note that despite the convexity of the losses, the resulting problem is not convex due to the non-linearity of the models.
>
> In this setting, (Elenter et al., 2024, Prop. 4.1) guarantees much more than simply near-optimality of the solution [as is the case, e.g., of (Chamon et al., 2023, Thm. 2)], but also near-feasibility, side-stepping primal recovery issues found even in convex optimization [see, e.g., (Nedic & Ozdaglar, "Approximate primal solutions and rate analysis for dual subgradient methods," 2009.)]. Transfering these results to Alg. 1 requires an additional step size separation condition [see (Yang et al., 2020) mentioned above for specific results]. Though studying the convergence properties of Alg. 1 are beyond the scope of this paper, we will include a remark with the above discussion in the revised manuscript.

---

> > ### Comment · Reviewer_RnPx · 2024-11-22
> >
> > Thank you for the clarification and discussion, which corrected some of my misunderstandings. I have also gone through other reviews and author's responses, and I found that the paper would benefit from a major revision in my opinion.
> > Could you submit a revised paper? I would like to read it before sharing additional comments and replies; I am willing to change my score after reading it and rebuttal.
> >
> > Thank you.

---

> > > ### Author Response · Authors · 2024-11-25
> > >
> > > Following the reviewer suggestion, we have updated our manuscript (see main comment for a list of modifications, currently marked in blue in the paper). We hope they find that these changes make the paper clearer and address their concerns directly in the manuscript. But we are happy to provide further details and clarifications if necessary.

---

> > > > ### Comment · Reviewer_RnPx · 2024-12-01
> > > >
> > > > I sincerely appreciate the thorough revisions and discussions, and I am sorry for my late response. I have revisited the manuscript and could understand it more deeply, thanks to the improved readability.
> > > > I have changed my review scores accordingly: Soundness (1 to 2), Presentation (1 to 2), Contribution (1 to 2), Rating (3 to 5).
> > > > I have also changed Confidence from 4 to 3.
> > > >
> > > > I would like to share my additional feedback. I understand it may be too late to share, but I hope it helps.
> > > >
> > > > - Error bars: Thank you for the additional experiments. I encourage the authors to add error bars to all the remaining experiments because I would like to know whether performance gain is marginal or not and *when we should use the proposed method instead of previous ones*, a practically important point of view. I understand the main contribution of the paper is the theoretical part, but this point will underscores the practical relevance of the proposed theoretical structure and algorithm.
> > > >
> > > > - (Lines 161-163) "...they are the wrong problems in the first place.": It would sound like an exaggeration to say that the previous approaches (PI) and (PII) are "wrong": It is difficult to prove these approaches are wrong ways to go, after all. Readers may be a bit confused, like an "impossibility theorem" has been proved in the paper. Instead, to avoid potential confusion and for consistency, I would recommend modifying the logic of the contributions like "Our novel approach alleviates the requirements of ...hyperparameter tuning, etc..., compared to the previous methods, as a result of ...(key ideas of the proposed method)....", which I think sounds more natural and is a standard logic we often find in the literature, avoiding potential overstatements and clarifying the key ideas.
> > > >   - (Why I recommend this) I was a bit confused when I read around Lines 146-170: Did the authors address the known problems listed in the paragraph "Limitations." or propose a completely new paradigm indicated in the beginning of Section 3?
> > > >
> > > > - (Lines 207-210) "While Prop. 3.1 describes a sufficient condition ... a necessary condition can be obtained ...": I would recommend elaborating on this statement (possibly proving it), for clarity.
> > > >
> > > > - How to choose $\alpha$? Proposition 4.1 is an existence theorem and does not provide an exact $\alpha$, to my understanding.
> > > >   - (Lines252-355) "It is often possible to replace ...": I would recommend adding evidences for this empirical statement.
> > > >
> > > > - (Lines 323-328): I appreciate that the differences between the proposed Lagrangean formulation (DIV) and previous approaches in (PI) are additionally highlighted in the revised version, improving the clarity of the paper.
> > > >
> > > > - (Minor) In Line 163, a space is missing in "...worst-case losses.Hence, is not..."
> > > >
> > > > - (Minor) In Line 525, a space is missing in "is (virtually) independent of the choice of$u_\theta$."
> > > >
> > > > I wish you the best of luck with your work.

---

> > > > > ### Author Response · Authors · 2024-12-02
> > > > >
> > > > > We thank the reviewer for revisiting their assessment and for the additional feedback. We address their latest comments below in the hope that they will feel more positive about the paper after these clarifications. We also thank them for spotting our typos: we will proof-read the full manuscript again after we have finishing our camera-ready edits.
> > > > >
> > > > > - **Error bars:** We completely agree. As we mentioned in our response, we continue to run experiments to include error bars on all of our results. We provided only preliminary results due to time and compute limitations. Though our contributions are indeed mainly theoretical, we believe that they may also have an impact in practice (as corroborated by our experiments).
> > > > >
> > > > > - **(Lines 161-163) "...they are the wrong problems in the first place.":** We completely agree with the reviewer and we in no way believe (or claim) that previous approaches are "wrong." As we state in the manuscript, (PI) and (PII) can and have successfully found solutions to many PDEs (see both our experiments in Sec. 5 and related work in App. G) and have been "effective in many applications" (line 147).
> > > > >
> > > > >   Our statement is that they target the "wrong problem" because "regardless of how the weights $\mu$ in (PI) are adapted [...], it *need not* provide a solution of (SCL)" (line 289, our emphasis). In other words, (PI) is an optimization problem that is *not guaranteed* provide a (weak) solution of (BVP). In contrast, "by (approximately) solving (PIII) [...], we indeed (approximately) solve (BVP)" (line 204). Indeed, there are no impossibility results for (PI), but neither are there guarantees [in contrast to (PIII)].
> > > > >
> > > > >   We will clarify those statements to ensure we do not suggest claims we do not prove.
> > > > >
> > > > >   - **Lines 146-170:** We address the known limitations of current approaches by proving (in the now Prop. 3.1) that the optimization problem that solves (BVP) is (PIII). We would certainly not say this is a "new paradigm," it is well within the general ML approach. What we argue is that overcoming the issues of previous methods requires revisiting which ML problem is being solved, i.e., that "the challenges faced by previous NN-based BVP solvers arise not because of *how* (PI) and (PII) are solved," but because to solve (BVP) we actually need to solve (PIII).
> > > > >
> > > > > - **(Lines 207-210):** We will expand on that remark noting that for $\psi \in W^{k^\prime,2}(\mathcal{D})$ (and under the smoothness restrictions stated in the theorem) the worst-case loss essentially recovers (1).
> > > > >
> > > > > - **How to choose $\alpha$?** In Sec. 4, we explain that we take $\alpha=0$ in Alg. 1 because "the $\psi_\alpha$ are smooth, fully-supported, square-integrable distributions for $\alpha = 0$. They are therefore amenable to be sampled using [...] MCMC" (in our case, the MH algorithm). We find this approximation (guaranteed by Prop. 4.1) to be good enough in our experiments, so that the use of "algorithms adapted to discontinuous distributions (e.g., (Nishimura et al., 2020)) to enable [...] better approximations (increase $\alpha$) is left for future work" (lines 344-351).
> > > > >
> > > > >     - **(Lines 252-355)** The reviewer has a point. We are missing a reference to our own experiments (Sec. 5). As we explain in App. E, we replace $\psi_\alpha$ by the uniform distribution for the BC objective in all of our results.
> > > > >
> > > > > - **(Lines 323-328):** We are glad that our response and revisions made the distinction between the empirical dual problem (and the primal-dual method in Alg. 1) and its distinction to previous approaches (PI) clearer. This is one of the fundamental point of our work, so we appreciate the reviewer's feedback in improving its presentation.

---

### Official Review · Reviewer_4CoR · 2024-10-25

**Soundness:** 3
**Presentation:** 3
**Contribution:** 2
**Rating:** 5
**Confidence:** 3

**Summary:**

Neural network-based approaches, such as physics-informed neural networks and neural operators, offer a mesh-free alternative by directly fitting those models to the PDE solution. They are known to be highly sensitive to hyperparameters such as collocation points and the weights associated with each loss. This paper addresses these challenges by developing a science-constrained learning (SCL) framework. It demonstrates that finding a (weak) solution of a PDE is equivalent to solving a constrained learning problem
with worst-case losses.

**Strengths:**

The strength of the paper is their proposed solution to the hyperparameter issue:  They develop a science-constrained learning (SCL) framework. It demonstrates that finding a (weak) solution of a PDE is equivalent to solving a constrained learning problem
with worst-case losses.

The paper provides ways of incorporating structural knowledge, observational knowledge, and science-contrained learning.  The algorithm is easy to follow, and the mathematical proofs are helpful in terms of understanding the benefits of the method.

**Weaknesses:**

The reviewer is having a difficult time distinguishing the contribution from the work of Paris Perdikaris and his group.  Examples are:

A. Daw, J. Bu, S. Wang, P. Perdikaris, A. Karpatne, Rethinking the Importance of Sampling in Physics-informed Neural
Networks, arXiv preprint arXiv:2207.02338

 S. Wang, S. Sankaran, P. Perdikaris, Respecting causality is all you need for training physics-informed neural networks,
arXiv preprint arXiv:2203.07404

Paris references in one of his talks a paper by this former advisor and colleagues "A unified scalable framework for causal sweeping strategies for ..." that has a categorization (and references).

The important point is that the algorithm in the current paper (Algorithm 1) has many of the same characteristics you find in Paris' work which you incorporate learning the hyperparameters into the learning.

At some point, the reviewer is wondering how common this is now considered when application papers now reference the constrained paper (NIPS) as say that they use PINNs + Constrained learning:

https://www.sciencedirect.com/science/article/pii/S1385894724012919#b78

Is the current paper really 'novel' (i.e. a contribution) if others now consider things commonplace.  The reviewer is open to being convinced of the contributions if the literature search were broader and the comparisons where against the various adaptive sampling methods mentioned by Paris and others.

In terms of concrete actions, can the authors:
+  Provide a more comprehensive literature review that clearly positions their work relative to existing constrained learning approaches for PINNs?

+  Explicitly discuss how their method differs from or improves upon other adaptive sampling techniques, particularly those by Perdikaris and others (as highlighted by the question below)?

+ Highlight any novel theoretical results or empirical improvements over state-of-the-art methods

**Questions:**

The specific questions (associated with the items above) are:

+ Can the authors clarify how their approach (for instance Algorithm 1 updating schedule) compares to the update schedule in Paris' paper:   https://epubs.siam.org/doi/pdf/10.1137/20M1318043    Section 2.4 (not the specific updates, but the general updates as given by the spirit of the logic presented)?  In others of Paris' papers, he points to :   https://arxiv.org/pdf/2007.04542.  How does the new method compare to the "adding weights to the loss function" section 2.2.1 (Which is analogous to Paris' work also)?

**Details Of Ethics Concerns:**

No ethics issues

---

> ### Author Response · Authors · 2024-11-21
>
> > **W1:** The reviewer is having a difficult time distinguishing the contribution from the work of Paris Perdikaris and his group. Examples are:*
> >
> > A. Daw, J. Bu, S. Wang, P. Perdikaris, A. Karpatne, Rethinking the Importance of Sampling in Physics-informed Neural Networks, arXiv preprint arXiv:2207.02338
> > S. Wang, S. Sankaran, P. Perdikaris, Respecting causality is all you need for training physics-informed neural networks, arXiv preprint arXiv:2203.07404
> >
> > Paris references in one of his talks a paper by this former advisor and colleagues "A unified scalable framework for causal sweeping strategies for ..." that has a categorization (and references).
>
>
> **Response:**
> The main contributions of this work (summarized in lines 54-65) are
>
> - prove that obtaining a (weak) solution of (BVP) is equivalent to solving a constrained learning problem with worst-case losses, namely (PIII) (Prop. 3.2);
> - incorporate other scientific prior knowledge to (PIII), such as structural (e.g., invariance) and observational (e.g., measurements, known solutions) information,  without resorting to specialized models or data transforms (SCL, Sec. 3.2);
> - develop a practical hybrid sampling-optimization algorithm (Alg. 1) that requires neither tuning weights to balance different objectives nor the careful selection of collocation points. Alg. 1 also yield trustworthiness measures by capturing the difficulty of fitting PDE instances or data points (Sec. 4);
> - illustrate the effectiveness (accuracy and computational cost) of this method in a diverse set of PDEs, NN architectures (e.g., MLPs and NOs), and problem types (solving a single or a parametric family of PDEs; interpolating known solutions; identifying PDE instances or data points that are difficult to fit, Sec. 5).
>
> As suggested by x438, we will include this explicit list in the introduction.
>
> Prop. 3.2 in particular is key as it shows that the limitations of previous approaches are not methodological, but epistemological. It is not an issue of *how* they solve the problem, but *which* problem they solve. Indeed, Prop. 3.2 shows that it is not enough to use either worst-case losses as in (Wang et al., 2022a; Daw et al., 2023) or adapting the loss weights as in (Wang et al., 2021a; Wang et al., 2022b; Maddu et al., 2022; McClenny & Braga-Neto, 2023), even using constrained formulations (Lu et al., 2021b; Basir & Senocak, 2022).
>
> These are not purely theoretical issues: using (PI) with R3 to approximate worst-case losses fails to recover a solution for the Eikonal equation (Table 1) and using a constrained formulation with fixed collocation points fails to recover a solution for the convection PDE (Fig. 2). Both are needed simultaneously, which is why Alg. 1 succeeds in both cases (see Table 1).
>
> (continued in the next comment)

---

> ### Author Response · Authors · 2024-11-21
>
> > **W1 (continued)**
>
> With respect to the specific papers mentioned by the reviewer, they are (Daw et al., 2023) and (Wang et al., 2024) in our related work (Appendix F). Explicitly:
>
> - **(Daw et al., arXiv:2207.02338)** is a preliminary version of (Daw et al., 2023) that proposes *R3* which we use as a benchmark in our experiments (Table 1). R3 uses a sort of rejection sampling approach to approximate the worst-case loss for PDE residuals. The first important distinction is that, as we prove in Prop. 3.2, this is not enough to yield a weak solution of BVPs (see eikonal in Table 1). Doing so requires solving a *constrained* problem involving worst-case losses, namely (SCL). Alg. 1 does that by leveraging MCMC techniques (namely, Metropolis-Hastings) and duality. Aside from Metropolis-Hastings mixing faster than rejection sampling (Robert & Casella, 2004), our approach can accommodate other modern sampling technique (e.g., Langevin Monte Carlo). Indeed, we prove in Prop. 3.1-3.2 that the $\psi_0$ in Alg. 1 (and more generally, $\psi_\alpha$) are square-integrable, i.e., belong to $\mathcal{P}^2$.
>
>     In our comparisons to R3 (Sec. 5.1), we obtain very similar results in general, except when R3 fails to recover a solution (see eikonal in Table 1). Our method continues to provide good results in those cases, aside from being adapted to other settings, such as solving parametric BVPs (Sec. 5.2) and interpolating existing solutions (Sec. 5.4).
>
> - **(Wang et al., arXiv:2203.07404)** is a preliminary version of (Wang et al., 2024) which separately weights the PDE residuals within different time windows. The weights are adjusted to respect some form of temporal causality during training, an idea that has also been explored in, e.g., (Krishnapriyan et al., 2021). We note that we do not **need** to account for such form of causality: the fact that (SCL')(M) tends to focus on fitting the solution at earlier times first (e.g., Fig. 1b) is a consequence of the training dynamics of Alg. 1. What is more, by sampling from $\psi_0$ Alg. 1 adapts to the model fit in both time and space. This is particularly relevant for BVPs that do not have a clear notion of causality (e.g., Helmholtz equation in Fig. 10).
>
> - **"A unified scalable framework for causal sweeping strategies for ..."**: we assume the reviewer refers to
>
>     [Penwarden et al., "A unified scalable framework for causal sweeping strategies for Physics-Informed Neural Networks (PINNs) and their temporal decompositions," Journal of Computational Physics, 2023]
>
>     If this is not the case, please let us know. Once again, the distinction is that we do not enforce any form of causality during training nor do we need to. The fact that (SCL)(M) first focuses on fitting the solution at earlier times (as in Fig. 1b) is a natural behavior of Alg. 1 and not one that we encourage in any way. As we mention above, this is particularly important for BVPs that lack a clear notion of causality (e.g., the Helmholtz equation) or when solving parametrized families of BVPs (it is not clear which parameter value should we fit first, as in Fig. 6). Alg. 1 adapts to the underlying needs of the problem without any modification.
>
>
> We appreciate the reviewer bringing (Penwarden et al., 2023) to our attention, especially since the architectures based on domain decomposition it reviews can be used as $u_\theta$ in (SCL) to tackle problems with discontinuous solutions or improve computational complexity. We will include it and expand our remarks on the other works in our literature review (Appendix F).

---

> ### Author Response · Authors · 2024-11-21
>
> > **W3:** At some point, the reviewer is wondering how common this is now considered when application papers now reference the constrained paper (NIPS) as say that they use PINNs + Constrained learning:
> >
> > https://www.sciencedirect.com/science/article/pii/S1385894724012919#b78
> >
> > Is the current paper really 'novel' (i.e. a contribution) if others now consider things commonplace. The reviewer is open to being convinced of the contributions if the literature search were broader and the comparisons were against the various adaptive sampling methods mentioned by Paris and others.
>
> **Response:**
> We assume that by "constrained paper (NIPS)" the reviewer is referring to (Chamon & Ribeiro, 2020).
>
> We note that the paper referred to by the reviewer does not use constrained optimization techniques, but modifies the underlying architecture to satisfy the PDE and BCs. As such, it is a completely different approach than the one from the current work. This approach also makes it difficult to incorporate additional prior knowledge that (SCL) explicitly considers, such as structural (e.g., invariances) and observational (e.g., prior solutions and measurements) information.
>
> While we do use constrained learning as a tool in this work, it is not straightforward that (BVP), a functional, feasibility problem, can be written as (SCL), a finite dimensional, statistical, constrained optimization problem. This is only possible once we deploy Propositions 3.1 and 3.2. The latter, in fact, shows that there are limitations to this transformation in the sense that learning approaches can only determine fairly smooth solutions for high-dimensional state spaces (essentially, solutions in the Sobolev space $W^{(d+1) / 4,2}$). This can be an issue for large-scale dynamical systems, such as those found in smart grid applications, or when transforming higher-order PDEs in higher-dimensional first-order systems.

---

> ### Author Response · Authors · 2024-11-21
>
> > **W4:** Provide a more comprehensive literature review that clearly positions their work relative to existing constrained learning approaches for PINNs?
>
> **Response:**
> We refer the reviewer to our related work section (Appendix F), where we review the literature in general. More specifically, we compare there and throughout our paper to (Lu et al., 2021b; Basir & Senocak, 2022) which are the two references we found that explicitly use constrained formulations. However, as we have argued in our response to W3, they do not show that their Lagrangian-based algorithm actually solves the constrained problem of interest given that it is non-convex. Hence, (Lu et al., 2021b; Basir & Senocak, 2022) provide solutions to dual problems of the form ($\hat{\text{D}}$-CSL) in Sec. 3.1 and not to their constrained counterparts. We overcome this challenge by using the *constrained learning* formulation (SCL), i.e., a statistical rather than deterministic optimization problem (see more details in the response to W3).
>
> Note also that Alg. 1 is different from all the papers mentioned so far in that is *also* uses statistical formulations of worst-case losses (namely, the $\psi_0$). This is not a small distinction as it is *necessary* to obtain (weak) solutions of BVPs (as we prove in Prop. 3.2). In fact, Prop. 3.2 shows that neither the constrained formulations of (Lu et al., 2021b; Basir & Senocak, 2022) nor the adaptive loss weighting schemes of (Wang et al., 2021a; Wang et al., 2022b; Maddu et al., 2022; McClenny & Braga-Neto, 2023) are suuficient to solve (BVP). We refer the reviewer to W3 for a detailed discussion and failure examples.
>
> We will include update Appendix F to include this more detailed discussions as well as the two additional references mentioned by the reviewer, namely (Penwarden et al., 2023) and (Rehman & Lienhard, 2024). Do not hesitate to let us know if we left out any additional important work.

---

> ### Author Response · Authors · 2024-11-21
>
> > **W5:** Explicitly discuss how their method differs from or improves upon other adaptive sampling techniques, particularly those by Perdikaris and others (as highlighted by the question below)?
>
> **Response:**
> As we noted before, Alg. 1 approximates worst-case losses (such as the PDE residuals) by using MCMC techniques (namely, Metropolis-Hastings). This is justified by Prop. 3.1 which proves that sampling from $\psi_0$ is an explicit approximation of the worst-case loss. Such an approximation guarantee does not exist for any of the adaptive sampling techniques mentioned by the reviewer or in our literature review (see Appendix F). Prop. 3.2, however, proves that adaptive sampling techniques are not enough to obtain weak solutions of (BVP). Indeed, they must be combined with a constrained formulation as in (SCL). The adaptive sampling papers mentioned by the reviewer lack such guarantees as they focus on weighted loss formulations of the form (PI).
>
> Observe that we directly compare to R3 from (Daw et al., 2023), where it was shown to outperform many of the adaptive sampling techniques mentioned by the reviewer and in the paper (Sec. 5.1). As we noted in W1, our results are generally similar to R3, except when it fails (see Eikonal equation in Table 1). Our method provides good results even in those cases, aside from being adapted to other settings, such as solving parametric BVPs (Sec. 5.2) and interpolating existing solutions (Sec. 5.4).

---

> ### Author Response · Authors · 2024-11-21
>
> > **W6:** Highlight any novel theoretical results or empirical improvements over state-of-the-art methods
>
> **Response:**
> We repeat here our list of contributions (as summarized in lines 54-65):
>
> - prove that obtaining a (weak) solution of (BVP) is equivalent to solving a constrained learning problem with worst-case losses, namely (PIII) (Prop. 3.2);
> - incorporate other scientific prior knowledge to (PIII), such as structural (e.g., invariance) and observational (e.g., measurements, known solutions) information,  without resorting to specialized models or data transforms (SCL, Sec. 3.2);
> - develop a practical hybrid sampling-optimization algorithm (Alg. 1) that requires neither tuning weights to balance different objectives nor the careful selection of collocation points. Alg. 1 also yield trustworthiness measures by capturing the difficulty of fitting PDE instances or data points (Sec. 4);
> - illustrate the effectiveness (accuracy and computational cost) of this method in a diverse set of PDEs, NN architectures (e.g., MLPs and NOs), and problem types (solving a single or a parametric family of PDEs; interpolating known solutions; identifying PDE instances or data points that are difficult to fit, Sec. 5).
>
> As we have elaborated our responses so far, none of the previous works solve a constrained learning problem with worst-case losses, namely (PIII). As we prove in Prop. 3.2, this is required to obtain weak solutions of BVPs. Hence, in contrast to other approaches, Alg. 1 does (approximately) solve (BVP) (see response to W2). What is more, Alg. 1 can seemlessly tackle both unsupervised (as in most papers mentioned by the reviewer) and supervised problems (Sec. 5.4).
>
> Our empirical results show that we either match or outperform traditional PINN baselines and R3 (Daw et al., 2023). Even in cases where they fail (eikonal equation in Table 1), our method finds a good solutions (without modifications, see Appendix D). Alg. 1 can also simultaneously solve entire families of PDEs (Sec. 5.2) and incorporate both structural knowledge (Sec. 5.3) and data (Sec. 5.4). In the latter case, it also provides measure of trustworthiness for the solution by pointing out data points that are hard to fit (by means of the dual variables $\lambda$, e.g., Fig. 3).

---

> ### Author Response · Authors · 2024-11-21
>
> > **Q1:** Can the authors clarify how their approach (for instance Algorithm 1 updating schedule) compares to the update schedule in Paris' paper:  https://epubs.siam.org/doi/pdf/10.1137/20M1318043 Section 2.4 (not the specific updates, but the general updates as given by the spirit of the logic presented)? In others of Paris' papers, he points to :  https://arxiv.org/pdf/2007.04542. How does the new method compare to the "adding weights to the loss function" section 2.2.1 (Which is analogous to Paris' work also)?
>
> **Response:**
> The references mentioned by the reviewer are (Wang et al., 2021a) and (Wight & Zhao, 2021) in our manuscript respectively and we compare our work to them in Appendix F. In particular, we recall that Prop. 3.2 shows that neither adaptive sampling techniques nor loss weighting schemes are sufficient to solve BVPs. In that sense, both references fall short of solving BVPs. Specifically,
>
> - **https://epubs.siam.org/doi/pdf/10.1137/20M1318043, i.e., (Wang et al., 2021a) in our manuscript.** In Sec. 2.4 and 2.5, it proposes to set the weights of the different loss terms [$\mu$ in (PI)] so that the gradients of the different terms have similar magnitudes. In that sense, (Wang et al., 2021a, eq. 40-41) are reminiscent of Adamax (Kingma & Ba, 2015). This is fundamentally different from Alg. 1. While (Wang et al., 2021a, eq. 35) may have the same form as the Lagrangian in (2), they are used in different ways. The dual problem ($\hat{\text{D}}$-CSL), that Alg. 1 targets, does not seek to balance gradient updates. In fact, the resulting contribution of **the losses may very much need to be unbalanced** (see, e.g., Fig. 16a in Appendix E which shows certain dual variables $\lambda$ vanishing completely). The dual problem that Alg. 1 targets actually approximates the solution of the constrained problem (SCL) (as we explain in W4 above). This is essential, together with sampling from $\psi_0$, to obtain a solution of (BVP) (as we prove in Prop. 3.2).
>
> - **https://arxiv.org/pdf/2007.04542, i.e., (Wight & Zhao, 2021) in our manuscript.** In Sec. 2.2.1, it proposes to add a **fixed weight** to the "data loss," i.e., to reweight the loss relative to initial training data with respect to the losses relative to the PDE and BCs. This is not sufficient to obtain a solution of the constrained problem (SCL) (due to its non-convexity, see response to W2) that must be solved in order to obtain a weak solution of (BVP) (see Prop. 3.2). What is more, (SCL)(O) and Alg. 1 weight each data point individually to optimize the worst-case (rather than the average) fit. This leads to measures of trustworthiness for the solution as it singles out the data points that are hard to fit (by means of the dual variables $\lambda$, e.g., Fig. 3).
>
> In summary, "adding weights to the loss function" is not sufficient to solve (BVP), especially if the weights are fixed or attempting to balance the contributions of the gradients. While we extensively cover the theoretical reasons for this here (see responses to W1, W2, and W4), we also provide substantial empirical evidence that this theoretical foundation does manifest in practice (such as Table 1, Fig 2, and Fig. 16a).

---

> ### Author Response · Authors · 2024-11-25
>
> Following the reviewers comments, we have updated our manuscript (see main comment for a list of modifications, currently marked in blue in the paper). We hope the reviewer finds these minor additions and changes make the paper clearer and address their concerns directly in the manuscript. We are happy to provide further details and clarifications if necessary.

---

### Official Review · Reviewer_x438 · 2024-10-30

**Soundness:** 3
**Presentation:** 3
**Contribution:** 3
**Rating:** 8
**Confidence:** 3

**Summary:**

The paper develops a technique called ``Science constrained learning'' in order to address some of the issues with Physics Informed Neural Networks (PINNs) by allowing structural constraints and known solutions to be taken into account in addition to the usual constraints considered by PINNs.

**Strengths:**

Overall the paper is well written and formatted and the development of Science Constrained Learning (SCL) is well described. There are several different background techniques described in the paper, in addition to the new technique. I think this work is worthy of inclusion in the conference and I am leaning towards an acceptance.

**Weaknesses:**

There are several different background techniques described in the paper, in addition to the new technique. Therefore, I think it would be beneficial to the readability of the paper to include a clear list of the scientific contributions of SCL to help the reader. I note that the paper is a full 10 pages long, however I think that some of the background could be shortened somewhat (especially in the introductory sections) in order to give the description of the new technique a bit more room to breathe.

One of the main claims for the new method is that it is not sensitive to the balancing of weights for the different components of the loss function. This is well justified by the use of the dual formulation. However this is not mentioned until the end of the 4th page. Due to the importance of this aspect of the solution, I would recommend mentioning it earlier in the paper. Otherwise the reader is left wondering what the difference is to e.g.\ the work in [1], which uses derivative constraints and empirical observations.

The use of the Lagrangian does indeed support the idea that the method addresses the issue of balancing the weights in the loss function which can affect PINNs. However the paper also makes the statement that this new methodology addresses sensitivity to the number of training points. I cannot really see a clear explanation in the paper for why this might be the case. Either this explanation should be brought out more and clarified in the paper or, alternatively, it should be explained that this is an empirical observation.

[1] Kentaro Hoshisashi, Carolyn E Phelan, and Paolo Barucca. No-arbitrage deep calibration for volatility smile and skewness. arXiv preprint arXiv:2310.16703, 2023.

**Questions:**

I would be very pleased to hear the authors' response to my comments above, especially with respect to the proof of the reduction in conditioning number in Appendix B.

---

> ### Author Response · Authors · 2024-11-21
>
> > **W1:** There are several different background techniques described in the paper, in addition to the new technique. Therefore, I think it would be beneficial to the readability of the paper to include a clear list of the scientific contributions of SCL to help the reader.
>
> **Response:**
> We thank the reviewer for their positive assessment of our work. The contributions of this paper are summarized in the last paragraph of the introduction (lines 54-65), but we list them here more explicitly:
>
> - prove that obtaining a (weak) solution of (BVP) is equivalent to solving a constrained learning problem with worst-case losses, namely (PIII) (Prop. 3.2);
> - incorporate other scientific prior knowledge to (PIII), such as structural (e.g., invariance) and observational (e.g., measurements, known solutions) information,  without resorting to specialized models or data transforms (SCL, Sec. 3.2);
> - develop a practical hybrid sampling-optimization algorithm (Alg. 1) that requires neither tuning weights to balance different objectives nor the careful selection of collocation points. Alg. 1 also yield trustworthiness measures by capturing the difficulty of fitting PDE instances or data points (Sec. 4);
> - illustrate the effectiveness (accuracy and computational cost) of this method in a diverse set of PDEs, NN architectures (e.g., MLPs and NOs), and problem types (solving a single or a parametric family of PDEs; interpolating known solutions; identifying PDE instances or data points that are difficult to fit, Sec. 5).
>
> We point out that the result in Prop. 3.2 is key as it shows that it is not enough to use either worst-case losses as in (Wang et al., 2022a) or constrained formulations as in (Lu et al., 2021b; Basir & Senocak, 2022). Both are needed to find solutions of BVPs. Indeed, note that using a constrained formulation with fixed collocation points does not recover a solution for a simple PDE (Fig. 2), while Alg. 1 does (Table 1).
>
> What is more, Prop. 3.2 establishes the limitations of learning approaches that can only determine very smooth solutions for high-dimensional state spaces (essentially, solutions in the Sobolev space $W^{(d+1) / 4, 2}$). This can be an issue for large-scale dynamical systems, such as those found in smart grid applications, or when transforming higher-order PDEs in higher-dimensional first-order systems.

---

> ### Author Response · Authors · 2024-11-21
>
> > **W2:** I note that the paper is a full 10 pages long, however I think that some of the background could be shortened somewhat (especially in the introductory sections) in order to give the description of the new technique a bit more room to breathe.
>
> **Response:**
> We will include the above explicit list in the camera-ready following the reviewer's recommendations of shortening the background parts (moving details to the appendix).

---

> ### Author Response · Authors · 2024-11-21
>
> > **W3:** One of the main claims for the new method is that it is not sensitive to the balancing of weights for the different components of the loss function. This is well justified by the use of the dual formulation. However this is not mentioned until the end of the 4th page. Due to the importance of this aspect of the solution, I would recommend mentioning it earlier in the paper.
>
> **Response:**
> The reviewer is correct that this is one of the main benefits of our constrained learning approach. Using duality is fundamentally different from existing loss balancing methods that rely on various heuristics (listed in our literature review in Appendix F). We hope that the addition of the explicit list of contributions above will make this clearer to the reader from the beginning. We will also mention this fact more explicitly in the abstract.

---

> ### Author Response · Authors · 2024-11-21
>
> > **W4:** Otherwise the reader is left wondering what the difference is to e.g. the work in [1], which uses derivative constraints and empirical observations.
>
> **Response:**
> The problem that Alg. 1 is tackles is substantially different from that in [1]. First, [1] controls the average value of derivatives on an empirical dataset (see [1, eq. 9]). In contrast, (BVP) [and (SCL)] require that the differential equation hold for all points in the domain. Second, [1] uses fixed weights to incorporate the derivative constraints as penalties in the loss. In that sense, it is similar to (PI) and (Wang et al., 2021a; Wang et al., 2022b; Maddu et al., 2022; McClenny & Braga-Neto, 2023). This, however, is not enough to solve (SCL) due to its non-convexity (see duality discussion in, e.g., response W2 to Reviewer 4CoR).
>
> We address the first point by relying on Prop. 3.1, which turns a worst-case loss into a statistcal loss with respect to a specially crafted distribution ($\psi_\alpha$). We can then write the constrained *learning* problem (SCL) using statistical rather than deterministic losses, collocation points. This allows us to handle the second point by applying the non-convex duality results from (Chamon & Ribeiro, 2020; Chamon et al., 2023) to provide approximation guarantees for the solution of its dual problem, of the form ($\hat{\text{D}}$-CSL). Combined with results from (Cotter et al., 2019; Elenter et al., 2024), this yields convergence guarantees for (3), a variant of Alg. 1 (as we explain in line 377) towards a (probably approximately) near-feasible and near-optimal solution of (SCL). By Prop. 3.2, this provides a (weak) solution of (BVP).
>
> We hope that this parallel with [1] clarifies our differences in terms of problem and methodologies. We agree with the reviewer that these clarifications should appear earlier in the paper and expect that modifying the contributions and abstract (as well as reducing the introductory sections) will help with this goal.

---

> ### Author Response · Authors · 2024-11-21
>
> > **W5:** The use of the Lagrangian does indeed support the idea that the method addresses the issue of balancing the weights in the loss function which can affect PINNs. However the paper also makes the statement that this new methodology addresses sensitivity to the number of training points. I cannot really see a clear explanation in the paper for why this might be the case. Either this explanation should be brought out more and clarified in the paper or, alternatively, it should be explained that this is an empirical observation.
>
> **Response:**
> We assume that by "sensitivity to the number of training points" the reviewer refers to our claim that our method is not as sensitive as PINNs to the choice of collocation points (please do not hesitate to correct us if that is not the case).
>
> That the choice of collocation points can greatly impact the quality of the solution of PINNs has been observed and investigated before (Nabian et al., 2021; Daw et al., 2023; Wang et al., 2024). We illustrate this in Fig. 2, where even constrained methods are unable to solve a convection BVP when using *fixed* collocation points. This hyperparameter is completely removed in (SCL), since evaluation points are chosen in Alg. 1 according to distributions $\psi_0$ that are adapted both to the PDE and the model $u_{\theta}$. However, this is not just a convenience: Prop. 3.2 clearly shows that only by combining constrained formulations with worst-case losses as in (SCL) can we solve (BVP).
>
> Fig. 2 also illustrates how (SCL) can be made less sensitive to collocation points by incorporating scientific knowledge. Indeed, when using a small number of *fixed collocation points* neither (PI) (which uses hand-tuned weights to combine the losses) nor (SCL)(M) (which uses a constrained formulation) are able to find a solution of the BVP. Yet, when incorporating structural information (in this case, invariance), an accurate solution is obtained by (SCL)(M+S).
>
> This insensitivity also appears when solving for parametric families of PDEs. Note from Fig. 1 that the finer the discretization of the parameter range, the more collocation points are used by (PI) (as the legend suggests, (PI) uses 1000 collocation points per parameter). Clearly, automatically selecting collocation points can bring performance advantages (see, e.g., Table 7). Once again, this is not only convenient, but also necessary to provide (weak) solutions of BVPs (Prop. 3.2).
>
> We have included this point in the contribution list of the introduction and will more explicitly point to evidence that support it (both methodological, i.e., Alg. 1, and empirical, such as Figs. 1 and 2) wherever it appears.

---

> ### Author Response · Authors · 2024-11-21
>
> > **Q1:** I would be very pleased to hear the authors' response to my comments above, especially with respect to the proof of the reduction in conditioning number in Appendix B.
>
> **Response:**
> We hope that we have addressed the reviewer's concerns and questions above. We did not address the comment on the "proof in Appendix B" since we did not find any mention of Appendix B in the "Weaknesses." We would be happy to elaborate on the proof if the reviewer has any concerns.

---

> ### Author Response · Authors · 2024-11-25
>
> Following the reviewers comments, we have updated our manuscript (see main comment for a list of modifications, currently marked in blue in the paper). We hope the reviewer finds these minor additions and changes make the paper clearer and address their concerns directly in the manuscript. We are happy to provide further details and clarifications if necessary.

---

### Official Review · Reviewer_JwZx · 2024-10-31

**Soundness:** 2
**Presentation:** 4
**Contribution:** 2
**Rating:** 3
**Confidence:** 4

**Summary:**

This paper aims to adapt ideas from adversarial robustness and use them to find neural network solutions to partial differential equations with improved guarantees over existing neural solutions to PDEs.  The scheme, as I understand it, is to observe that the weak form of the solution to a PDE can be achieved by finding the minimum of a kind of "worst-case PINN loss" in which the distribution over collocation points is chosen to maximize the expected PINN loss.  This framing allows a particular result from adversarial robustness to apply: the worst-case (wrt distributions on inputs) expected loss on a region can be approximated by an expectation under a distribution proportional to the shifted-and-truncated loss on that region.  The amount of shifting governs the accuracy of the approximation.  I understand this to be a kind of continuity argument: if the worst-case distribution is concentrated on the points with the highest loss, then smoothing those out a bit doesn't change the value all that much.

The result is that you can get a reframing of the weak problem to look sort of like a PINN with a particular distribution on the collocation points.  The paper discusses how various kinds of constraints, e.g., symmetries, can then be framed in this way.  The actual sampling itself of the \psi_\alpha distribution is done with Metropolis-Hastings, or it is discarded in favor of a uniform distribution.  The approach is empirically evaluated on a couple of small problems and compared to some neural network baselines. No comparisons to conventional PDE solvers are performed.

**Strengths:**

I think the main strength of this paper is that it allows one to reason more formally about the PINN-type setup and connect it directly to the weak form solution.  It's a creative idea to adapt the result from adversarial robustness. The paper is mostly well written.

**Weaknesses:**

The main weakness of the paper is that we don't get any sense of how well this actually works for solving PDEs.  Like many "deep learning for PDEs" papers, it does not compare to conventional methods for solving PDEs, but only other neural networks.  It is currently unclear whether PINNs and related ideas are actually ever a good idea.  Very few papers examine the actual Pareto curve of computational cost versus accuracy with respect to, e.g., FEM.  A recent paper reviewing the area has shed light on just how bad this situation is from a scientific perspective:

McGreivy, Nick, and Ammar Hakim. "Weak baselines and reporting biases lead to overoptimism in machine learning for fluid-related partial differential equations." Nature Machine Intelligence (2024): 1-14.

The MH24 paper criticizes weak standard numerical baselines, but the present paper does not compare to any conventional numerical methods at all.  Do we conclude from this paper that I should stop using my PDE solver and start using this method?  The abstract claims "accurate solutions" but then only demonstrates that relative to deep learning baselines on tiny problems.

And to be clear, this situation is not really the authors' fault.  They are just following the trend and applying the standard that the neural PDE solver community has created.  Nevertheless, the for the larger scientific community to find this work valuable, we must apply a standard that makes sense for people who want to actually solve PDEs.

So we might instead take the contribution of the paper to be the guarantee, except:

1) As stated in the paper, existing methods already have guarantees.

2) The guarantee itself centers on sampling from \psi_\alpha which, as mentioned in the paper, becomes difficult exactly when the error bound becomes tight.  Using Metropolis--Hastings essentially throws that guarantee away unless you can prove, e.g., uniform ergodicity of the resulting Markov chain. So in the end we are faced with what looks like just another difficult integral, just with a different form and a lack of a guarantee.  Note that one could raise the same criticism of the original adversarial robustness result: sampling from little regions around the worst case inputs is just as difficult as finding them in the first place.

**Questions:**

What did you use to compute the ground truth solutions?  How long did it take to compute those ground truth solutions relative to the methods compared in the paper?  What I'd like to see here is a thoughtful comparison with conventional methods and points along the Pareto curve trading off compute time against accuracy (here, that means the fineness of discretization for, e.g., FEM).

---

> ### Author Response · Authors · 2024-11-21
>
> > **W1:** The main weakness of the paper is that we don't get any sense of how well this actually works for solving PDEs. Like many "deep learning for PDEs" papers, it does not compare to conventional methods for solving PDEs, but only other neural networks...
>
> **Response:**
> We share the reviewer's concerns about the standards for neural PDE solvers and the lack of comparison to methods largely used by the scientific community. However, we respectfully disagree that "we don't get any sense of how well this actually works for solving PDEs." We provide extensive reports on the L2 errors achieved by our approach as well as the methodology we use to compute those errors (Appendix D). This enables quantitative comparisons of our *implementation* of Alg. 1 with any other solutions (including FEM, see preliminary results below).
>
> That is not to say that the reviewer (or anyone) "should stop using [their] PDE solver." We completely agree that classical methods can achieve more precise solutions than NN-based ones, include (SCL). Yet, (SCL) enable new use cases. As we explicitly state in the manuscript (line 111): "While they [NN-based methods] may not achieve the precision of classical methods [...], they are able to provide solutions for whole families of BVPs, extrapolating new solutions from existing ones, and leverage extrinsic information, such as real-world measurements." Hence, while Alg. 1 can be used to solve single PDEs (Sec. 5.1), we believe use cases not covered by traditional methods are more interesting. In fact, they form the bulk of our experiments in Sec. 5 and Appendix E. We believe that by simultaneously solving for a wide range of PDEs, Alg. 1 can decouple the computation from the use of PDE solutions, which is of great value in engineering applications, particularly in the design phase.
>
> To showcase this fact, we provide preliminary results for solving the Helmholtz equation for a range of parameters using FEM and our method. In particular, we considered the Helmholtz equation with $(a_1, a_2) \in [1,2] \times [1,2]$ and compare SCL'(M) (see Sec. 5.2 and e.g., Table 9) with FEM:
> - SCL'(M)
>     * Average relative $L_2$ error across an equispaced grid of 10 000 PDE parameters: 0.0125
>     * Training time: 31.4 hours
> - FEM
>     * Average relative $L_2$ error across an equispaced grid of 25 PDE parameters: 0.0360
>     * Time: 1.67 hours
>
> For the above results for FEM, we computed 25 solutions corresponding to the 25 PDE parameters considered. Thus, for every new solution we wish to compute, it takes $\frac{1.67 \times 60}{25} = 4$ minutes. In contrast, SCL'(M) is trained only once and can thereafter provide new solutions very fast by making inference thourgh the trained NN. For example, if we would want to compute solutions on a dense grid of 10 000 PDE parameters as done for SCL'(M), this would take FEM approximatley 667 hours. Note that these numbers compare a highly-optimized C++ implementation of FEM with 20 years of development (implementation from Fenics) with a higher-level PyTorch implementation of Alg. 1.
>
> That being said, we appreciate the reviewer for pointing out (McGreivy & Hakim, 2024), which supports our philosophy that learning approaches should be focusing on tackling the alternative use cases we mentioned above.

---

> ### Author Response · Authors · 2024-11-21
>
> > **W2:** So we might instead take the contribution of the paper to be the guarantee, except: As stated in the paper, existing methods already have guarantees.
>
> **Response:**
> To the best of our knowledge, no prior methods guarantee the solution of BVPs (even approximately). We also could not find such claim in our manuscript. Explicitly, the contributions of our paper are (as summarized in lines 54-65):
>
> - prove that obtaining a (weak) solution of (BVP) is equivalent to solving a constrained learning problem with worst-case losses, namely (PIII) (Prop. 3.2);
> - incorporate other scientific prior knowledge to (PIII), such as structural (e.g., invariance) and observational (e.g., measurements, known solutions) information,  without resorting to specialized models or data transforms (SCL, Sec. 3.2);
> - develop a practical hybrid sampling-optimization algorithm (Alg. 1) that requires neither tuning weights to balance different objectives nor the careful selection of collocation points. Alg. 1 also yield trustworthiness measures by capturing the difficulty of fitting PDE instances or data points (Sec. 4);
> - illustrate the effectiveness (accuracy and computational cost) of this method in a diverse set of PDEs, NN architectures (e.g., MLPs and NOs), and problem types (solving a single or a parametric family of PDEs; interpolating known solutions; identifying PDE instances or data points that are difficult to fit, Sec. 5).
>
> Alg. 1 is different from prior work as it arises directly from Prop. 3.2. Prop. 3.2 proves that weak solutions of BVPs are obtained by **solving (1) constrained learning problems with (2) worst-case losses** as in (SCL). Hence, it is not enough to use only constrained or Lagrangian formulations as in (Lu et al., 2021b; Basir & Senocak, 2022) or other adaptive loss weighting schemes as in (Wang et al., 2021a; Wang et al., 2022b; Maddu et al., 2022; McClenny & Braga-Neto, 2023). Indeed, note that using a constrained formulation with fixed collocation points does not recover a solution of the convection PDE (Fig. 2). It is also not enough to use only worst-case loss approximations as in (Wang et al., 2022a; Daw et al., 2023). Indeed, R3 from (Daw et al., 2023) fails to recover a solution for the Eikonal equation (Table 1). In contrast, Alg. 1 succeeds in both cases (see Table 1).
>
> Prop. 3.2 shows that the limitations of previous approaches are not methodological, but epistemological. It is not an issue of *how* they solve the problem, but *which* problem they solve. In contrast, by (approximately) solving (SCL), Alg. 1 (approximately) solves (BVP). What is more, Alg. 1 also applies to other settings, such as solving parametric BVPs (Sec. 5.2) and interpolating existing solutions (Sec. 5.4).
>
> Prop. 3.2 also establishes the limitations of learning approaches that can only find very smooth solutions for high-dimensional state spaces (essentially, solution in the Sobolev space $W^{(d+1)/4,2}$). This can be an issue for large-scale dynamical systems, such as those found in smart grid applications, or when transforming higher-order PDEs in higher-dimensional first-order systems.

---

> ### Author Response · Authors · 2024-11-21
>
> > **W3:** The guarantee itself centers on sampling from \psi_\alpha which, as mentioned in the paper, becomes difficult exactly when the error bound becomes tight. Using Metropolis--Hastings essentially throws that guarantee away unless you can prove, e.g., uniform ergodicity of the resulting Markov chain. So in the end we are faced with what looks like just another difficult integral, just with a different form and a lack of a guarantee. Note that one could raise the same criticism of the original adversarial robustness result: sampling from little regions around the worst case inputs is just as difficult as finding them in the first place.
>
> **Response:**
> Note that we only sample from the bounded, closed domain $\bar{\Omega} \times \Pi$. Hence, the target $\psi_\alpha$ has finite tails for any $\alpha$, i.e., it has a moment generating function, satisfying sufficient conditions for uniform ergodicity (see, e.g.,  [Jarner & Hansen, "Geometric ergodicity of Metropolis algorithms," 2000]). Additionally, we prove in Prop. 3.1-3.2 that the $\psi_0$ in Alg. 1 (and more generally, $\psi_\alpha$) are square-integrable, i.e., belong to $\mathcal{P}^2$. Thus, alternative modern sampling technique, such as Langevin Monte Carlo, could be used in Alg. 1 to improve its performance (albeit at a higher computational cost). This is an important point and we will include a note to this effect in Sec. 4.
>
> The reviewer has a point that Alg. 1 approximates the solution of (SCL). That is the case of every numerical algorithm, from optimization to traditional PDE solver. Contrary to previous approaches, however, we show both that (SCL) is the *right problem* to solve, in that it provides a weak solution of the BVP (Prop. 3.2), and that Alg. 1 approximately solves it. Previous approaches do not solve (or even attempt to solve) (SCL), therefore failing to find a weak solution altogether.

---

> ### Author Response · Authors · 2024-11-21
>
> > **Q1:** What did you use to compute the ground truth solutions?
>
> **Response:**
> As we explain in Appendix D, the ground-truth solutions for the convection and reaction-diffusion PDEs were evaluated analytically; for the eikonal equation, we use the signed distance field reported in (Daw et al., 2023); the Burgers' and Navier-Stokes solutions were obtained from (Li et al., 2021) and the diffusion-sorption from (Takamoto et al., 2022).

---

> ### Author Response · Authors · 2024-11-21
>
> > **Q2:** How long did it take to compute those ground truth solutions relative to the methods compared in the paper? What I'd like to see here is a thoughtful comparison with conventional methods and points along the Pareto curve trading off compute time against accuracy (here, that means the fineness of discretization for, e.g., FEM).
>
> **Response:**
> As we mention in our response to Q1, we use published ground truth solutions so that our results are easy to compare to. Those source do not provide computation times. We do provide relative complexity comparisons in Appendix E for solving parametric families of BVPs. However, we compare the number of "bottleneck, expensive operations" (differential operator evaluations) instead of compute time, as the latter is difficult to replicate and highly dependent on hardware [as (McGreivy & Hakim, 2024) argue] and specific implementation. Still, we provide preliminary comput time results in our response to W1 (which we will include in the revised manuscript). We also reiterate that we do not believe the best use of Alg. 1 is solving single BVPs, but to target use cases for which classical methods are not well-suited (line 111).

---

> ### Author Response · Authors · 2024-11-25
>
> Following the reviewers comments, we have updated our manuscript (see main comment for a list of modifications, currently marked in blue in the paper). We hope the reviewer finds these minor additions and changes make the paper clearer and address their concerns directly in the manuscript. We are happy to provide further details and clarifications if necessary.

---

### Author Response · Authors · 2024-11-21

We thank the reviewers and AC for their time and feedback. We feel that it has greatly improved the presentation and clarity of our results. Below, we summarize the main concerns raised in the reviews. We address these in more detail in our point-by-point response to the reviewers.

### Contributions
Explicitly, the main contributions of our paper are:
- Prove that obtaining a (weak) solution of (BVP) is equivalent to solving a constrained learning problem with worst-case losses, namely (PIII) (Prop. 3.2).
- Incorporate other scientific prior knowledge to (PIII), such as structural (e.g., invariance) and observational (e.g., measurements, known solutions) information, without resorting to specialized models or data transforms (SCL, Sec. 3.2).
- Develop a practical hybrid sampling-optimization algorithm (Alg. 1) that requires neither tuning weights to balance different objectives nor the careful selection of collocation points. Alg. 1 also yields trustworthiness measures by capturing the difficulty of fitting PDE instances or data points (Sec. 4).
- Illustrate the effectiveness (accuracy and computational cost) of this method in a diverse set of PDEs, NN architectures (e.g., MLPs and NOs), and problem types (solving a single or a parametric family of PDEs; interpolating known solutions; identifying PDE instances or data points that are difficult to fit, Sec. 5).

Prop. 3.2, in particular, is key as it shows that the limitations of previous approaches are not methodological but epistemological. **It is not an issue of how they solve the problem, but which problem they solve.** Indeed, Prop. 3.2 proves that weak solutions of BVPs are obtained by **solving (i) constrained learning problems with (ii) worst-case losses.** Hence, it is not enough to use either worst-case losses as in (Wang et al., 2022a; Daw et al., 2023) or adapt the loss weights as in (Wang et al., 2021a; Wang et al., 2022b; Maddu et al., 2022; McClenny & Braga-Neto, 2023), or even using constrained formulations (Lu et al., 2021b; Basir & Senocak, 2022). In contrast, Alg. 1 tackles both (i) and (ii) by (approximately) solving (SCL), the constrained learning problem with worst-case losses of Prop. 3.2, therefore also (approximately) solving (BVP).

### Novelty of Alg. 1
As noted above, Alg. 1 tackles both (i) and (ii) and is therefore the first method to actually tackle (SCL) and, consequently, (BVP) (see Prop. 3.2). The algorithm itself is based on constrained optimization duality, so that the updates in steps 8-9 are reminiscent of classical primal-dual or dual ascent methods going back to [Arrow et al., "Studies in linear and non-linear programming," 1958]. Crucially, Alg. 1 combines such primal-dual methods with sampling-based approximations of worst-case losses (steps 4-7).

- **Solving Constrained Learning Problems** Alg. 1 and other Lagrangian-based methods such as (Lu et al., 2021b; Basir & Senocak, 2022) tackle max-min problems of the form ($\hat{\text{D}}$-CSL) from Sec. 3.1. This is **not** the same as solving (SCL), a constrained optimization problem, due to non-convexity. We overcome this issue by posing a constrained *learning* problem (i.e., with statistical losses), namely (SCL), which allows us to use non-convex duality results from (Chamon & Ribeiro, 2020; Chamon et al., 2023) to provide approximation guarantees between (P-CSL) and ($\hat{\text{D}}$-CSL). Combined with results from (Cotter et al., 2019; Elenter et al., 2024), we can obtain convergence guarantees for (3), a variant of Alg. 1, towards a (probably approximately) near-feasible and near-optimal solution of (SCL).

- **Worst-Case Losses** Using steps 8-9 with fixed collocation points (regardless of their distribution) is still not enough to obtain a solution of a BVP: Prop. 3.2 also requires the use of worst-case losses. Alg. 1 leverages Prop. 3.1 to replace these worst-case losses by statistical losses against specially designed distributions, namely the $\psi_0$ in steps 4-7. It is the combination of these two properties (duality of constrained learning problems and worst-case formulation) that ensures that Alg. 1 seeks a weak solution of the BVP.

---

> ### Author Response · Authors · 2024-11-25
> **Revised manuscript**
>
> Following the reviewers comment, we have updated our manuscript as per our responses. We have marked all changes in blue. In particular, here is a general list of modifications:
>
> - we replaced our summary of contributions at the end of the introduction by an explicit list;
> - we summarized the introductory material in Sections 2 and 3 to bring our main development (Sec. 3) earlier in the manuscript. In particular, we have moved our preliminaries on constrained learning (Sec. 3.1) to the algorithm development section (Sec. 4);
> - we included new remarks on Prop. 3.2 (now Prop. 3.1), the main theoretical contribution of the paper, taken from our responses to the reviewers;
> - we expanded the derivations of our algorithm (Sec. 4), including a more detailed development of how to go from the problem (SCL) to Alg. 1. We also included more details on generalization and convergence of MH and primal-dual methods (see also App. C and D).
> - we reinforced our previous remarks on the use cases where we believe NN-based solvers could be advantageous and included the preliminary results of our comparison to FEM solvers (Sec. 5.2);
> - we added our preliminary results for the convection PDE ("error bars" in Table 1). We continue to run new simulations and will update the results as we obtain them.
>
> We hope the reviewers will find that these additions and rearrangements make the paper clearer and address their concerns directly in the manuscript. We are happy to continue discussing if they find other points that could improve our paper.

---

### Meta-Review · Area_Chair_V8QT · 2024-12-20

**Metareview:**

This work shows that obtaining weak solutions to PDEs can be formulated as a constrained learning problem with worst case losses. Furthermore, it develops an efficient algorithm for solving this problem and incorporating further structural information that maybe be known. The method is evaluated on a variety of tasks and the comparisons are extensive.

**Additional Comments On Reviewer Discussion:**

While I also share one of the reviewers' concern that many works which solve PDEs with ML do not give a fair comparison with traditional methods,I believe that authors have managed to do this well here (in rebuttal) and their overall numerics are extensive. For some problems, it well known that traditional methods are very slow and it need not be that every new paper in this field re-iterates this point. The method proposed by the authors is, as far as I know, novel, and their results show significant improvement over other PINN-based methodologies. While I do think that general claims about beating traditional solvers should be toned down, I think the added numerics make a significant case for the publication of this work. As the first work to introduce the methodology, it cannot be expected that it will beat all traditional methods on all problems. The addition of cost/accuracy trade-off curves is the right direction for all works in this field.

---

### Decision · Program_Chairs · 2025-01-22

Accept (Poster)